# LEARNING MULTIMODAL TRAJECTORY REPRESENTA­TIONS FOR WEB AGENT PLANNING

## ABSTRACT

Trajectory data, capturing multimodal human actions and states, are pivotal for building autonomous GUI agents and transferring skills across tasks, encoding knowledge by compressing past experience into structured Markov sequences. Yet current methods for trajectory modeling remain fragmented, often relying on task-specific heuristics or textual signals. Progress on multimodal trajectories has been limited by the difficulty of representing visual information within long-step histories that exceed model context windows. Hence, how to effectively learn from multimodal trajectories remains a major and insufficiently addressed challenge amid ever-growing datasets. In this work, we introduce Multimodal Trajectory Retrieval, bridging the gap between universal retrieval and agent-centric trajectory modeling. We construct the Unified Agent Trajectory Dataset (UATD) from anno­tated demonstrations and states across diverse real-world scenarios. Building on this, we present GAE-Bench, a benchmark containing a large number of trajectory-based retrieval pairs. Our proposed GAE-Retriever, a multimodal retriever based on vision-language models that uses token selection and GradCache to optimize the contrastive objective. Over multiple web-agent datasets, it surpasses strong baselines on retrieval recall. To demonstrate potential downstream applications, we develop WebRAGent, a retrieval-augmented web agent that integrates GAE-Retriever and supports both DOM- and vision-based observations. WebRAGent proves effective on both textual and visual retrieved knowledge, achieving perfor­mance gains of over 15% vs. non-retrieval on the Online-Mind2Web benchmark.

## 1 INTRODUCTION

Human experience, meticulously recorded across diverse media like text, images, videos, and structured processes, forms a rich repository of knowledge known as *trajectories*. These trajectories encapsulate not only the actions performed but also the environmental states in which they occurred. The vast amount of trajectory data already available in human-generated content, such as instructional videos (Tang et al., 2019) and illustrated guides (Zhou et al., 2022), is continuously expanding through the efforts of AI agent researchers and the deployment of agent products. This wealth of experiential data offers significant value, not only for human reuse and learning but also for enhancing the intelligence in fields such as embodied intelligence (Yue et al., 2024) and computer-use agents (Zhao et al., 2024). Given the ever-increasing volume of trajectory data, a critical question arises: *How can we effectively model these trajectories to boost more advanced intelligence?*

Prior methodologies have explored the representation of states and actions to facilitate the retrieval of optimal actions, diverging from the generation-based approach (Yao et al., 2022). Trajectory-level experience data have shown value within in-context reasoning paradigms (Sridhar et al., 2024; Wang et al., 2024c; Liu et al., 2025a) and reinforcement training (Goyal et al., 2022; Humphreys et al., 2022; Kang et al., 2024), highlighting the rich semantic information embedded within task instructions, states, and trajectories in a latent space that can significantly aid agent inference and learning. However, trajectory-based data inherently involve substantial token consumption, making the efficient retrieval of the most pertinent trajectory data a critical challenge during the trajectory data explosion. Furthermore, state representation modeling holds immense potential for tasks such as world modeling (Gu et al., 2024) and search algorithms (Putta et al., 2024; Koh et al., 2024b), presenting a viable alternative to the conventional strategies of prediction and actual interaction with environments. Despite the promise of trajectory and state representation modeling, existing

studies lack a systematic evaluation of model capacities and a comprehensive understanding of how to scale these capacities with increasing data and task complexity. This work chooses GUI-based environments as an initial ground for exploration, driven by the practical value of web automation across diverse applications and the abundance of existing data resources, presented by pioneering works in this field such as Mind2Web (Deng et al., 2023) and WebArena (Zhou et al., 2024).

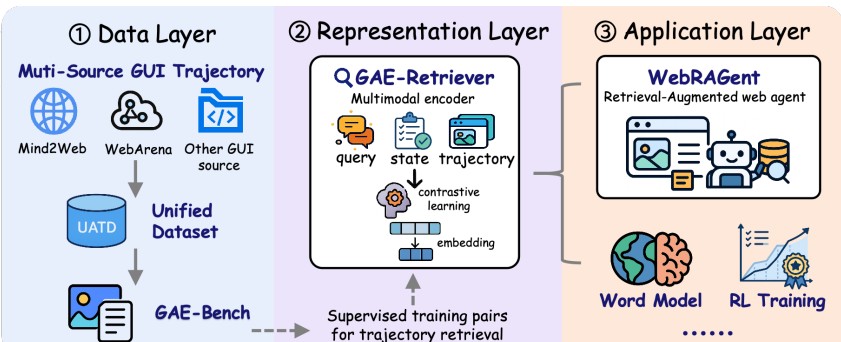

Figure 1: Overview of the proposed framework. The pipeline spans three layers: (1) Data Layer, unifying heterogeneous trajectories into **UATD**; (2) Representation Layer, where **GAE-Retriever** learns multimodal embeddings; and (3) Application Layer, featuring **WebRAGent** for retrieval-augmented planning for web navigation. Furthermore, the learned trajectory representations demonstrate promising generalization capabilities, paving the way for advanced paradigms such as world modeling and RL training.

Figure 1 illustrates that our study adopts a pipeline encompassing unified data collection, representation learning, and downstream applications. Building on this framework, we propose a multimodal trajectory retrieval as a cornerstone for future research, consisting of four main contributions:

1. To unify heterogeneous trajectories across web (current), mobile, desktop, and embodied environments (future), we construct the **Unified Agent Trajectory Dataset** (UATD), built from five open-source GUI agent datasets. UATD includes 7,747 human-annotated demonstrations and 82,793 states, covering diverse real-world use cases such as shopping, travel, business, and social media. Each trajectory features high-definition visual observations, a standardized action format, and natural language state descriptions to support customization and the development of downstream tasks.

2. We are the first to propose the task of **Multimodal Trajectory Retrieval**, bridging the gap between universal retrieval and trajectory modeling. This task captures both temporal and semantic correlations within and across trajectories, targeting fine-grained intra-trajectory components as well as coarse-grained inter-trajectory semantic relations. To formalize the task, we design 12 extraction patterns that derive six types of retrieval samples from a single trajectory, where the query and target can be a state, a trajectory, or a subtrajectory. These include (1) text-to-state, (2) text-to-trajectory, (3) state-to-state, (4) state-to-trajectory, (5) trajectory-to-state, and (6) trajectory-to-trajectory retrieval.

Based on this formulation, we annotate the **GUI Agent Embedding Benchmark** (GAE-Bench) by converting the UATD using the predefined extraction patterns, yielding a total of 714,628 positive retrieval pairs. To accommodate the context length limitations of current multimodal language models and enhance applicable use in downstream applications, we also release GAE-Bench-lite, a constrained version in which trajectory lengths are capped at 10 steps. It contains 514,956 training samples, 21,805 in-domain samples, and 27,139 out-of-domain samples, with 574,222 candidates.

3. We implement the **GUI Agent Embedding Retriever** (GAE-Retriever), a novel multimodal trajectory retrieval framework built on VLM2Vec (Jiang et al., 2025), which adopts vision-language models (VLMs) as its backbone. In contrast to CLIP-based models (Radford et al., 2021; Li et al., 2022; 2023), VLMs are pretrained on large-scale, instruction-following multimodal data and can process arbitrary-length combinations of visual and textual inputs, making them well-suited for modeling multimodal trajectories. To enable effective contrastive learning over multiple high-resolution trajectory screenshots and a large number of in-batch negatives under limited computing resources, GAE-Retriever incorporates a token selection mechanism and GradCache optimization.

4. We propose **WebRAGent**, a retrieval-augmented multimodal web agent framework that integrates the GAE-Retriever. WebRAGent supports both DOM and Vision modes and comprises observation, retrieval, memory, planning, and reward modules. WebRAGent applies the GAE-Retriever to encode

the task query and the current state screenshot into embeddings, retrieving step-level guidance for the next action. We validate this multimodal, model-based retrieval approach on the Online-Mind2Web benchmark (Xue et al., 2025), where it improves success rate by **15-22%** over non-retrieval baselines and yields larger gains on hard tasks. Finally, we analyze the respective contributions of single-modal and multi-modal in-context learning to retrieval-augmented agents.

We conduct comprehensive evaluations of GAE-Retriever against multimodal backbone models, retrieval models, and trajectory planning models, showcasing its strong performance across all baselines on Recall@1/5/10. Compared to the best-performing baselines, GAE-Retriever achieves an average improvement of 10.22 points across five datasets, demonstrating the effectiveness of our training approach. Beyond this, our work establishes a bridge between retrieval and execution by effectively preserving and leveraging knowledge from similar trajectories. We further demonstrate the generalization and effectiveness of WebRAGent on the Online-Mind2Web benchmark.

## 2 RELATED WORK

### 2.1 GUI AGENTS

The evolution of language models has introduced strong capabilities in tool use, environmental grounding, and complex reasoning for agentic tasks (Su et al., 2024), such as web browsing (Yao et al., 2022), travel planning (Xie et al., 2024a; Zhang et al., 2024d), and societal simulation (Park et al., 2023). GUI navigation (Zhang et al., 2024a), originally derived from web-based tasks, has become an active area of research across domains including web (Deng et al., 2023; Zhou et al., 2024; Koh et al., 2024a), mobile (Rawles et al., 2023; 2024), and computer control (Xie et al., 2024b).

To better reflect realistic conditions, GUI agents have evolved from reactive systems (Lee et al., 2023; Li et al., 2024), to proactive agents that take actions based on conversational understanding or situational reasoning. These two paradigms have led to distinct approaches: user interaction through dialogue (Deng et al., 2024), environment-based reasoning (Yao et al., 2023), and tree search (Koh et al., 2024b). However, real-time exploration can be inefficient and may sacrifice user experience. To reduce this overhead, some approaches retrieve reusable subroutines (Wang et al., 2024c; Zheng et al., 2025) or demonstration histories from memory (Kagaya et al., 2024; Liu et al., 2025a). Nonetheless, they perform similarity search using only textual features, neglecting richer multimodal signals.

### 2.2 MULTIMODAL RETRIEVAL

Recent advances in foundational vision-language models (Liu et al., 2023; Peng et al., 2023) have shifted research focus from generation to retrieval. Early approaches (Muennighoff et al., 2023; Fu et al., 2023) primarily addressed single-modality retrieval tasks such as text-to-image and text-to-text retrieval. More recent efforts (Wei et al., 2024; Jiang et al., 2024; 2025; Zhang et al., 2024b; Lin et al., 2024b) have expanded to composed and cross-modal retrieval settings, constructing large-scale benchmarks from vision-language datasets that extend to classification and grounding tasks.

Recent studies have targeted more practical domains, including visual documents (Faysse et al., 2024), videos (Wang et al., 2024b), and screenshots (Liu et al., 2025b). VLM2Vec-V2 (Meng et al., 2025), for instance, integrates data from diverse modalities to learn universal discriminative embeddings. Despite substantial progress in multimodal retrieval, few works have explored integrating trajectory data into retrieval model training. Notably, contemporary findings (Fu et al., 2024; Sridhar et al., 2024; Yue et al., 2024) highlight the potential of agent trajectories for downstream in-context planning tasks, underscoring a promising but underexplored direction in this research area.

While retrieval is a long-standing topic in multimodal learning, our focus is on autonomous GUI agents, where most systems remain purely reactive and predict the next action directly from previous observations without an explicit trajectory memory. In contrast, we advocate retrieval-augmented planning with explicit long-horizon trajectory representations as a queryable memory, and our multi-image, high-resolution states ($\sim 1000 \times 1000$) fall outside the standard CLIP-style single-image $224 \times 224$ (or extended composed video) setting, motivating a dedicated trajectory retriever.

# 3 DATASET

To build datasets for the multimodal trajectory retrieval task, a unified agent trajectory format is required for generating valid positive and negative samples. Current datasets, whether sourced from digital environments (Zhou et al., 2024; Chai et al., 2024) or embodied platforms (Shridhar et al., 2021), encode trajectories using heterogeneous structures, making consistent retrieval data extraction nontrivial. To address this challenge, we design a pipeline that converts these datasets into a standardized format, resulting in the **U**nified **A**gent **T**rajectory **D**ataset (UATD). From UATD, we define 12 extraction schemes that convert each individual trajectory into labeled samples covering six core retrieval tasks, forming the **G**UI **A**gent **E**mbedding **Bench**mark (GAE-Bench) series.

## 3.1 UATD

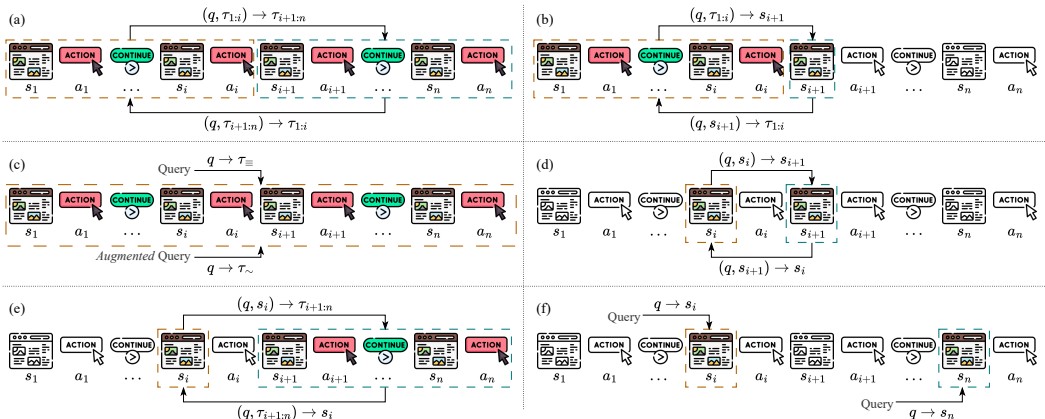

Figure 2: Illustration of positive pair extraction from UATD. Subfigures (a), (b), (d), and (e) depict temporal retrieval; (c) and (f) show semantic retrieval. We use $\tau_{i:j}$ to denote a subsequence of trajectory $\tau$ from state $s_i$ to action $a_j$, i.e., $\tau_{i:j} = (s_i, a_i, \ldots, s_j, a_j)$. $q$ serves as the retrieval query, composed of a task-specific instruction (shown in Table 1) along with a corresponding trajectory description.

Table 1: The overview of the GUI Agent Embedding Benchmark (GAE-Bench) series. This table summarizes the six fundamental retrieval tasks and their twelve corresponding subtasks, developed from the predefined extraction schemes in Figure 2. For every subtask, we present a representative instruction (see Appendix E.1 for more instruction templates). The table also reports the number of subtask examples in GAE-Bench and the split statistics in GAE-Bench-lite: training, in-domain (IND), out-of-domain (OOD), and total instances.

| Task | Subtask | Instruction (shown 1 out of 10) | GAE-Bench | GAE-Bench-lite | | | |
|---|---|---|---|---|---|---|---|
| | | | | Train | IND | OOD | Total |
| 1. $(q,\tau) \to \tau'$ | $(q, \tau_{1:i}) \to \tau_{i+1:n}$ | Apply the request to the previous web navigation steps to derive the next trajectory. | 75,046 | 40,748 | 2,523 | 2,665 | 45,936 |
| | $(q, \tau_{i+1:n}) \to \tau_{1:i}$ | Find the previous web browsing trajectory based on the user input and the current trajectory. | 75,046 | 40,748 | 723 | 2,665 | 44,136 |
| 2. $(q,\tau) \to s$ | $(q, \tau_{1:i}) \to s_{i+1}$ | Locate the following state from the given former web navigation trajectory and the task input. | 75,046 | 51,661 | 3,197 | 2,665 | 57,523 |
| | $(q, \tau_{i+1:n}) \to s_i$ | Locate the former observation using the instruction and the upcoming web interaction trajectory. | 75,046 | 51,661 | 3,197 | 2,665 | 57,523 |
| 3. $q \to \tau$ | $q \to \tau_{\equiv}$ | Identify the unique navigation trajectory for web agents according to the provided instruction. | 7,747 | 3,717 | 424 | 470 | 4,611 |
| | $q \to \tau_{\sim}$ | Retrieve the analogous interaction history for GUI agents based on the provided command. | 38,735 | 20,081 | 624 | 2,350 | 23,055 |
| 4. $(q,s) \to s'$ | $(q, s_i) \to s_{i+1}$ | Using the provided instruction and the former state, what is the next GUI navigation state? | 75,046 | 68,849 | 3,532 | 2,665 | 75,046 |
| | $(q, s_{i+1}) \to s_i$ | Taking the description and the current state into account, search the previous web agent state. | 75,046 | 68,849 | 3,532 | 2,665 | 75,046 |
| 5. $(q,s) \to \tau$ | $(q, s_i) \to \tau_{i+1:n}$ | Locate the next GUI navigation trajectory by applying the instruction to the previous state. | 75,046 | 54,124 | 734 | 2,665 | 57,523 |
| | $(q, s_{i+1}) \to \tau_{1:i}$ | Identify the web navigation trajectory preceding the current state according to the task. | 75,046 | 54,124 | 734 | 2,665 | 57,523 |
| 6. $q \to s$ | $q \to s_i$ | From the description, identify the specific navigation observation for GUI navigation. | 60,031 | 53,892 | 3,610 | 2,529 | 60,031 |
| | $q \to s_n$ | Search the terminal observation in the web navigation for the task instruction. | 7,747 | 6,502 | 775 | 470 | 7,747 |

To identify the essential element in trajectory representations, we model a typical agent interacting with an environment as a deterministic Markov Decision Process (MDP) $\mathcal{E} = (\mathcal{S}, \mathcal{A}, \mathcal{O}, \mathcal{T})$, where $\mathcal{S}$ denotes the state space, $\mathcal{A}$ the action space, $\mathcal{O}$ the observation space, and $\mathcal{T} : \mathcal{S} \times \mathcal{A} \to \mathcal{S}$ is the environment transition function. A trajectory is thus represented as a sequence of state $s_i$ and action $a_i$ with $n$ steps, $\tau = (s_1, a_1, s_2, a_2, \ldots, s_n, a_n)$, where each transition satisfies $s_{i+1} = \mathcal{T}(s_i, a_i)$.

Inspired by AGUVIS (Xu et al., 2024), we take each observation $o_i \in \mathcal{O}$ to be the raw visual content of the interface (i.e., a screenshot). This choice eliminates reliance on platform-specific textual representations and promotes broader generalization across visual contexts. For each action $a_i \in \mathcal{A}$, we stipulate three components: (1) operation: the name or type of the action; (2) target: the object or region within the environment on which the action is executed; (3) value: additional arguments required to perform the action. To address cross-platform discrepancies in action spaces that hinder

scalability, we follow the action README from ShowUI (Lin et al., 2024a), allowing each trajectory to be associated with its own customizable textual action definitions.

To create the dataset under a uniform trajectory representation, we collect real-world human trajectories with their action definitions from five existing GUI data sources. The details of dataset construction and statistics are provided in Appendix B.1.

## 3.2 GAE-BENCH

After acquiring the unified trajectory dataset, we introduce 12 extraction patterns to generate ground truth retrieval pairs from each trajectory based on a query $q$, as depicted in Figure 2. These patterns encompass two categories of multimodal trajectory retrieval: **temporal retrieval** (Task 1, 2, 4, and 5 in Table 1), which captures sequential relationships within a trajectory, and **semantic retrieval** (Task 3 and 6 in Table 1), which targets underlying intent or operation over different trajectories and states.

Figure 2 subfigures (a) and (d) detail scenarios to retrieve the next state $s_{i+1}$ or the remaining trajectory $\tau_{i+1:n}$ given a current state $s_i$ or a prefix trajectory $\tau_{1:i}$, and vice versa. Subfigures (b) and (e) demonstrate situations involving different granularities, such as identifying the correspondence between the partial trajectory $\tau_{1:i}$ and the next state $s_{i+1}$, or between the present state $s_i$ and a remaining trajectory $\tau_{i+1:n}$. These positive pairs capture sequential relations within a trajectory.

Figure 2 subfigures (c) and (f) expound two types of semantic retrieval tasks. In particular, subfigure (c) defines retrieval between a query and its reference-aligned trajectory (known as the gold trajectory), denoted as $q \rightarrow \tau_{\equiv}$, which is directly from the UATD ground truth. Conversely, subfigure (f) constructs retrieval pairs between a query and a semantically similar variant (known as the silver trajectory), denoted as $q \rightarrow \tau_{\sim}$. This represents a trajectory that preserves the same high-level intent and functional objective but varies in contextual details. For instance, the silver trajectory corresponding to "Buy a t-shirt for children on Amazon" could be "Order a laser printer on eBay".

Because generating silver trajectories is more challenging than augmenting queries, we design a three-step procedure to produce silver instructions for trajectories: (1) Identify named entities in the original query using named entity recognition; (2) Generate alternative expressions for the identified entities while preserving their types; (3) Rewrite the original query after substituting the identified entities with the corresponding alternatives. Appendix E.1 outlines the prompts for silver generation.

To implement $q \rightarrow s_i$ and $q \rightarrow s_n$ in subfigure (f), state descriptions from UATD are utilized to formulate retrieval of a specific intermediate state $s_i$ and the final state $s_n$.

Table 2: Distribution of GAE-Bench by retrieval tasks and data sources (left; 714,628 positive pairs total) and its candidate sets (right). For each of the data sources, we define three types of candidate sets (i.e., *state*, *trajectory*, and *interval*), corresponding to retrieval targets of a single state $s_i$, a full trajectory $\tau$, and a trajectory subsequence $\tau_{i:j}$. Note that $\tau_{i:j}$ can range in length from a single state-action pair to the entire trajectory.

| | Retrieval Tasks | | | | | | | | Candidate Sets | | | |
|---|---|---|---|---|---|---|---|---|---|---|---|---|
| Data Source | $(q,\tau)\rightarrow\tau'$ | $(q,\tau)\rightarrow s$ | $q\rightarrow\tau$ | $(q,s)\rightarrow s'$ | $(q,s)\rightarrow\tau$ | $q\rightarrow s$ | Total | | State | Trajectory | Interval | Total |
| **Mind2Web** | 16,306 | 16,306 | 8,808 | 16,306 | 16,306 | 10,943 | **84,975** | | 9,475 | 1,468 | 48,238 | **59,181** |
| **AutoWebGLM** | 960 | 960 | 840 | 960 | 960 | 760 | **5,440** | | 620 | 140 | 2,980 | **3,740** |
| **WebArena** | 2,040 | 2,040 | 1,206 | 2,040 | 2,040 | 1,305 | **10,671** | | 1,104 | 201 | 6,640 | **7,945** |
| **WebLINX** | 12,794 | 12,794 | 2,910 | 12,794 | 12,794 | 6,337 | **60,423** | | 5,852 | 485 | 76,095 | **82,432** |
| **GUIAct** | 117,992 | 117,992 | 32,718 | 117,992 | 117,992 | 48,433 | **553,119** | | 42,980 | 5,453 | 529,528 | **577,961** |

Table 1 outlines the extracted positive pairs organized into six retrieval tasks and twelve subtasks that comprise GAE-Bench. To accommodate the limited context window of current multimodal models and maintain the feasibility of training, we also release GAE-Bench-lite, a constrained version of GAE-Bench in which each trajectory sequence contains fewer than 10 steps. To construct this version, we first randomly sample trajectories to form the out-of-domain evaluation subset. From the remaining positive pairs, we then partition the in-domain evaluation set and the training set, with approximately 90% of the examples assigned to the training set. To support efficient and applicable encoding during evaluation, we additionally introduce a mini partition of the GAE-Bench-lite candidate sets, referred to as GAE-Bench-lite (mini), which contains only candidates relevant to the evaluation subset. Detailed distributions of GAE-Bench is displayed in Table 2. See Appendix B.2 for details of GAE-Bench-lite and GAE-Bench-lite (mini).

## 4 FRAMEWORK

This section presents the problem definition for the multimodal trajectory retrieval, along with our model, GAE-Retriever, which extends VLM2Vec for trajectory learning.

### 4.1 PROBLEM DEFINITION

In real-use cases, trajectories may be gathered either as complete sequences $\tau$ or as partial segments, such as individual observations $s_i$ or trajectory subsequences $\tau_{i:j}$. This gives rise to diverse types of retrieval queries and candidate sets, as outlined in Table 1. The retrieval key (i.e., the input) $\mathbf{k}$ may consist of a textual query, a state, a full trajectory, or a trajectory subsequence. Similarly, the retrieval value (i.e., the target) $\mathbf{v}$ can take the form of a state, a complete trajectory sequence, or a subsequence.

$$
\begin{aligned}
s_i &\to \texttt{Observation:}\ [\texttt{image}_i] \\
a_i &\to [\texttt{action}_i] \\
u_{i:j} &\to [\texttt{action space}]\ \tau_{i:j} \\
\tau_{i:j} &\to \begin{cases} s_i\ a_i, & \text{if } i = j, \\ s_i\ a_i\ \tau_{i+1:j}, & \text{if } i < j \end{cases} \\
\mathbf{k} &\to \tilde{q}\ \mathbf{v} \\
\mathbf{v} &\to s_i\ \mid\ u_{i:j}
\end{aligned}
$$

Figure 3: Recursive grammar definition for model input.

To handle different retrieval intentions, we formulate the structural definition of retrieval keys and values for model input using the following recursive grammar in Figure 3.

In this grammar, text in teletype font (like `Observation:`) denotes terminal symbols, and square brackets [·] indicate token placeholders. An $n$-step trajectory is denoted by $\tau_{1:n}$, where $\tau_{i:i}$ is a state-action pair at step $i$, and $u_{i:j}$ refers to $\tau_{i:j}$ with action definitions. $\tilde{q}$ denotes $q$ with a task-specific instruction (refers to Table 1) and contextual trajectory description.

Formally, we can build a multimodal trajectory retrieval model $f$ that accepts any type of valid key and returns any type of value as specified by instruction $\tilde{q}$:

$$
\mathbf{v}^* = \underset{\mathbf{v} \in \mathcal{V}}{\arg\max}[f(\mathbf{k})^{\mathsf{T}} \cdot f(\mathbf{v})]
$$

where $\mathcal{V}$ corresponds to the heterogeneous set of candidates, $f(\cdot)$ is the retrieval function optimized via maximum dot-product similarity, and $\mathbf{v}^*$ is the predicted result.

### 4.2 GAE-RETRIEVER

We implement GAE-Retriever, a contrastive retrieval model implemented on top of VLM2Vec (Jiang et al., 2025) for multimodal trajectory tasks, as portrayed in Figure 4. The main technical challenge in modeling GAE-Retriever lies in memory constraints. Encoding these regions introduces excessive visual tokens, leading to higher inefficient computational costs and limited scalability, particularly as trajectory lengths increase. To address this, we follow prior work (Raposo et al., 2024; Lin et al., 2024a) and apply token selection by constructing a UI-connected graph in RGB space to guide attention toward salient elements while skipping redundant tokens. Specifically, adjacent image patches with differences below a threshold are merged into a single token, reducing redundant attention computation and improving inference efficiency.

In our multimodal trajectory retrieval task, obtaining hard negatives is difficult, thus making large batch sizes essential for learning high-quality embeddings through contrastive learning. However, GPU memory limitations constrain the batch size and the number of in-batch negatives, especially since each training instance may include multiple high-resolution images, increasing memory consumption. To address this, we adopt GradCache (Gao et al., 2021), as in VLM2Vec, a gradient caching method that decouples backpropagation between the encoder and the contrastive loss to support larger batch training. Appendix C.1 presents the action JSON schema and examples.

Given a pretrained VLM, we feed $\mathbf{k}$ and $\mathbf{v}$ to obtain embeddings, $f(\mathbf{k})$ and $f(\mathbf{v})$, by taking the final layer representation of the last token. In order to train the model, we minimize the InfoNCE loss:

$$
\mathcal{L} = -\log \frac{\phi\left(f(\mathbf{k})^{\mathsf{T}} f(\mathbf{v}^+)\right)}{\sum_{\mathbf{v} \in \mathcal{B}} \phi\left(f(\mathbf{k})^{\mathsf{T}} f(\mathbf{v})\right)}
$$

where $\mathcal{B}$ is the set of in-batch candidates, $\mathbf{v}^+$ is the positive sample associated with the retrieval key $\mathbf{k}$, $t$ is the temperature parameter, and $\phi(\cdot)$ is the temperature-scaled exponential function defined as $\phi(x) = \exp\left(\frac{x}{t}\right)$. Here, the in-batch set $\mathcal{B}$ comprises the positive $\mathbf{v}^+$ and all other value embeddings from the current mini-batch; for a given key $\mathbf{k}$, elements in $\mathcal{B} \backslash \{\mathbf{v}^+\}$ serve as implicit negatives.

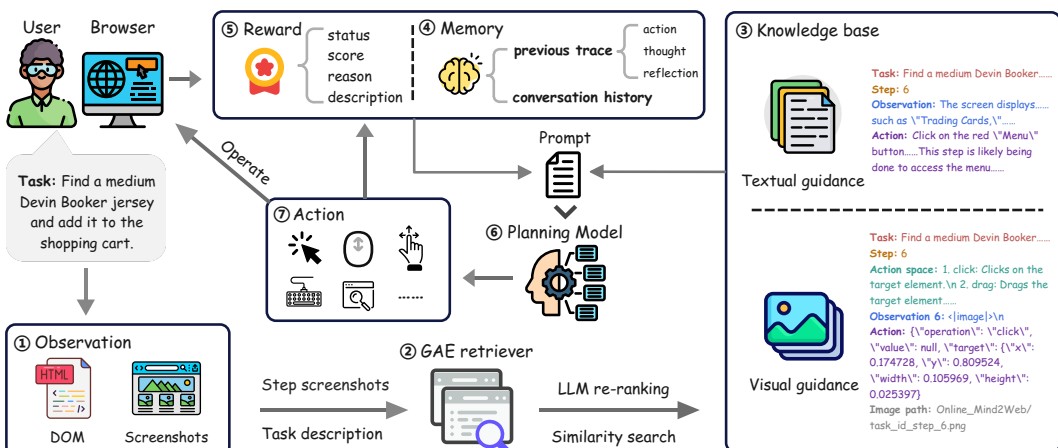

Figure 4: Illustration of GAE-Retriever on the subtask $(q, s_i) \rightarrow \tau_{i+1:n}$. Given a retrieval query and either a visual observation (left, used as the query) or a sequence of observation-action pairs (right, used as the key), GAE-Retriever employs a VLM backbone to process multimodal inputs, filtering out redundant visual tokens through token selection (red blocks in observation indicate similar token clusters). Training is performed via contrastive loss between query **k** and target **v** sides, conditioned on task-specific instructions.

## 4.3 WEBRAGENT: RETRIEVAL-AUGMENTED GENERATION FOR WEB NAVIGATION

Figure 5: WebRAGent workflow. The system supports two observation modes, DOM or page screenshots ①. Before each decision, WebRAGent uses GAE-Retriever ② to encode the task instruction and step screenshot into a query. Next, it retrieves from the knowledge base ③ step-relevant textual or visual guidance. The retrieved knowledge, with memory ④ and prior reward feedback ⑤, is composed into prompts for the planning model ⑥, resulting in actions ⑦. This design applies external experience for decision-making and execution.

We introduce WebRAGent, a framework that integrates multimodal retrieval augmentation with a web agent (see Figure 5). Given a web navigation task $q$, the agent operates in an environment $\mathcal{E} = (\mathcal{S}, \mathcal{A}, \mathcal{O}, \mathcal{T})$. At step $i$, the agent in state $s_i \in \mathcal{S}$ receives an observation $o_i \in \mathcal{O}$ and selects an action $a_i \in \mathcal{A}$; the environment transition function $\mathcal{T}$ then yields the next state $s_{i+1} = \mathcal{T}(s_i, a_i)$. Each component in the framework is elaborated below. Further details are shown in Appendix C.3.

**Observation.** (1) DOM mode: interactive DOM tree elements; (2) Vision mode: screenshot.

**Step-Level Knowledge Retrieval.** The GAE-Retriever model retrieves the next state $s_{i+1}$ from the current state $s_i$ and user query $q$, i.e., $(q, s_i) \rightarrow s_{i+1}$, shown in Figure 2(d). To further improve matching precision, we apply an LLM re-ranking stage. We align each step in the UATD trajectories by pairing every observation-action tuple $(\tilde{o}, \tilde{a})$. Based on these aligned pairs, we construct two knowledge bases: (1) a *textual knowledge base* $\mathcal{D}_{\text{text}}$, where each element $t \in \mathcal{D}_{\text{text}}$, $t = (\tilde{o}_{\text{text}}, \tilde{a}_{\text{text}}, \mu)$, with $\tilde{o}_{\text{text}}$ and $\tilde{a}_{\text{text}}$ denoting descriptions of the observation and action, and $\mu$ denoting associated metadata; (2) a *visual knowledge base* $\mathcal{D}_{\text{vision}}$, where each element $v \in \mathcal{D}_{\text{vision}}$, $v = (\tilde{o}_{\text{vision}}, \tilde{a}_{\text{str}}, \mu)$, with $\tilde{o}_{\text{vision}}$ denoting an exemplar screenshot and $\tilde{a}_{\text{str}}$ a structured representation of the action.

**Memory and Reward.** We formalize the step-$i$ cycle of Thought–Action–Reflection–Reward as the historical memory $M_i$. The reward module adopts an *LLM-as-judge* scheme to evaluate the preceding trajectory, yielding a score (1–10), the grading rationale, and a follow-up plan, denoted by $R_i$.

**Planning and Action.** We use modality-specific planners $\pi$ for different observation modes. More details for planning models are shown in Appendix C.3. Given task instruction $q$, observation $o_i$, memory $M_i$, reward $R_i$, and retrieved guidance $v$, the model outputs the action $a_i = \pi(q, o_i, M_i, R_i, v)$.

# 5 EXPERIMENTS

Table 3: Overall evaluation of all methods on each data source, reported in Recall@1/5/10.

| Method | Mind2Web | | | AutoWebGLM | | | WebArena | | | WebLINX | | | GUIAct | | |
|---|---|---|---|---|---|---|---|---|---|---|---|---|---|---|---|
| | R@1 | R@5 | R@10 | R@1 | R@5 | R@10 | R@1 | R@5 | R@10 | R@1 | R@5 | R@10 | R@1 | R@5 | R@10 |
| **Multimodal Backbone Models** | | | | | | | | | | | | | | | |
| Qwen2-VL-2B (Wang et al., 2024a) | 0.7 | 14.5 | 18.2 | 1.2 | 6.3 | 10.7 | 1.4 | 8.8 | 12.2 | 3.1 | 14.2 | 18.0 | 3.1 | 8.1 | 9.4 |
| Qwen2.5-VL-3B (Bai et al., 2025) | 1.0 | 7.8 | 9.7 | 0.9 | 3.8 | 6.3 | 0.7 | 6.5 | 9.9 | 3.4 | 13.0 | 15.9 | 3.0 | 7.9 | 9.5 |
| **Multimodal Retrieval Models** | | | | | | | | | | | | | | | |
| LamRA-Ret (Liu et al., 2024) | 1.1 | 15.1 | 19.2 | 4.9 | 15.2 | 22.8 | 2.0 | 10.3 | 14.5 | 3.4 | 16.9 | 21.5 | 4.2 | 10.4 | 12.4 |
| ColQwen2-v1.0 (Faysse et al., 2024) | 3.2 | 22.0 | 29.9 | 3.9 | 17.7 | 26.3 | 2.9 | 13.7 | 20.0 | 4.2 | 19.6 | 25.1 | 6.2 | 15.5 | 19.2 |
| GME-Qwen2VL-2B (Zhang et al., 2024c) | 3.7 | 24.2 | 33.4 | 8.7 | 27.9 | 37.4 | 4.2 | 17.7 | 24.7 | 5.2 | 22.4 | 29.7 | 6.0 | 16.7 | 20.7 |
| UniSE-MLLM (Liu et al., 2025b) | 0.8 | 12.6 | 16.0 | 0.3 | 4.9 | 9.5 | 1.2 | 9.0 | 12.8 | 2.9 | 14.2 | 17.6 | 3.1 | 7.9 | 9.7 |
| VLM2Vec-Qwen2VL-2B (Jiang et al., 2025) | 6.7 | 37.5 | 51.5 | 12.7 | 41.8 | 54.1 | 5.5 | 22.5 | 31.3 | 6.7 | 27.6 | 38.3 | 7.0 | 20.2 | 25.6 |
| VLM2Vec-V2.2 (Meng et al., 2025) | 10.2 | 44.0 | 60.1 | 15.7 | 51.2 | 67.1 | 9.1 | 29.1 | 37.8 | 10.7 | 38.4 | 50.5 | 12.2 | 33.1 | 40.6 |
| **Multimodal Trajectory Planning Models** | | | | | | | | | | | | | | | |
| UGround-V1-2B (Gou et al., 2024) | 0.8 | 12.6 | 16.0 | 0.3 | 4.9 | 9.5 | 1.2 | 9.0 | 12.8 | 2.9 | 14.2 | 17.6 | 3.1 | 7.9 | 9.7 |
| ShowUI-2B (Lin et al., 2024a) | 1.0 | 13.3 | 17.0 | 0.8 | 6.0 | 8.2 | 1.6 | 8.5 | 11.7 | 3.3 | 13.7 | 17.3 | 3.1 | 7.9 | 9.2 |
| UI-TARS-2B-SFT (Qin et al., 2025) | 0.7 | 12.5 | 15.6 | 0.6 | 4.8 | 8.4 | 1.1 | 7.4 | 11.2 | 3.0 | 13.7 | 17.3 | 3.1 | 8.0 | 9.4 |
| TongUI-3B (Zhang et al., 2025) | 1.3 | 9.3 | 11.4 | 0.5 | 3.9 | 7.3 | 1.5 | 7.0 | 10.5 | 3.4 | 13.8 | 17.4 | 3.0 | 8.1 | 9.7 |
| **Multimodal Trajectory Retrieval Models** | | | | | | | | | | | | | | | |
| **GAE-Retriever (Ours)** | **15.0** | **50.7** | **67.6** | **22.1** | **63.6** | **76.3** | **10.3** | **31.7** | **44.1** | **13.7** | **41.7** | **54.1** | **25.7** | **59.2** | **67.9** |

This section showcases our experimental setup and analyzes both aggregate and task-specific level results. Subtask-level performance details are available in Appendix D.1.

## 5.1 EXPERIMENTAL SETUPS

**Baselines** We evaluate three groups of zero-shot baselines: (1) **Multimodal Backbone**: Qwen2-VL (Wang et al., 2024a) and Qwen2.5-VL (Bai et al., 2025); (2) **Multimodal Retrieval**: VLM2Vec-Qwen2VL-2B (Jiang et al., 2025), UniSE-MLLM (Liu et al., 2025b), etc.; (3) **Multimodal Trajectory Planning**: UGround-V1-2B (Gou et al., 2024), UI-TARS-2B-SFT (Qin et al., 2025), ShowUI-2B (Lin et al., 2024a), etc. Model descriptions and evaluation setup are in Appendix D.2.

**Implementation Details** Data are annotated with `gpt-4o-mini-2024-07-18`. We train a LoRA-tuned Qwen2-VL-Instruct retriever on GAE-Bench-lite for 256 steps using GradCache and long-context inputs on 16×H800 GPUs; UI-graph token selection is enabled only during training for positional consistency. For more implementation details, please refers to Appendix C.2.

## 5.2 OVERALL EVALUATION

Table 3 summarizes Recall@1/5/10 of all baseline models across five datasets. Our proposed method, *GAE-Retriever*, surpasses all baselines by a substantial margin on all evaluation metrics. Its consistently outstanding performance over all datasets reflects the robustness and reliability of the proposed benchmark for evaluating trajectory retrieval. A task-wise breakdown (Recall@5 by task under in-domain and out-of-domain settings) is provided in Appendix D.4 and visualized in Figure 8.

**Multimodal Backbone Models** The two backbone models, Qwen2-VL-2B and Qwen2.5-VL-3B, display relatively low retrieval performance compared to their understanding and reasoning skills. Their Recall@1 scores remain below 4.0 across all datasets, indicating the need for task-specific adaptation to develop effective retrieval abilities. Interestingly, Qwen2.5-VL-3B performs slightly worse than the smaller Qwen2-VL-2B on most Recall@K metrics, underscoring a substantial gap between current multimodal LLM pretraining and the demands of trajectory-based retrieval.

**Multimodal Retrieval Models** Across retrieval-focused baselines, **VLM2Vec-v2.2** attains the best and most stable results, consistently surpassing **VLM2Vec-Qwen2VL-2B**. Overall, the VLM2Vec family outperforms other methods, suggesting that diverse fused-modality training with interleaved batches is key to retrieval effectiveness. In contrast, **LamRA-Ret** (text-image only) trails models exposed to structured visual content, such as **ColQwen2-v1.0** and **GME-Qwen2VL-2B**. **GME-Qwen2VL-2B** further outperforms **ColQwen2-v1.0**, while **UniSE-MLLM** (screenshot-focused) is the weakest, sometimes below backbone baselines. Further analysis can be found in Appendix D.3.

**Multimodal Trajectory Planning Models**   Methods in this group (e.g., ShowUI-2B, TongUI-3B, UGround-V1-2B) show no significant performance improvement over the backbone models. Their modest retrieval results imply that capabilities in planning or grounding generation do not directly translate to multimodal retrieval proficiency. Additionally, the difference in model size, i.e., 2B vs. 3B, appears to have minimal impact in our experiment.

**Trajectory Retrieval Models**   GAE-Retriever achieves the best results across all datasets and evaluation metrics. Notably, it reaches a Recall@10 of 76.3 on AutoWebGLM and 67.9 on GUIAct, with Recall@1 consistently above 10.0 across all benchmarks. Compared to the strongest baseline, VLM2Vec-V2.2, GAE-Retriever improves Recall@1 by up to 7.1 points on GUIAct and 6.4 points on AutoWebGLM. These gains validate the design of our training strategy.

## 5.3   ONLINE WEB TASK EVALUATION

Our evaluation is conducted on Online-Mind2Web (Xue et al., 2025), which is markedly more challenging for task execution and evaluation than offline benchmarks such as WebArena (Zhou et al., 2024). In total, we select 100 tasks spanning 51 websites for comparative evaluation; the full task list and website details are provided in Appendix B.3. Experiments are performed under both DOM and Vision observation modes across three representative methods. Figure 6 presents the overall task success rates and the average number of tokens consumed per step, while Table 4 reports task success rates and average execution steps broken down by task difficulty levels. Furthermore, Appendix D.7 analyzes the inference efficiency of incorporating GAE-Retriever: each step adds ∼10% latency (+0.38sec), the reranking stage takes 3.5sec, and enabling caching yields a 9.24× speedup.

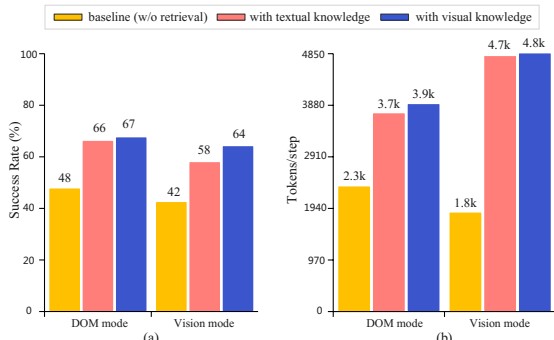

Figure 6: Online-Mind2Web evaluation: baseline vs. models augmented with textual/visual knowledge.

Table 4: Results in DOM and Vision modes.

| Method | Success Rate (%) | | | #Average | |
| | Easy | Medium | Hard | Step | Reward |
| --- | --- | --- | --- | --- | --- |
| **DOM mode** | | | | | |
| baseline | 70.45 | 35.89 | 21.42 | 8.72 | 7.60 |
| + $\mathcal{D}_{\text{text}}$ | **79.54** | **61.53** | 35.71 | 7.47 | **8.19** |
| + $\mathcal{D}_{\text{vision}}$ | 77.27 | 58.97 | **64.28** | 10.63 | 7.96 |
| **Vision mode** | | | | | |
| baseline | 59.09 | 28.20 | 28.57 | 14.88 | 7.35 |
| + $\mathcal{D}_{\text{text}}$ | 77.27 | 43.58 | **35.71** | 16.37 | 8.08 |
| + $\mathcal{D}_{\text{vision}}$ | **79.50** | **56.41** | 35.71 | 19.85 | **8.29** |
| **SeeAct** | | | | | |
| SeeAct | 71.42 | 47.36 | 6.25 | 18.45 | - |
| + $\mathcal{D}_{\text{text}}$ | **78.57** | 55.26 | 18.75 | 22.87 | - |
| + $\mathcal{D}_{\text{vision}}$ | 76.19 | **60.52** | **25.00** | 21.92 | - |

**Why Retrieval Matters: Multimodal Retrieval vs. None-Retrieval Baseline**   Compared with the non-retrieval setting, the multimodal retrieval-augmented approach achieves substantial gains under both observation modes (Figure 6), with improvements of up to 22%. As shown in Table 4, the gains are particularly pronounced on medium and hard tasks, effectively alleviating the performance collapse caused by longer action sequences and more complex requirements. Appendix F provides detailed execution cases along with side-by-side comparisons between retrieval and non-retrieval settings. We further compare our method with a purely text-based trajectory retrieval approach $(q \rightarrow \tau)$ (Kim et al., 2024) in Appendix D.5, and observe that it is less effective than multimodal retrieval. In contrast, our model-based retrieval method (Gu et al., 2024) more accurately aligns with the decision needs of $(q, s) \rightarrow s'$, thereby reducing the noise introduced by irrelevant exemplars. To verify the effectiveness of trajectory retrieval across different state-of-the-art agent frameworks, we further find that introducing trajectory knowledge into the SeeAct framework (Zheng et al., 2024) improves the overall success rate by 8-10%, with particularly large gains on medium and hard tasks (shown in Appendix D.6). This demonstrates that our method is highly scalable.

**Which Knowledge Format Helps More: Textual or Visual?**   Table 4 shows that visual exemplars slightly outperform textual ones across both observation modes, with a clearer margin on hard tasks , highlighting the importance of visual anchors (e.g., elements, button shapes, relative positions)

for cross-site transfer. For efficiency, textual guidance lowers steps and token usage by providing explicit rules and clearer goals, whereas visual guidance prompts more exploratory alignment, raising token use but improving robustness. Appendix F provides case studies and qualitative analyses, with Appendix F.5 offering an in-depth discussion of how retrieved knowledge helps address hard tasks.

## 6 CONCLUSION

In this work, we introduce the task of Multimodal Trajectory Retrieval and present UATD, a unified-format dataset of real-world, GUI-based agent trajectories. Building on this, we construct two standardized benchmarks, GAE-Bench and GAE-Bench-lite, to support evaluation. To address the challenges of long multimodal sequences, we propose GAE-Retriever, a VLM2Vec-based retrieval framework enhanced with token selection and GradCache for efficient contrastive learning. Experimental results show that GAE-Retriever achieves the best performance across all five environments. Compared to the strongest retrieval baseline (VLM2Vec-V2.2), it improves Recall@1 by up to 12.9 points. We introduce WebRAGent, a retrieval-augmented web agent, achieving 15-22% gains over non-retrieval baselines on the Online-Mind2Web benchmark.More importantly, as an ongoing effort, our recipe for constructing a multimodal trajectory retrieval system lays the groundwork for future investigations into retrieval-based context learning, reinforcement learning, and world modeling.

**Limitations and Future Work** Our research also has several limitations. First, although we position the learned trajectory representation as a foundation for broader agentic capabilities (e.g., world modeling, RL training), in this paper we primarily validate it through retrieval-augmented planning, and its effectiveness in other downstream tasks remains to be established. Second, our experiments are currently concentrated on web-agent data; while the methodology is in principle generalizable, we have not yet scaled training and evaluation to desktop, mobile, or embodied environments, which limits our ability to assess cross-platform generalization. Finally, our intention in positioning this paper is to highlight future exploration directions for the research community, encouraging further investigation into this valuable yet often overlooked area.

## REPRODUCIBILITY STATEMENT

We have made every effort to ensure that the results reported in this paper are reproducible. All code and datasets are publicly available in an anonymous repository[1] to facilitate replication and verification. We provide detailed descriptions of the experimental setup, including training procedures, model configurations, and hardware specifications. In addition, the appendix contains numerous task execution examples and side-by-side comparisons. We believe these measures will enable other researchers to reproduce our work and further advance the field.

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

## A  THE USE OF LLMS

We used a large language model (LLM) to assist with linguistic polishing of the manuscript, including refining expressions, enhancing textual coherence, improving readability, and performing sentence rewrites and grammar checking. In this work, the LLM did not participate in research ideation, method design, or experimental analysis; all scientific content was completed independently by the authors. The LLM's role was strictly limited to language-level improvements. The authors take full

responsibility for the entire content and have ensured that the use of the LLM adheres to academic ethics, precluding plagiarism and other inappropriate practices.

# B DATASET DETAILS

## B.1 DETAILS OF THE UNIFIED AGENT TRAJECTORY DATASET

This subsection introduces the methodology for constructing the Unified Agent Trajectory Dataset and presents its statistical properties. The real-world human trajectories and their action definitions in this dataset are derived from the following five GUI datasets: Mind2Web (Deng et al., 2023), AutoWebGLM (Lai et al., 2024), WebArena (Zhou et al., 2024; Koh et al., 2024a), WebLINX (Lu et al., 2024), and GUIAct (Chen et al., 2024). The raw data is manually cleaned by removing invalid trajectories and corrupted states. For sources without native visual observations (e.g., AutoWebGLM), we first complete the HTML using `gpt-4o-mini`, and then render the content via Playwright to produce screenshot-based observations. Additionally, textual descriptions are generated for each state, conditioned on the corresponding screenshot, to facilitate subsequent retrieval tasks. The prompts used for this annotation process and definition of action spaces are listed in Appendix E.1.

Consequently, we obtain the Unified Agent Trajectory Dataset, with statistics shown in Table 5. In this dataset, all screenshots are preserved at their original resolution, and target elements are annotated using bounding boxes.

Table 5: Statistics of the Unified Agent Trajectory Dataset (UATD). For each data source, we include the total number of tasks along with the average, minimum, maximum, and total number of states.

| Source | Task | State | | | |
|---|---|---|---|---|---|
| | | Min | Max | Avg | Total |
| Mind2Web | 1,468 | 2 | 31 | 6.65 | 9,621 |
| AutoWebGLM | 140 | 1 | 34 | 4.43 | 620 |
| WebArena | 201 | 1 | 26 | 6.07 | 1,221 |
| WebLINX | 485 | 1 | 68 | 14.19 | 6,882 |
| GUIAct | 5,453 | 1 | 85 | 11.82 | 64,449 |

## B.2 GAE-BENCH-LITE / (MINI) DETAILS

GAE-Bench-lite contains **563,900** positive pairs; Table 6 summarize its overall composition and the per-task, per-source distribution (Mind2Web, AutoWebGLM, WebArena, WebLINX, GUIAct). We further introduce GAE-Bench-lite (mini), which contains only the candidates relevant to the evaluation subset. Table 7 lists the candidate sets of GAE-Bench-lite.

Table 6: Distribution of GAE-Bench-lite by retrieval tasks and data sources. This compact variant includes 563,900 positive pairs.

| Retrieval Task | Mind2Web | AutoWebGLM | WebArena | WebLINX | GUIAct |
|---|---|---|---|---|---|
| $(q, \tau) \rightarrow \tau'$ | 13,630 | 718 | 1,580 | 4,376 | 67,968 |
| $(q, \tau) \rightarrow s$ | 14,912 | 822 | 1,796 | 7,746 | 89,770 |
| $q \rightarrow \tau$ | 7,566 | 792 | 1,026 | 1,242 | 17,040 |
| $(q, s) \rightarrow s'$ | 16,306 | 960 | 2,040 | 12,794 | 117,992 |
| $(q, s) \rightarrow \tau$ | 14,912 | 822 | 1,796 | 7,746 | 89,770 |
| $q \rightarrow s$ | 10,943 | 760 | 1,305 | 6,337 | 48,433 |
| **Total** | 78,269 | 4,874 | 9,543 | 40,241 | 430,973 |

## B.3 ONLINE MIND2WEB TASK LIST

This setting is more realistic and challenging but also much less stable. In preliminary runs over all 300 tasks, we frequently encountered network failures, SSL errors, region-based access restrictions, login verification, and various anti-crawling or Turing-test mechanisms; despite engineering mitigations, robustness remained limited, and running multiple trials over all tasks would incur prohibitive token costs. For both cost and reproducibility, we therefore construct an online evaluation set of 100 tasks via a two-stage selection procedure: first, we run each of the 300 tasks at least three times on six

Table 7: Overview of candidate sets in the GAE-Bench-lite

| | Mind2Web | WebLINX | WebArena | GUIAct | AutoWebGLM | Total |
|---|---|---|---|---|---|---|
| **GAE-Bench-lite** | | | | | | |
| **State** | 9,475 | 5,852 | 1,104 | 42,980 | 620 | 60,031 |
| **Trajectory** | 1,261 | 207 | 171 | 2,840 | 132 | 4,611 |
| **Interval** | 44,323 | 48,663 | 5,882 | 408,311 | 2,401 | 509,580 |
| **GAE-Bench-lite (mini)** | | | | | | |
| **State** | 2,842 | 2,127 | 588 | 10,530 | 349 | 16,436 |
| **Trajectory** | 239 | 67 | 58 | 484 | 48 | 896 |
| **Interval** | 15,237 | 26,191 | 3,093 | 46,515 | 1,720 | 92,756 |

different servers and retain 156 tasks that can be executed reliably; second, we perform stratified sampling over these 156 tasks to preserve website and task-category diversity and to match the easy/medium/hard difficulty distribution of the original dataset.

Our final evaluation set contains **100** tasks from **51** distinct websites. A detailed description of each task is provided below, and Figure 7 summarizes the distribution of websites in this set.

1. Add Elevate at Chicago, IL, to favorites and show a virtual tour.
2. Book 4 tickets in the upper for any Kevin Hart show in New York in the next three months and view ticket prices with estimated fees.
3. Browse iPhone X for sale that is in good condition, has a max price of 400, and searches in titles only.
4. Browse recipes for gluten-free chocolate chip cookies that can be made without nuts.
5. Check the hourly forecast for Boston.
6. Check the specifications of the best-selling HP FHD laptop with 16 GB RAM and core i7 running on Windows 11.
7. Compare the breeds Afghan Hound, Akita and Azawakh.
8. Display the figure comparing unemployment trends among women in Illinois and Michigan.
9. Estimate the federal income tax I would owe on $158,500 of taxable income in ZIP code 97007, filing as single.
10. Find a condo for rent in Houston, TX, with a monthly rent of no more than 30% of an income of $8000. The condo should have a minimum area of 600 square feet, and the move-in date is the 1st of next month.
11. Find a day-use park that offers horseback riding near Nashville.
12. Find a dog groomer for nail trimming within 100 miles of zip code 10005 and check the detailed service prices of the first one.
13. Find a list of houses for sale in zip code 85747 with a private pool.
14. Find a personal trainer service at 10040 for a 25-year-old client aiming to build muscle.
15. Find a recipe that includes eggplant and mushrooms.
16. Find a Single-Family House for Rent in Houston, TX with 1 bed.
17. Find a walkthrough for the game "The Legend of Zelda: Breath of the Wild" on ign.
18. Find an editor's choice review with a score of 10 in the boardgame category on ign.
19. Find and open the earliest press release.
20. Find discussions of the community and open one with the most replies on Flightaware.
21. find electricians near 10203.
22. Find Farms land in Wilkes County, NC with the lowest price.
23. Find healthy savory vegan snack recipes which can be cooked within 5 minutes and contain a high level of protein.
24. Find Linux platform software developers in 10080 who master the Python language and Java language with web interface project type.
25. Find obedience trials in state of New York during the month of May.
26. Find out the cold and flu forecast and today's air quality in Champaign, IL.
27. Find the closing stock price for Tesla on March 17, 2023.
28. Find the lowest-priced Student housing near Liverpool International College which has been priced between 100 to 300 pounds and has a private bathroom.
29. Find the most cited publication at the 2022 CVPR main conference.
30. Find the most frequent word that rhymes with "thought" and has three syllables.
31. Find the race time for who wins the first place in the last race of the 2023 Formula 1 (F1).
32. Find UA or AA flights from London to New York that arrive between 8:00 PM and 11:00 PM on FlightAware.
33. Get the frozen vegan cheese pizza between 5 to 10 USD on Target.
34. Get the report from the final environmental impact statement for the Jamaica Bus Depot expansion on new.mta.info.
35. Identify a pill with a pink color and oval shape with 894 5 number on it.
36. Look for the largest hunting land for auction in Kansas high plain region with mineral rights posted in the last seven days.

37. Open the page with an overview of the submission of releases on Discogs.
38. Search for a beginner's course in computer science that includes advertisement skills.
39. Search for a job in Miami, Florida, in Human Resources on target.
40. Search for NordicTrack with the lowest price.
41. Search for rentals in Corning, CA with a maximum price of $1500.
42. See the monthly forecast for Atlanta, GA.
43. Show crazy credits for the movie "Prometheus" on IMDb.
44. Show daily weather for New York City.
45. Show houses for sale in Maryland with a maximum price of $60,000.
46. Show me the monthly weather forecast for Florida City.
47. Show me the page with average wait times for U.S. citizens arriving at Raleigh-Durham International Airport on 2025-03-12.
48. Show me the wind flow map for Belo Horizonte.
49. Submit a request for vehicle registration renewal with title number X123456 and last 4 digits of VIN is 1234.
50. View the cheapest apartment available for students at the University of Leeds with bills that include WIFI and cleaning services.
51. What are the Nearby Attractions from the most popular attraction in Hong Kong?
52. What is the ownership cost of the first car in the list "top buys 2025"?
53. Check the current wind speed in Calgary, Alberta.
54. Open the most helpful 5-star reviews of Alpine Ridge.
55. Browse the first top news of Microsoft stock on Google Finance.
56. Find the list of neighborhood maps for Brooklyn on new.mta.info.
57. Find the park that offers the cheapest paddling permits.
58. Browse Marriott Bonvoy credit cards on Marriott.
59. Find the recommended dosage for Vivitrol.
60. Show me the comparison of the first two personal credit cards that do not charge foreign transaction fees.
61. Search for papers related to reinforcement learning under the topics of computer science and mathematics on arxiv, with recent submission dates between September 2024 and January 2025.
62. Open the reviews of a recipe with beef sirloin.
63. Can you show me the page with the filing fee for a self-petitioned I-140 application?
64. Compare available plans for the AeroAPI on Flightaware.
65. Find the interactions between Eulexin and hepatic dysfunction.
66. Find the stock price for WWE over the last month.
67. Find the next available dates for Alley Creek Camp.
68. Find the latest climate news.
69. Show me historical data for EUR/USD.
70. Browse the list of Civil Division forms.
71. Find the side effects of taking Montelukast.
72. View the list of the Most Popular TV on rotten tomatoes.
73. Find the tech specs of the MacBook Pro 16-inch introduced in November 2023.
74. Show Teen Driver Safety program information.
75. Add a $100 Best Buy gift card for a birthday to my cart.
76. Add a $50 Uber gift card to the cart.
77. Check the interaction between Novolin N and Novolin R.
78. Find support services jobs in Bentonville, in the state of Arkansas.
79. Find out what to do when I lose an item on a bus on us.megabus.
80. Check the status of bus S92 for any disruptions on new.mta.info.
81. Look up the current temperature for zip code 10019.
82. Show me the rules and cancellation for Alley Spring.
83. Open the XRP yearly chart.
84. Find the weather for Vancouver, British Columbia for the next seven days.
85. Find the next available date for Albion Basin.
86. Browse the final skin in the list for the champion Ahri.
87. Find a DMV center in Richmond.
88. Check drug interaction for melatonin and Folate Forte.
89. Identify the ongoing competition that offers the highest prize and find the code that received the most votes in that competition.
90. Find the 5-day price chart for Bitcoin.

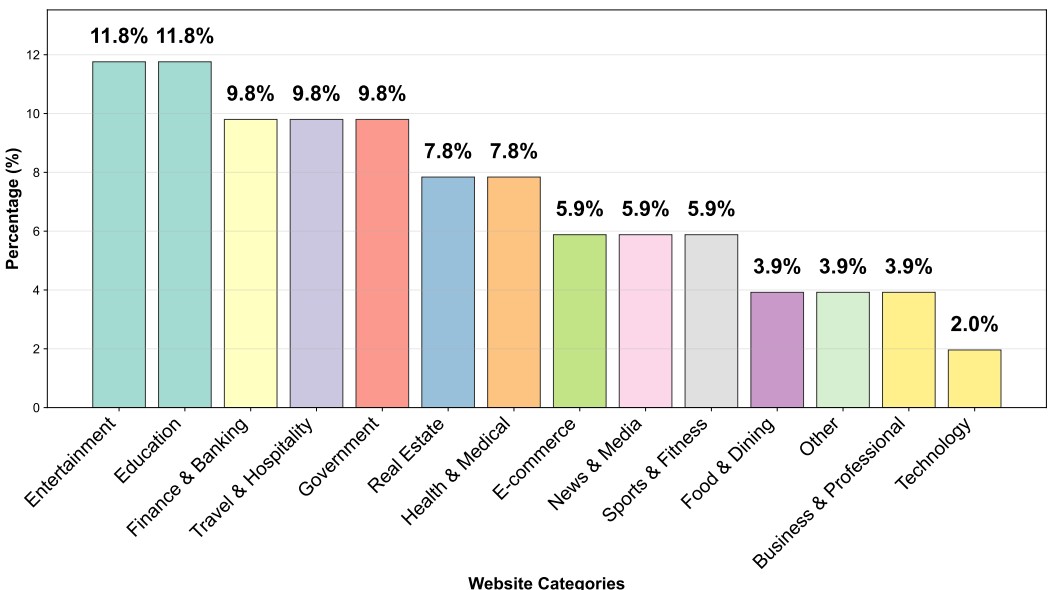

Figure 7: Distribution of website categories across all tasks.

91. Show theatre events for Las Vegas and select one.

92. Browse Humira dosage information.

93. Find the app for iOS.

94. Show me recipes for pancakes with wheat and without beetroot.

95. Check the current standings for MLS on Fox Sports.

96. Find technical specs for the latest Macbook Air on Apple.

97. Find the Drug Interaction Report for Viagra and alcohol.

98. Browse the upcoming SuperBike events taking place in Italy.

99. Find possible causes for the symptoms of chest pain which is sharp which is accompanied by anxiety.

100. Take a weight management quiz to find a motivating article for a non-exercising, mostly eating out and can't control portions and cravings, and who has a strong support system, enjoys traveling, loves family time and cooking.

## C  IMPLEMENTATION DETAILS

### C.1  INPUT SEQUENCE FORMATION

As described in Subsections 3.1 and 4.1, we represent a state $s_i$ using a screenshot, and each action $a_i$ is expressed as a JSON object with three keys: *operation*, *value*, and *target*. To support action modeling across diverse platforms, we standardize this JSON schema across environments; the *target* is specified by relative bounding-box coordinates in $[0, 1]$ (e.g., $(x_{\min}, y_{\min}, x_{\max}, y_{\max})$), and the action space definition is prepended as context.

A trajectory subsequence $\tau_{i:j}$ or a full trajectory $\tau_{1:n}$ (or simply $\tau$) is composed of state-action pairs. To standardize the representation across different environments, we prepend the action space definition to $\tau_{i:j}$ and $\tau_{1:n}$, resulting in $u_{i:j}$ and $u_{1:n}$. For the target side of retrieval, $\mathbf{v}$ can be either a single state $s_i$ or a trajectory sequence with action definitions, denoted as $u_{i:j}$. For the query side $\mathbf{k}$, we prepend the retrieval query $\tilde{q}$, including a task-specific instruction and trajectory description to $s_i$ or $u_{i:j}$. Table 8 provides concrete examples, where [image] denotes image tokens.

### C.2  IMPLEMENTATION DETAILS OF THE RETRIEVER

We adopt the OpenAI model gpt-4o-mini-2024-07-18 for data annotation. The retrieval model is based on Qwen2-VL-Instruct and trained with LoRA of rank 8 on 16 NVIDIA H800 GPUs. We employ GradCache during training with a sub-batch size of 1 and a total accumulated batch size of 2,048. The schedule consists of 256 steps on GAE-Bench-lite, summing to 1,044 GPU hours. Unless otherwise noted, we use a learning rate of $5 \times 10^{-5}$, a warm-up ratio of 5%, and an interleaved

Table 8: Examples of model input representations for a state ($s_i$), an action ($a_i$), a trajectory sequence or subsequence with action definitions ($u_{i:j}$), and an augmented retrieval query ($\tilde{q}$). The retrieval key **k** consists of $\tilde{q}$ combined with either $s_i$ or $u_{i:j}$, while the retrieval value **v** is composed of either $s_i$ or $u_{i:j}$.

| Variable | Representation Instance |
|---|---|
| $s_i$ | Observation: [image] |
| $a_i$ | Action $i$: {"operation": "type", "value": "Elon Musk", "target": {"x": 0.5704, "y": 0.2142, "width": 0.3678, "height": 0.0663}} |
| $u_{i,j}$ | Action Space:
1. click: Simulates a mouse click on the target bounding box.
2. type: Types the value (str) into the target bounding box.
Positions are represented in relative coordinates within the range [0,1] on the observation screenshot.
Observation 1: [image]
Action 1: {"operation": "click", "value": null, "target": {"x": 0.0021, "y": 0.1424, "width": 0.0243, "height": 0.0519}}
Observation 2: [image]
Action 2: {"operation": "type", "value": "", "target": {"x": 0.8343, "y": 0.5659, "width": 0.1034, "height": 0.2496}} |
| $\tilde{q}$ | Apply the request "Creating a template on Trello in a new tab." to the previous web navigation steps to derive the next trajectory. |

batch size of 0.2; training runs in Bfloat16 precision with a loss temperature of 0.02. The maximum token length is 65,536 for both training and evaluation. We enable UI-graph-based token selection only during training, as it adds no learnable parameters and preserves the positional structure of the full token sequence; following Lin et al. (2024a), we apply a 0.5 mask ratio across all transformer layers. We compute final sequence representations using EOS pooling followed by normalization. For image preprocessing, we rely on *qwen-vl-utils*, which resizes inputs to tensors with spatial bins of $28 \times 28$ and channel depth ranging from 4 to 1280, depending on content. Evaluation is conducted on GAE-Bench-lite with the mini candidate sets using a batch size of 6 on 8 NVIDIA H800 GPUs, requiring 22.5 GPU hours.

## C.3 IMPLEMENTATION DETAILS OF THE WEBRAGENT FRAMEWORK

**Observation** The framework supports two independent observation modes: DOM (textualized structure) and Vision (page screenshots). In DOM mode, we apply strict filtering to remove redundant web-structure noise, selecting only interactive elements and exposing their tags and content. In Vision mode, full-page screenshots capture cues missing from the DOM view.

**Planning and Action** We use different planning models for the two observation modes. In DOM mode, we adopt GPT-4.1, which excels at textual/structural understanding and supports controllable reasoning traces. In Vision mode, we use OpenAI's computer-use-preview, which performs screen understanding and pointer-style action decisions. The action space covers: click, double_click, type, scroll, keypress, drag, search, wait, get_final_answer. The detailed prompt for the planning model is shown in snippet 4.

**Reward** Our framework's reward module evaluates tasks based on the historical trajectory and the current web state. It uses the agent's thoughts, actions, and reflections to dynamically assess progress and produce a quantitative score. The evaluator considers the full trace, accessibility-tree features, and (when available) screenshot cues to rate the trajectory's validity and effectiveness on a discrete scale 1, 3, 7, 9, 10: **1** indicates a loop requiring course correction, while **10** denotes a perfectly completed task. Results are emitted as structured JSON containing the status, score, rationale, and description, enabling clear feedback on current progress, next steps, and improvement directions to efficiently guide subsequent actions.

**Retrieve from the Step-Level Knowledge Base** To reuse experience more accurately and efficiently, we employ the GAE-Retriever to encode the current-step state (task instruction + screenshot) into a retrieval query for similarity matching, and then apply LLM re-ranking to improve match precision, directly locating the most similar textual or visual guidance in the step-level knowledge base. The knowledge base is step-aligned: each record represents an "observation → action" example. It offers two guidance types. Visual guidance includes the step image plus a brief structured summary (task description, action space, and action metadata). Textual guidance is generated with GPT-4o on the Unified Agent Trajectory Dataset (UATD), providing detailed natural-language descriptions of

the observation and action. Detailed data examples are shown in Snippets 7–8. The knowledge base can be incrementally updated to rapidly adapt to UI changes.

# D EXPERIMENTAL DETAILS

## D.1 SUBTASK PERFORMANCE

In this section, we present the evaluation of all baseline methods described in Subsection 5.1, along with our proposed GAE-Retriever, on 12 retrieval subtasks spanning all data sources. Results are given in terms of Recall@1/5/10. Table 9 provides the outcomes under the in-domain setting, while Table 10 covers the out-of-domain evaluation.

Table 9: Recall@1/5/10 for all methods on each subtask in the in-domain setting.

| Source | $(q, \tau_{1:i}) \to \tau_{i+1:n}$ | $(q, \tau_{1:n}) \to \tau_{1:i}$ | $(q, \tau_{1:i}) \to s_{i+1}$ | $(q, \tau_{1:n}) \to s_i$ | $q \to \tau_=$ | $q \to \tau_\sim$ | $(q, s_i) \to s_{i+1}$ | $(q, s_{i+1}) \to s_i$ | $(q, s_i) \to \tau_{1:n}$ | $(q, s_{i+1}) \to \tau_{1:i}$ | $q \to s_i$ | $q \to s_n$ |
|---|---|---|---|---|---|---|---|---|---|---|---|---|
| **Qwen2-VL-2B** | | | | | | | | | | | | |
| Mind2Web | 0.0/0.0/1.5 | 0.0/1.0/2.0 | 0.1/0.3/0.6 | 0.0/0.1/0.7 | 0.0/0.9/1.8 | 0.0/1.5/2.0 | 3.2/54.9/70.8 | 2.6/55.0/71.2 | 0.0/2.0/2.5 | 0.0/0.0/0.0 | 0.0/0.3/0.6 | 0.0/0.7/0.7 |
| AutoWebGLM | 0.0/0.0/4.0 | 0.0/0.0/0.0 | 2.0/2.0/4.0 | 0.0/0.0/0.0 | 7.1/17.9/35.7 | 8.9/16.1/39.3 | 0.0/18.0/32.0 | 2.0/6.0/8.0 | 0.0/0.0/0.0 | 0.0/0.0/0.0 | 0.0/0.0/0.0 | 3.3/3.3/3.3 |
| WebArena | 0.0/0.0/0.0 | 0.0/0.0/0.0 | 0.0/2.4/4.8 | 0.0/3.6/6.0 | 0.0/10.8/21.6 | 0.0/9.3/20.0 | 3.1/39.6/51.0 | 10.4/47.9/54.2 | 0.0/0.0/2.4 | 0.0/0.0/0.0 | 0.0/0.0/0.0 | 0.0/0.0/0.0 |
| WebLINX | 0.0/0.5/0.5 | 0.0/0.5/1.0 | 0.3/0.8/1.3 | 0.0/0.0/0.0 | 0.0/4.3/10.9 | 1.1/5.4/15.1 | 1.9/35.8/48.1 | 17.5/48.2/58.3 | 0.0/0.0/0.5 | 0.0/1.0/1.0 | 0.0/2.0/2.5 | 0.0/0.0/0.0 |
| GUIAct | 0.0/0.0/0.0 | 0.0/0.0/0.0 | 0.1/0.1/0.3 | 0.0/0.2/0.4 | 1.0/4.0/4.5 | 0.0/0.0/0.5 | 0.6/18.9/23.8 | 23.4/35.9/39.1 | 0.0/0.0/0.0 | 0.0/0.0/0.0 | 0.0/0.1/0.3 | 0.0/0.2/0.2 |
| **Qwen2.5-VL-3B** | | | | | | | | | | | | |
| Mind2Web | 0.0/0.0/0.5 | 0.0/0.0/0.0 | 0.0/0.7/0.7 | 0.0/0.1/0.3 | 0.9/2.7/12.4 | 1.5/4.5/10.0 | 3.7/30.7/39.2 | 4.1/30.3/37.7 | 0.0/0.0/0.0 | 0.0/0.0/0.0 | 0.2/0.5/0.6 | 0.0/0.7/0.7 |
| AutoWebGLM | 0.0/0.0/0.0 | 0.0/0.0/0.0 | 0.0/2.0/4.0 | 0.0/2.0/4.0 | 3.6/14.3/21.4 | 1.8/14.3/19.6 | 0.0/4.0/8.0 | 0.0/0.0/2.0 | 0.0/0.0/0.0 | 2.0/2.0/2.0 | 0.0/1.8/3.6 | 0.0/3.3/6.7 |
| WebArena | 0.0/0.0/0.0 | 0.0/0.0/0.0 | 0.0/0.0/2.4 | 0.0/0.0/1.2 | 2.7/8.1/24.3 | 2.7/8.0/20.0 | 2.1/34.4/42.7 | 4.2/37.5/44.8 | 0.0/1.2/1.2 | 0.0/0.0/0.0 | 0.0/1.0/1.0 | 0.0/2.2/4.4 |
| WebLINX | 0.0/1.0/1.0 | 0.5/0.5/1.0 | 0.0/3.0/3.0 | 0.0/0.0/3.0 | 2.2/8.7/23.9 | 2.2/11.8/21.5 | 5.1/34.7/41.7 | 16.6/44.7/53.2 | 0.0/0.0/0.0 | 0.0/0.0/0.5 | 0.0/0.2/0.3 | 0.0/2.0/2.0 |
| GUIAct | 0.0/0.0/0.0 | 0.0/0.0/0.0 | 0.0/1.0/1.1 | 0.0/1.0/2.2 | 1.0/1.0/2.5 | 0.0/2.5/5.0 | 1.9/18.7/23.4 | 20.6/33.0/36.9 | 0.0/0.0/0.0 | 0.0/0.0/0.0 | 0.0/0.0/0.0 | 0.0/0.0/0.0 |
| **LamRA-Ret** | | | | | | | | | | | | |
| Mind2Web | 0.0/0.0/0.0 | 0.0/0.0/0.0 | 0.0/0.3/0.6 | 0.0/0.7/1.2 | 6.2/13.3/19.5 | 0.5/9.0/16.0 | 2.6/54.5/69.3 | 3.3/54.6/70.1 | 0.0/0.0/0.5 | 0.0/0.0/0.0 | 1.0/5.0/7.7 | 0.7/2.2/4.5 |
| AutoWebGLM | 0.0/0.0/0.0 | 0.0/0.0/0.0 | 0.0/10.0/16.0 | 4.0/6.0/16.0 | 25.0/39.3/53.6 | 16.1/35.7/50.0 | 0.0/18.0/28.0 | 0.0/24.0/36.0 | 2.0/6.0/16.0 | 0.0/0.0/0.0 | 0.0/7.1/17.9 | 0.0/6.7/16.7 |
| WebArena | 0.0/0.0/1.4 | 0.0/0.0/0.0 | 0.0/2.4/4.8 | 0.0/2.4/4.8 | 13.5/32.4/48.6 | 6.7/22.7/33.3 | 0.0/34.4/49.0 | 9.4/40.6/51.0 | 0.0/0.0/0.0 | 0.0/1.2/1.2 | 0.0/0.0/1.9 | 0.0/0.0/0.0 |
| WebLINX | 0.0/0.0/1.0 | 0.0/0.0/0.5 | 0.3/0.5/1.1 | 0.0/0.0/0.3 | 4.3/21.7/32.6 | 2.2/5.4/23.7 | 0.5/38.9/48.9 | 17.8/50.3/58.3 | 0.0/0.0/0.0 | 0.0/0.0/0.0 | 1.7/7.0/12.9 | 2.0/6.0/14.0 |
| GUIAct | 0.0/0.0/0.0 | 0.0/0.0/0.0 | 0.1/0.1/0.3 | 0.1/0.4/0.7 | 21.0/46.5/56.5 | 9.0/24.0/32.0 | 0.7/18.7/23.6 | 23.1/35.1/38.2 | 0.5/0.5/0.5 | 0.0/0.0/0.5 | 0.3/0.8/1.5 | 0.2/0.6/0.8 |
| **ColQwen2-v1.0** | | | | | | | | | | | | |
| Mind2Web | 0.0/0.0/0.0 | 0.0/0.0/0.0 | 1.6/8.9/14.2 | 1.9/7.7/13.5 | 9.7/23.0/36.3 | 6.0/17.0/28.0 | 2.4/56.2/73.6 | 3.2/56.2/74.1 | 0.0/1.5/3.0 | 1.0/5.0/8.5 | 3.2/8.0/13.8 | 1.5/9.0/11.2 |
| AutoWebGLM | 0.0/0.0/0.0 | 0.0/0.0/2.0 | 6.0/22.0/28.0 | 0.0/8.0/22.0 | 21.4/35.7/53.6 | 21.4/30.4/51.8 | 0.0/24.0/42.0 | 0.0/30.0/50.0 | 0.0/0.0/0.0 | 0.0/6.0/12.0 | 5.4/8.9/16.1 | 16.7/26.7/26.7 |
| WebArena | 0.0/0.0/0.0 | 0.0/4.1/4.4 | 2.4/5.6/10.7 | 1.2/11.9/16.7 | 8.1/27.0/54.1 | 2.7/25.3/44.0 | 3.1/42.7/52.1 | 11.5/43.8/52.1 | 0.0/0.0/0.0 | 1.2/4.8/7.1 | 1.9/3.9/9.7 | 0.0/8.9/13.3 |
| WebLINX | 0.0/0.0/1.0 | 0.0/0.0/1.5 | 1.6/6.9/11.5 | 1.3/10.1/15.7 | 21.7/58.7/67.4 | 9.7/34.4/45.2 | 1.1/39.3/51.6 | 17.2/48.6/59.9 | 0.5/0.5/2.5 | 1.5/3.0/3.5 | 0.3/4.4/7.5 | 4.0/10.0/12.0 |
| GUIAct | 0.0/0.0/0.5 | 0.0/0.0/0.0 | 0.4/1.6/2.6 | 1.1/4.7/7.6 | 57.0/81.0/89.0 | 20.5/48.5/63.0 | 0.7/24.1/30.1 | 23.3/38.9/42.9 | 0.5/4.0/8.5 | 1.0/1.5/2.0 | 0.8/1.9/3.2 | 0.4/1.7/2.3 |
| **GME-Qwen2VL-2B** | | | | | | | | | | | | |
| Mind2Web | 0.0/0.0/1.0 | 0.0/0.0/0.0 | 2.0/8.7/16.7 | 1.5/8.4/15.8 | 13.3/39.8/55.8 | 10.5/24.0/36.5 | 2.0/57.5/74.7 | 2.8/55.8/74.1 | 0.5/2.0/6.5 | 0.5/4.0/8.5 | 2.5/7.2/12.8 | 1.5/6.7/9.0 |
| AutoWebGLM | 0.0/0.0/0.0 | 0.0/0.0/0.0 | 6.0/16.0/30.0 | 6.0/18.0/28.0 | 71.4/82.1/92.9 | 30.4/60.7/75.0 | 0.0/40.0/60.0 | 0.0/46.0/60.0 | 0.0/0.0/2.0 | 0.0/4.0/6.0 | 5.4/16.1/26.8 | 13.3/33.3/50.0 |
| WebArena | 0.0/0.0/0.0 | 0.0/0.0/0.0 | 1.2/9.5/16.7 | 3.6/16.7/22.6 | 16.2/45.9/59.5 | 10.7/36.0/53.3 | 2.1/40.6/52.1 | 11.5/47.9/53.1 | 1.2/2.4/7.1 | 1.2/2.4/4.8 | 4.9/16.5/18.4 | 0.0/0.0/17.8 |
| WebLINX | 0.0/0.0/1.5 | 0.0/0.5/1.0 | 2.1/14.1/24.3 | 4.3/13.6/25.6 | 34.8/63.0/73.9 | 11.8/40.9/57.0 | 1.1/40.4/51.9 | 17.0/50.3/62.3 | 0.0/0.0/6.5 | 0.0/4.0/7.0 | 0.5/4.2/7.2 | 2.0/6.0/6.0 |
| GUIAct | 0.0/0.0/0.0 | 0.0/0.0/0.0 | 0.9/4.4/6.9 | 1.7/8.0/11.6 | 50.0/80.5/93.0 | 10.0/34.5/48.0 | 1.1/24.4/30.7 | 22.1/39.4/43.7 | 0.0/4.0/5.0 | 1.5/4.0/5.5 | 0.9/2.8/4.2 | 0.4/1.0/2.7 |
| **UniSE-MLLM** | | | | | | | | | | | | |
| Mind2Web | 0.0/0.0/0.5 | 0.0/0.5/4.0 | 0.9/3.1/6.1 | 0.7/3.9/7.6 | 5.3/9.7/13.3 | 1.5/6.0/11.5 | 5.8/57.1/73.4 | 4.5/55.3/74.8 | 2.0/4.0/5.5 | 0.0/0.5/1.0 | 1.7/6.0/11.5 | 0.7/7.5/9.0 |
| AutoWebGLM | 0.0/0.0/0.0 | 0.0/0.0/2.0 | 4.0/16.0/20.0 | 0.0/4.0/12.0 | 25.0/32.1/39.3 | 8.9/19.6/37.5 | 0.0/24.0/34.0 | 0.0/30.0/38.0 | 0.0/0.0/0.0 | 0.0/0.0/0.0 | 1.9/5.8/9.7 | 0.0/6.7/8.9 |
| WebArena | 0.0/0.0/0.0 | 0.0/1.4/2.7 | 1.2/4.8/7.1 | 0.0/13.1/15.5 | 8.1/13.5/37.8 | 5.3/14.7/32.0 | 5.2/42.7/53.1 | 9.4/44.8/54.2 | 1.2/4.8/7.1 | 0.0/0.0/0.0 | 1.9/5.8/9.7 | 0.0/2.0/2.0 |
| WebLINX | 0.0/1.0/2.0 | 0.0/0.0/1.5 | 0.5/3.7/5.1 | 0.0/0.8/1.6 | 2.2/8.7/21.7 | 1.1/7.5/22.6 | 3.5/36.9/48.1 | 17.5/48.4/59.4 | 0.0/1.5/2.0 | 0.0/0.5/0.5 | 0.3/0.9/1.4 | 0.0/2.0/2.0 |
| GUIAct | 0.0/0.0/0.5 | 0.0/0.0/0.0 | 0.4/0.4/0.9 | 0.4/1.3/1.6 | 3.5/10.5/14.0 | 1.0/2.5/3.5 | 0.7/20.2/26.6 | 23.2/37.4/41.0 | 0.0/0.0/0.0 | 0.0/0.0/0.0 | 0.1/0.3/0.6 | 0.0/2.0/0.4 |
| **VLM2Vec-Qwen2VL-2B** | | | | | | | | | | | | |
| Mind2Web | 0.0/0.0/0.0 | 0.0/0.5/0.5 | 5.5/27.0/48.0 | 4.5/31.8/57.8 | 24.8/60.2/76.1 | 10.5/33.0/49.0 | 2.4/62.4/79.2 | 2.0/59.6/80.2 | 0.5/5.5/7.5 | 0.5/4.5/5.5 | 9.8/30.2/46.7 | 10.4/33.6/45.5 |
| AutoWebGLM | 0.0/0.0/0.0 | 0.0/0.0/2.0 | 8.0/26.0/56.0 | 8.0/38.0/66.0 | 67.9/82.1/89.3 | 32.1/58.9/71.4 | 0.0/44.0/64.0 | 0.0/42.0/56.0 | 2.0/8.0/14.0 | 0.0/2.0/8.0 | 16.1/44.6/57.1 | 13.3/42.8/42.9 |
| WebArena | 0.0/0.0/0.0 | 0.0/0.0/0.0 | 2.4/13.1/21.4 | 4.8/20.2/27.4 | 24.3/59.5/78.4 | 14.7/40.0/57.3 | 4.2/43.8/53.1 | 8.3/46.9/59.4 | 0.0/4.8/9.5 | 0.0/2.4/3.6 | 5.8/23.3/47.6 | 13.3/24.4/28.9 |
| WebLINX | 0.0/0.0/0.5 | 0.0/0.0/0.5 | 5.6/23.7/39.7 | 6.4/23.2/41.9 | 37.0/56.5/78.3 | 14.0/39.8/51.6 | 1.3/43.0/55.4 | 16.9/51.1/65.4 | 1.5/3.5/8.5 | 0.5/2.5/6.5 | 4.9/20.1/30.4 | 2.0/4.0/6.0 |
| GUIAct | 0.0/0.0/0.0 | 0.0/0.0/0.0 | 0.9/5.2/8.0 | 3.9/14.0/19.3 | 67.0/91.0/95.0 | 15.0/35.5/55.0 | 0.3/28.0/35.8 | 24.0/42.3/47.2 | 7.5/16.0/23.5 | 0.0/1.5/2.0 | 3.6/11.1/16.0 | 0.4/1.2/2.1 |
| **VLM2Vec-V2.2** | | | | | | | | | | | | |
| Mind2Web | 0.0/0.0/0.0 | 0.0/0.0/0.0 | 7.6/46.8/70.9 | 4.1/40.8/74.7 | 66.4/90.3/95.6 | 28.0/60.0/76.5 | 5.1/59.4/78.1 | 3.8/58.3/79.2 | 1.5/6.0/10.0 | 1.5/4.5/8.0 | 11.8/37.4/55.6 | 17.9/43.3/61.2 |
| AutoWebGLM | 0.0/0.0/0.0 | 0.0/0.0/10.0 | 6.0/44.0/80.0 | 4.0/32.0/72.0 | 85.7/100.0/100.0 | 48.2/73.2/83.9 | 0.0/58.0/76.0 | 0.0/58.0/70.0 | 0.0/4.0/6.0 | 0.0/4.0/8.0 | 16.1/62.5/85.7 | 8.9/35.6/51.1 |
| WebArena | 0.0/0.0/0.0 | 0.0/0.0/0.0 | 3.6/21.4/26.2 | 1.2/27.4/44.0 | 45.9/78.4/86.5 | 32.0/61.3/74.7 | 2.1/44.8/52.1 | 10.4/51.0/59.4 | 0.0/4.8/9.5 | 0.0/1.2/4.8 | 7.8/31.1/44.7 | 8.9/35.6/51.1 |
| WebLINX | 0.0/0.0/0.5 | 0.0/0.0/0.5 | 8.5/44.8/64.0 | 6.4/37.3/62.4 | 78.3/91.3/93.5 | 37.6/68.8/76.3 | 3.0/45.4/59.2 | 18.8/53.5/66.7 | 0.0/4.0/10.5 | 0.0/3.0/6.5 | 14.3/43.5/61.1 | 0.0/4.0/14.0 |
| GUIAct | 0.0/0.0/0.0 | 0.0/0.0/0.5 | 5.0/21.9/31.6 | 9.4/34.0/45.1 | 96.0/100.0/100.0 | 55.0/85.0/95.5 | 1.6/33.2/41.8 | 23.5/46.2/52.2 | 4.5/11.5/15.0 | 0.0/2.5/4.5 | 19.9/41.5/50.0 | 1.6/6.0/10.3 |
| **UGround-V1-2B** | | | | | | | | | | | | |
| Mind2Web | 0.0/0.0/0.0 | 0.0/1.0/3.0 | 0.0/0.1/0.1 | 0.0/0.0/0.1 | 0.0/0.0/0.0 | 0.0/0.0/0.0 | 3.4/51.3/67.5 | 3.6/50.5/67.4 | 0.0/0.0/0.0 | 0.5/0.5/0.5 | 0.0/0.1/0.2 | 0.0/0.0/0.7 |
| AutoWebGLM | 0.0/0.0/0.0 | 0.0/0.0/0.0 | 0.0/0.0/4.0 | 0.0/0.0/0.0 | 3.6/14.3/25.0 | 3.6/14.3/19.6 | 0.0/14.0/22.0 | 0.0/20.0/22.0 | 0.0/0.0/0.0 | 0.0/2.0/4.0 | 0.0/0.1/3.3 | 0.0/3.3/3.3 |
| WebArena | 0.0/0.0/0.0 | 0.0/0.0/0.0 | 0.0/0.0/3.6 | 0.0/3.6/4.8 | 0.0/0.0/2.7 | 0.0/0.0/2.7 | 1.0/40.6/52.1 | 9.4/44.8/52.1 | 0.0/0.0/0.0 | 0.0/0.0/1.2 | 0.0/1.0/3.9 | 0.0/0.0/0.0 |
| WebLINX | 0.0/0.5/0.5 | 0.0/0.5/0.5 | 0.3/1.1/1.6 | 0.0/0.0/0.0 | 2.2/10.9/13.0 | 2.2/9.7/12.9 | 1.3/35.5/48.4 | 17.4/49.0/57.2 | 0.0/0.0/0.0 | 0.0/0.0/0.0 | 0.0/2.1/0.0 | 0.0/0.0/0.0 |
| GUIAct | 0.0/0.0/0.0 | 0.0/0.0/0.0 | 0.0/0.1/0.1 | 0.0/0.0/0.1 | 0.0/0.5/1.5 | 0.0/0.5/3.5 | 0.5/17.9/24.6 | 23.4/34.9/39.2 | 0.0/0.0/0.0 | 0.0/0.0/0.0 | 0.0/0.1/0.1 | 0.2/0.2/0.4 |
| **ShowUI-2B** | | | | | | | | | | | | |
| Mind2Web | 0.0/0.0/0.0 | 0.0/0.0/0.0 | 0.0/4.0/0.6 | 0.0/0.6/1.2 | 1.8/2.7/5.3 | 1.0/3.0/5.5 | 4.2/52.9/69.9 | 4.9/51.3/69.3 | 0.0/0.5/0.5 | 0.0/0.0/0.0 | 0.1/0.1/0.5 | 0.0/0.0/0.0 |
| AutoWebGLM | 0.0/0.0/0.0 | 0.0/0.0/0.0 | 2.0/4.0/4.0 | 0.0/0.0/2.0 | 0.0/17.9/17.9 | 0.0/12.5/23.2 | 4.2/40.6/51.0 | 13.5/43.8/51.0 | 0.0/0.0/0.0 | 0.0/0.0/0.0 | 0.0/1.9/5.8 | 0.0/0.0/3.3 |
| WebArena | 0.0/0.0/0.0 | 0.0/0.0/0.0 | 0.0/1.2/3.6 | 1.2/2.4/6.0 | 5.4/10.8/27.0 | 2.7/13.3/22.7 | 4.2/40.6/51.0 | 13.5/43.8/51.0 | 0.0/0.0/0.0 | 0.0/0.0/0.0 | 0.0/1.9/5.8 | 0.0/0.0/0.0 |
| WebLINX | 0.0/0.0/0.5 | 0.0/0.0/0.5 | 0.0/3.0/5.0 | 0.3/0.5/1.3 | 2.2/8.7/10.9 | 2.2/8.6/11.8 | 3.5/36.9/46.8 | 17.5/46.0/56.2 | 0.0/0.0/0.0 | 0.0/0.0/0.0 | 0.2/0.9/0.9 | 0.0/0.0/0.0 |
| GUIAct | 0.0/0.0/0.0 | 0.0/0.0/0.0 | 0.0/0.1/0.1 | 0.0/0.0/0.1 | 0.0/0.5/0.5 | 0.0/0.0/0.0 | 0.9/18.4/23.4 | 22.9/35.1/38.4 | 0.0/0.0/0.0 | 0.0/0.0/0.0 | 0.0/0.1/0.1 | 0.0/0.0/0.0 |
| **UI-TARS-2B-SFT** | | | | | | | | | | | | |
| Mind2Web | 0.0/0.0/0.5 | 0.0/0.0/0.5 | 0.0/4.0/0.6 | 0.0/0.4/0.6 | 0.0/3.5/6.2 | 2.5/3.5/5.5 | 1.5/51.6/66.6 | 3.7/50.9/66.5 | 0.0/0.0/0.0 | 0.0/0.0/0.0 | 0.0/0.1/0.2 | 0.0/0.0/0.7 |
| AutoWebGLM | 0.0/0.0/0.0 | 0.0/0.0/0.0 | 0.0/4.0/4.0 | 0.0/4.0/4.0 | 0.0/7.1/14.3 | 0.0/7.1/10.7 | 0.0/14.0/18.0 | 0.0/14.0/18.0 | 0.0/0.0/0.0 | 0.0/2.0/2.0 | 0.0/1.8/3.6 | 0.0/0.0/0.0 |
| WebArena | 0.0/1.4/1.4 | 0.0/0.0/0.0 | 0.0/1.2/2.4 | 0.0/0.0/0.0 | 2.7/8.1/13.5 | 2.7/12.0/22.7 | 1.0/36.5/46.9 | 8.3/39.6/51.0 | 0.0/0.0/0.0 | 0.0/0.0/0.0 | 1.0/1.0/3.9 | 0.0/0.0/0.0 |
| WebLINX | 0.0/0.0/1.0 | 0.0/0.0/0.0 | 0.3/0.8/1.1 | 0.0/0.5/1.1 | 4.3/8.7/10.9 | 4.3/10.8/17.2 | 2.4/36.0/47.3 | 16.6/47.0/57.2 | 0.0/0.0/0.0 | 0.0/0.0/0.0 | 0.0/0.5/0.7 | 0.0/0.0/0.0 |
| GUIAct | 0.0/0.0/0.0 | 0.0/0.0/0.0 | 0.0/0.0/0.0 | 0.0/0.0/0.0 | 0.0/0.0/0.5 | 0.0/0.0/0.5 | 0.1/19.4/24.9 | 23.8/36.0/39.4 | 0.0/0.0/0.0 | 0.0/0.0/0.0 | 0.0/0.0/0.1 | 0.0/0.0/0.0 |
| **TongUI-3B** | | | | | | | | | | | | |
| Mind2Web | 0.0/0.0/0.0 | 0.0/0.0/0.0 | 0.0/0.1/0.1 | 0.0/0.0/0.0 | 0.0/1.8/3.5 | 0.0/0.0/1.0 | 5.4/38.1/48.9 | 5.5/39.2/50.0 | 0.0/0.0/0.0 | 0.0/0.5/0.5 | 0.0/0.1/0.3 | 0.0/0.0/0.7 |
| AutoWebGLM | 0.0/2.0/2.0 | 0.0/0.0/0.0 | 2.0/2.0/4.0 | 0.0/2.0/4.0 | 3.6/14.3/28.6 | 3.6/12.5/21.4 | 0.0/10.0/14.0 | 0.0/8.0/18.0 | 0.0/0.0/0.0 | 2.0/2.0/2.0 | 0.0/0.0/1.8 | 0.0/3.3/3.3 |
| WebArena | 0.0/0.0/0.0 | 0.0/0.0/1.4 | 0.0/1.2/3.6 | 0.0/0.0/1.2 | 0.0/5.4/13.5 | 0.0/2.7/8.0 | 7.3/35.4/49.0 | 6.2/32.3/41.7 | 0.0/0.0/1.2 | 0.0/0.0/0.0 | 0.0/0.0/0.0 | 0.0/0.0/0.0 |
| WebLINX | 0.0/0.0/0.0 | 0.0/0.0/0.5 | 0.0/0.0/0.3 | 0.0/0.0/0.0 | 4.3/4.3/8.7 | 3.2/4.3/7.5 | 4.9/35.4/46.0 | 16.1/47.5/57.2 | 0.0/0.0/0.0 | 0.0/0.0/0.0 | 0.2/0.9/0.9 | 0.0/0.0/0.0 |
| GUIAct | 0.0/0.0/0.0 | 0.0/0.0/0.0 | 0.0/0.0/0.0 | 0.0/0.0/0.0 | 0.0/0.5/0.5 | 0.0/0.0/0.5 | 2.7/18.8/24.1 | 20.0/34.9/39.0 | 0.0/0.0/0.0 | 0.0/0.0/0.0 | 0.0/0.1/0.1 | 0.0/0.0/0.2 |
| **GAE-Retriever** | | | | | | | | | | | | |
| Mind2Web | 0.0/1.0/3.5 | 0.0/0.0/0.5 | 14.8/60.0/84.3 | 4.7/38.7/71.1 | 75.2/95.6/99.1 | 46.5/80.5/92.0 | 9.9/56.1/78.5 | 9.8/49.6/74.0 | 2.0/8.5/13.0 | 1.0/7.0/13.0 | 16.2/45.8/66.5 | 17.2/54.5/70.9 |
| AutoWebGLM | 12.0/48.0/56.0 | 2.0/16.0/32.0 | 12.0/64.0/82.0 | 2.0/38.0/62.0 | 85.7/96.4/100.0 | 55.4/82.1/91.1 | 4.0/46.0/70.0 | 6.0/42.0/58.0 | 4.0/22.0/40.0 | 0.0/6.0/14.0 | 21.4/78.6/91.1 | 20.0/56.7/70.0 |
| WebArena | 1.4/1.4/9.6 | 0.0/1.4/4.1 | 9.5/21.4/40.5 | 7.1/21.4/36.9 | 51.4/83.8/94.6 | 28.0/72.0/88.0 | 6.2/36.5/51.0 | 5.2/41.7/55.2 | 2.4/10.7/13.1 | 0.0/2.4/9.5 | 13.6/45.6/65.0 | 13.3/31.1/44.4 |
| WebLINX | 0.0/1.5/5.0 | 0.0/0.5/1.0 | 13.6/56.5/70.1 | 3.5/36.8/59.7 | 80.4/91.3/95.7 | 59.1/84.9/89.2 | 7.5/43.6/56.8 | 13.9/43.9/62.7 | 4.0/8.0/12.0 | 1.0/2.5/11.5 | 22.5/58.3/72.3 | 0.0/4.0/10.0 |
| GUIAct | 30.5/66.5/82.5 | 2.5/13.0/27.5 | 16.4/42.0/50.4 | 17.9/43.0/49.6 | 95.0/99.0/99.5 | 55.0/85.0/95.5 | 10.5/39.2/47.0 | 17.0/45.5/51.6 | 15.5/58.5/74.0 | 4.0/24.5/40.5 | 29.4/60.2/70.5 | 4.7/13.0/17.4 |

## D.2 BASELINE MODELS AND EVALUATION DETAILS

**Multimodal Backbone Models** We include two widely-used multimodal LLMs for fair comparison while controlling model size: *Qwen2-VL-2B* (Wang et al., 2024a) and *Qwen2.5-VL-3B* (Bai et al., 2025). Both support multi-image/video inputs and strong general-purpose perception-reasoning. We use the official inference recipes with unified decoding settings across tasks.

Table 10: Recall@1/5/10 for all methods on each subtask in the out-of-domain setting.

| Source | $(q, \tau_{1:i}) \to \tau_{i+1:m}$ | $(q, \tau_{i+1:m}) \to \tau_{1:i}$ | $(q, \tau_{1:i}) \to s_{i+1}$ | $(q, \tau_{i+1:m}) \to s_i$ | $q \to \tau_=$ | $q \to \tau_\infty$ | $(q, s_i) \to s_{i+1}$ | $(q, s_{i+1}) \to s_i$ | $(q, s_i) \to \tau_{1:i}$ | $(q, s_{i+1}) \to \tau_{1:i}$ | $q \to s_i$ | $q \to s_m$ |
|---|---|---|---|---|---|---|---|---|---|---|---|---|
| **Qwen2-VL-2B** | | | | | | | | | | | | |
| Mind2Web | 0.0/0.3/1.0 | 0.2/1.6/2.8 | 0.2/0.7/1.2 | 0.2/0.2/0.3 | 0.8/4.0/6.3 | 0.8/4.6/7.8 | 2.3/62.5/70.7 | 2.4/61.8/71.2 | 0.2/0.3/0.7 | 0.0/0.0/0.3 | 0.1/0.7/1.6 | 0.0/0.0/0.0 |
| AutoWebGLM | 0.0/0.0/0.0 | 0.0/0.0/0.0 | 0.0/7.5/7.5 | 0.0/2.5/12.5 | 0.0/5.0/5.0 | 0.0/0.0/4.0 | 0.0/22.5/30.0 | 0.0/27.5/35.0 | 2.5/5.0/5.0 | 0.0/2.5/2.5 | 1.7/3.3/5.0 | 5.0/10.0/10.0 |
| WebArena | 0.0/1.7/3.4 | 0.0/1.7/5.2 | 0.0/0.0/0.0 | 0.0/0.0/0.0 | 5.0/5.0/15.0 | 5.0/6.0/15.0 | 1.7/20.7/24.1 | 3.4/34.5/39.7 | 0.0/1.7/1.7 | 0.0/0.0/0.0 | 0.0/0.0/0.0 | 0.0/0.0/0.0 |
| WebLINX | 0.0/0.0/0.9 | 0.9/0.9/0.9 | 0.9/0.9/0.9 | 0.0/0.0/0.0 | 5.0/10.0/25.0 | 6.0/14.0/24.0 | 0.9/47.8/50.4 | 10.4/49.6/53.0 | 0.0/0.0/0.0 | 0.0/0.0/0.0 | 0.0/0.0/0.0 | 0.0/0.0/0.0 |
| GUIAct | 0.0/0.0/0.0 | 0.0/0.0/0.1 | 0.0/0.1/0.1 | 0.0/0.1/0.2 | 0.4/3.5/5.6 | 0.5/2.4/4.2 | 5.5/27.6/34.5 | 23.7/40.4/44.6 | 0.1/0.1/0.1 | 0.0/0.0/0.0 | 0.0/0.0/0.1 | 0.0/0.0/0.0 |
| **Qwen2.5-VL-3B** | | | | | | | | | | | | |
| Mind2Web | 0.0/0.3/0.3 | 0.0/0.0/0.0 | 0.0/0.2/0.2 | 0.0/0.2/0.2 | 0.0/0.0/0.0 | 0.0/0.5/1.4 | 2.3/31.8/37.2 | 5.9/34.2/38.7 | 0.0/0.2/0.2 | 0.0/0.0/0.2 | 0.0/0.0/0.1 | 0.0/0.0/0.0 |
| AutoWebGLM | 0.0/0.0/0.0 | 0.0/0.0/0.0 | 2.5/2.5/2.5 | 0.0/2.5/2.5 | 5.0/15.0/35.0 | 4.0/12.0/21.0 | 0.0/5.0/5.0 | 0.0/5.0/5.0 | 0.0/0.0/2.5 | 0.0/0.0/0.0 | 1.7/1.7/5.0 | 0.0/0.0/0.0 |
| WebArena | 0.0/1.7/1.7 | 0.0/0.0/0.0 | 0.0/0.0/6.9 | 0.0/0.0/6.9 | 0.0/5.0/15.0 | 0.0/6.0/10.0 | 3.4/12.1/13.8 | 1.7/13.8/20.7 | 0.0/0.0/0.0 | 0.0/0.0/0.0 | 0.0/0.0/4.1 | 0.0/0.0/0.0 |
| WebLINX | 0.0/0.0/0.9 | 0.0/0.0/0.0 | 0.0/0.0/0.0 | 0.0/0.0/0.0 | 0.0/0.0/5.0 | 0.0/3.0/8.0 | 5.2/38.3/43.5 | 12.2/43.5/47.8 | 0.0/0.0/0.0 | 0.0/0.0/0.0 | 0.0/8.0/8.0 | 0.0/0.0/5.0 |
| GUIAct | 0.0/0.0/0.0 | 0.0/0.0/0.0 | 0.0/0.0/0.2 | 0.1/0.1/0.2 | 0.0/1.4/4.2 | 0.7/1.6/3.3 | 2.3/29.0/36.5 | 21.1/41.0/47.7 | 0.1/0.5/0.7 | 0.0/0.0/0.0 | 0.1/0.2/0.2 | 0.0/0.0/0.0 |
| **LamRA-Ret** | | | | | | | | | | | | |
| Mind2Web | 0.0/0.3/0.9 | 0.0/0.3/1.0 | 0.2/1.0/2.1 | 0.3/0.7/1.7 | 4.8/14.3/20.6 | 2.1/9.0/15.4 | 2.1/54.5/62.8 | 3.1/55.8/63.5 | 0.0/0.3/0.7 | 0.0/0.0/0.0 | 1.2/5.8/8.6 | 2.4/5.6/7.1 |
| AutoWebGLM | 0.0/0.0/0.0 | 0.0/0.0/0.0 | 0.0/5.0/5.0 | 5.0/10.0/15.0 | 30.0/65.0/70.0 | 21.0/43.0/60.0 | 0.0/25.0/27.5 | 0.0/25.0/32.5 | 10.0/17.5/22.5 | 0.0/2.5/2.5 | 1.7/8.3/23.3 | 5.0/5.0/5.0 |
| WebArena | 0.0/0.0/3.4 | 0.0/3.4/5.2 | 0.0/0.0/0.0 | 0.0/0.0/0.0 | 10.0/35.0/35.0 | 6.0/18.0/27.0 | 0.0/19.0/27.6 | 5.2/27.6/32.8 | 0.0/0.0/0.0 | 0.0/0.0/0.0 | 2.7/6.8/8.2 | 5.0/5.0/5.0 |
| WebLINX | 0.0/0.0/0.0 | 0.0/0.0/0.0 | 0.0/2.6/4.3 | 0.0/2.6/6.1 | 15.0/50.0/55.0 | 4.0/30.0/48.0 | 1.7/48.7/53.9 | 13.0/51.3/56.5 | 0.0/0.0/0.0 | 0.0/0.0/0.0 | 0.9/0.9/0.9 | 5.1/15.3/19.5 |
| GUIAct | 0.0/0.0/0.0 | 0.0/0.0/0.0 | 0.1/0.2/0.2 | 0.3/0.7/1.2 | 27.8/51.8/60.6 | 12.3/30.4/38.5 | 0.7/26.6/33.4 | 23.3/40.3/44.7 | 0.2/0.7/1.8 | 0.0/0.0/0.0 | 0.6/1.8/3.2 | 0.4/1.8/1.8 |
| **ColQwen2-v1.0** | | | | | | | | | | | | |
| Mind2Web | 0.0/1.6/3.1 | 0.0/1.6/4.0 | 4.0/13.4/20.2 | 1.0/5.8/11.0 | 20.6/38.9/49.2 | 13.0/26.0/36.8 | 1.2/63.2/77.0 | 2.4/66.3/77.3 | 1.9/6.6/11.2 | 2.8/11.0/18.8 | 5.4/18.0/25.3 | 11.1/21.4/27.0 |
| AutoWebGLM | 0.0/0.0/0.0 | 0.0/0.0/0.0 | 2.5/5.0/10.0 | 5.0/35.0/45.0 | 10.0/35.0/55.0 | 0.0/20.0/35.0 | 0.0/55.0/60.0 | 0.0/55.0/60.0 | 0.0/7.5/10.0 | 0.0/0.0/0.0 | 13.3/25.0/33.3 | 5.0/15.0/35.0 |
| WebArena | 0.0/0.0/3.4 | 0.0/3.4/3.4 | 3.0/12.1/17.3 | 0.0/1.7/3.4 | 20.0/40.0/55.0 | 11.0/29.0/45.0 | 1.7/20.7/25.9 | 3.4/31.0/43.1 | 0.0/1.7/5.2 | 3.4/6.9/10.3 | 2.7/8.2/12.3 | 0.0/5.0/15.0 |
| WebLINX | 0.0/0.0/0.9 | 0.0/0.9/0.9 | 3.5/12.2/17.4 | 1.7/6.1/6.1 | 20.0/50.0/70.0 | 16.0/41.0/51.0 | 2.6/48.7/52.2 | 10.4/54.8/58.3 | 0.0/0.9/0.9 | 0.9/7.8/9.6 | 4.2/11.0/12.7 | 5.0/15.0/15.0 |
| GUIAct | 0.1/0.3/1.6 | 0.0/0.4/2.2 | 1.0/3.1/4.1 | 2.7/8.2/11.2 | 71.1/85.6/90.1 | 28.9/56.1/67.5 | 0.5/34.1/42.0 | 23.5/46.6/53.2 | 1.6/6.5/12.2 | 0.6/2.4/3.8 | 0.6/2.9/4.7 | 2.1/7.0/10.2 |
| **GME-Qwen2VL-2B** | | | | | | | | | | | | |
| Mind2Web | 0.0/2.1/6.6 | 0.0/1.7/6.5 | 4.2/18.5/25.3 | 4.2/14.8/21.3 | 30.2/57.1/73.0 | 16.5/37.1/47.6 | 2.4/65.3/75.4 | 2.3/66.5/76.8 | 3.0/13.4/27.9 | 3.3/13.1/26.5 | 5.5/15.4/21.1 | 4.0/14.3/18.3 |
| AutoWebGLM | 0.0/0.0/0.0 | 0.0/0.0/0.0 | 5.0/37.5/50.0 | 0.0/25.0/40.0 | 25.0/45.0/60.0 | 19.0/30.0/49.0 | 0.0/70.0/70.0 | 0.0/70.0/72.5 | 0.0/5.0/7.5 | 7.5/40.0/57.5 | 0.0/36.7/48.3 | 0.0/10.0/20.0 |
| WebArena | 0.0/3.4/5.2 | 0.0/6.9/8.6 | 0.0/3.4/8.6 | 0.0/6.9/10.3 | 25.0/70.0/75.0 | 15.0/44.0/65.0 | 5.2/24.1/29.3 | 5.2/29.3/41.4 | 3.4/6.9/20.7 | 1.7/5.2/8.6 | 0.0/4.1/5.3 | 0.0/10.0/20.0 |
| WebLINX | 0.0/1.7/1.7 | 0.0/2.6/5.2 | 7.8/22.6/34.8 | 5.2/16.5/20.0 | 60.0/85.0/90.0 | 29.0/59.0/71.0 | 2.6/52.2/55.7 | 11.3/53.9/59.1 | 0.0/6.1/9.6 | 0.9/8.7/19.1 | 4.2/7.6/12.7 | 10.0/20.0/20.0 |
| GUIAct | 0.0/0.0/0.0 | 0.0/0.0/0.0 | 2.2/8.6/11.9 | 3.6/11.8/16.1 | 66.2/87.3/93.9 | 22.5/52.2/64.9 | 1.1/35.6/44.2 | 23.3/47.9/53.6 | 1.0/2.6/4.2 | 2.6/9.7/15.4 | 1.2/3.5/4.7 | 0.4/6.3/10.6 |
| **UniSE-MLLM** | | | | | | | | | | | | |
| Mind2Web | 0.0/1.9/4.7 | 0.3/2.1/5.1 | 3.3/8.7/12.2 | 1.9/6.3/10.1 | 4.8/8.7/16.7 | 4.1/9.4/12.1 | 6.3/65.4/74.5 | 2.4/63.0/74.3 | 3.0/10.3/15.9 | 0.9/4.2/8.2 | 4.7/12.2/21.4 | 0.8/3.2/7.9 |
| AutoWebGLM | 0.0/0.0/0.0 | 0.0/0.0/0.0 | 2.5/10.0/15.0 | 0.0/2.5/5.0 | 5.0/5.0/25.0 | 5.0/8.0/25.0 | 0.0/35.0/42.5 | 0.0/32.5/42.5 | 5.0/15.0/15.0 | 0.0/2.5/2.5 | 0.0/6.7/11.7 | 5.0/5.0/10.0 |
| WebArena | 1.7/3.4/6.9 | 0.0/6.9/6.9 | 0.0/0.0/1.7 | 1.7/1.7/3.4 | 0.0/15.0/25.0 | 0.0/8.0/21.0 | 8.6/31.0/34.5 | 3.4/36.2/41.4 | 0.0/5.2/8.6 | 0.0/0.0/1.7 | 2.7/2.7/2.7 | 0.0/0.0/0.0 |
| WebLINX | 0.0/0.0/0.9 | 0.0/0.9/4.3 | 0.9/1.7/2.6 | 0.0/0.9/1.7 | 15.0/20.0/65.0 | 7.0/23.0/36.0 | 3.5/49.6/51.3 | 11.3/51.3/55.7 | 1.7/5.2/7.0 | 0.0/0.9/1.7 | 0.0/0.0/0.0 | 0.0/0.0/0.0 |
| GUIAct | 0.0/0.0/0.1 | 0.0/0.0/0.1 | 0.3/0.8/1.2 | 0.5/1.5/2.5 | 3.9/10.2/14.1 | 1.8/5.8/8.6 | 1.6/31.6/38.3 | 23.5/42.6/48.0 | 0.1/0.8/1.1 | 0.0/0.0/0.0 | 0.1/0.3/0.4 | 0.0/0.4/0.7 |
| **VLM2Vec-Qwen2VL-2B** | | | | | | | | | | | | |
| Mind2Web | 0.0/2.3/7.0 | 0.0/2.1/9.1 | 9.6/45.2/62.5 | 6.1/46.3/65.6 | 38.9/69.0/79.4 | 21.9/42.5/55.1 | 1.2/73.6/84.1 | 1.6/75.6/83.3 | 3.1/18.5/33.2 | 1.7/11.7/23.4 | 21.4/56.6/69.6 | 23.0/57.1/65.1 |
| AutoWebGLM | 0.0/0.0/5.0 | 0.0/2.5/7.5 | 7.5/60.0/72.5 | 7.5/72.5/87.5 | 50.0/90.0/95.0 | 31.0/61.0/75.0 | 0.0/87.5/90.0 | 0.0/85.0/92.5 | 7.5/47.5/70.0 | 5.0/30.0/57.5 | 36.7/70.0/81.7 | 20.0/40.0/45.0 |
| WebArena | 0.0/5.2/5.2 | 0.0/5.2/8.6 | 6.9/13.8/17.2 | 1.7/19.0/20.7 | 35.0/70.0/80.0 | 13.0/43.0/65.0 | 3.4/27.6/31.0 | 1.7/34.5/48.3 | 1.7/17.2/27.6 | 5.0/30.0/57.5 | 12.3/27.4/37.0 | 5.0/15.0/15.0 |
| WebLINX | 0.0/0.9/6.1 | 0.0/1.7/5.2 | 8.7/28.7/42.6 | 7.0/30.4/41.7 | 70.0/90.0/100.0 | 29.0/62.0/83.0 | 4.3/60.9/66.1 | 10.4/58.3/67.8 | 0.0/5.2/16.5 | 0.9/8.7/19.1 | 15.3/36.4/43.2 | 0.0/10.0/15.0 |
| GUIAct | 0.0/0.0/0.3 | 0.0/0.1/0.9 | 2.3/8.7/12.0 | 5.1/16.3/20.7 | 59.9/79.2/86.3 | 15.8/39.7/54.7 | 0.6/39.2/47.7 | 23.6/50.4/58.0 | 10.8/32.8/45.8 | 0.1/1.7/4.9 | 4.8/12.6/16.2 | 2.1/5.6/6.7 |
| **VLM2Vec-V2.2** | | | | | | | | | | | | |
| Mind2Web | 0.0/1.7/10.8 | 0.0/3.5/13.4 | 10.1/57.6/77.5 | 2.8/58.6/79.2 | 79.4/90.5/95.2 | 40.3/67.8/78.7 | 4.2/73.6/85.3 | 2.3/73.3/86.4 | 2.3/16.2/33.7 | 1.7/11.7/29.5 | 27.7/67.1/79.6 | 37.3/76.2/83.3 |
| AutoWebGLM | 0.0/20.0/60.0 | 0.0/22.5/65.0 | 0.0/85.0/92.5 | 0.0/90.0/95.0 | 75.0/85.0/95.0 | 46.0/67.0/77.0 | 0.0/87.5/97.5 | 0.0/92.5/95.0 | 2.5/47.5/77.5 | 0.0/32.5/67.5 | 56.7/86.7/91.7 | 20.0/60.0/60.0 |
| WebArena | 0.0/8.6/10.3 | 0.0/10.3/12.1 | 1.7/13.8/24.1 | 3.4/29.3/44.8 | 55.0/75.0/85.0 | 31.0/60.0/75.0 | 10.3/37.9/44.8 | 6.9/34.5/44.8 | 3.4/19.0/25.9 | 0.0/5.2/19.0 | 20.5/42.5/54.8 | 25.0/45.0/55.0 |
| WebLINX | 0.0/0.9/6.1 | 0.0/0.0/3.5 | 7.0/65.2/78.3 | 3.5/50.4/67.0 | 90.0/100.0/100.0 | 55.0/87.0/95.0 | 6.1/67.8/73.9 | 12.2/65.2/72.2 | 4.3/10.4/20.0 | 2.6/9.6/18.3 | 36.4/77.1/89.0 | 30.0/65.0/70.0 |
| GUIAct | 0.0/0.1/1.4 | 0.0/0.1/1.9 | 8.7/35.6/44.7 | 9.9/39.3/49.1 | 96.5/99.3/99.6 | 59.9/77.2/84.3 | 1.8/42.5/50.0 | 22.3/52.8/60.7 | 6.4/24.0/34.1 | 0.6/5.7/10.9 | 28.2/58.3/68.4 | 10.6/38.0/41.9 |
| **UGround-V1-2B** | | | | | | | | | | | | |
| Mind2Web | 0.0/0.2/0.5 | 0.2/0.7/1.0 | 0.3/0.5/1.2 | 0.2/0.9/1.0 | 0.8/4.8/7.9 | 0.8/4.3/8.1 | 2.6/50.1/57.6 | 2.1/50.1/57.6 | 0.0/0.2/0.5 | 0.0/0.2/0.5 | 0.3/0.6/0.7 | 0.0/0.0/0.0 |
| AutoWebGLM | 0.0/0.0/0.0 | 0.0/0.0/0.0 | 0.0/0.0/2.5 | 0.0/0.0/0.0 | 0.0/0.0/25.0 | 0.0/9.0/30.0 | 0.0/12.5/20.0 | 0.0/15.0/20.0 | 0.0/0.0/2.5 | 0.0/0.0/0.0 | 0.0/1.7/5.0 | 0.0/5.0/5.0 |
| WebArena | 0.0/0.0/0.0 | 0.0/0.0/1.7 | 3.4/3.4/3.4 | 0.0/1.7/1.7 | 5.0/25.0/45.0 | 5.0/23.0/43.0 | 0.0/19.0/25.9 | 1.7/27.6/34.5 | 0.0/0.0/0.0 | 0.0/0.0/0.0 | 0.0/1.4/1.4 | 0.0/0.0/0.0 |
| WebLINX | 0.0/0.0/0.0 | 0.0/0.0/0.0 | 0.0/0.0/0.0 | 0.0/0.0/0.0 | 0.0/0.0/15.0 | 0.0/6.0/15.0 | 2.6/47.8/49.6 | 12.2/50.4/53.9 | 0.0/0.0/0.0 | 0.0/0.0/0.0 | 0.9/0.9/0.9 | 0.0/0.8/0.8 |
| GUIAct | 0.0/0.0/0.0 | 0.0/0.0/0.0 | 0.1/0.4/0.5 | 0.1/0.1/0.4 | 0.0/2.1/3.9 | 0.4/1.8/3.8 | 0.6/27.8/35.9 | 23.4/40.4/46.8 | 0.0/0.0/0.0 | 0.1/0.2/0.4 | 0.1/0.1/0.1 | 0.0/0.0/0.0 |
| **ShowUI-2B** | | | | | | | | | | | | |
| Mind2Web | 0.0/0.5/0.9 | 0.0/0.7/0.9 | 0.0/0.0/0.0 | 0.0/0.2/0.2 | 0.0/1.6/4.8 | 0.0/1.6/3.8 | 3.3/58.5/67.4 | 3.1/55.8/65.4 | 0.0/0.2/0.2 | 0.0/0.0/0.0 | 0.3/0.7/0.9 | 0.0/0.8/0.8 |
| AutoWebGLM | 0.0/0.0/0.0 | 0.0/0.0/0.0 | 0.0/2.5/2.5 | 0.0/2.5/7.5 | 5.0/15.0/25.0 | 5.0/15.0/20.0 | 0.0/17.5/17.5 | 0.0/17.5/17.5 | 0.0/0.0/2.5 | 0.0/0.0/0.0 | 1.7/3.3/5.0 | 5.0/5.0/5.0 |
| WebArena | 0.0/0.0/0.0 | 0.0/0.0/1.7 | 0.0/1.7/1.7 | 0.0/1.7/1.7 | 0.0/5.0/5.0 | 1.0/2.0/7.0 | 1.7/24.1/29.3 | 1.7/31.0/34.5 | 0.0/0.0/0.0 | 0.0/0.0/0.0 | 0.0/0.1/1.4 | 0.0/0.0/0.0 |
| WebLINX | 0.0/0.0/0.0 | 0.0/0.9/1.7 | 0.0/0.0/0.0 | 0.0/0.9/2.6 | 0.0/5.0/15.0 | 0.0/2.0/17.0 | 5.2/47.8/49.6 | 11.3/46.1/52.2 | 0.0/0.0/0.0 | 0.0/0.0/0.0 | 0.0/0.0/0.0 | 0.0/0.0/0.0 |
| GUIAct | 0.0/0.0/0.0 | 0.0/0.0/0.0 | 0.0/0.1/0.1 | 0.0/0.1/0.2 | 0.4/2.5/3.5 | 0.4/2.7/4.2 | 0.7/27.2/33.8 | 23.6/40.2/44.6 | 0.0/0.1/0.1 | 0.1/0.1/0.1 | 0.0/0.0/0.0 | 0.0/0.0/0.4 |
| **UI-TARS-2B-SFT** | | | | | | | | | | | | |
| Mind2Web | 0.2/0.5/0.7 | 0.0/0.0/0.2 | 0.0/0.2/0.2 | 0.0/0.0/0.7 | 0.8/0.8/1.6 | 0.5/0.6/2.5 | 1.0/51.3/57.8 | 3.3/50.6/58.3 | 0.0/0.0/0.0 | 0.0/0.2/0.3 | 0.0/0.1/0.1 | 0.0/0.8/0.8 |
| AutoWebGLM | 0.0/0.0/0.0 | 0.0/0.0/0.0 | 0.0/2.5/5.0 | 2.5/2.5/5.0 | 5.0/5.0/35.0 | 5.0/13.0/32.0 | 0.0/15.0/20.0 | 0.0/10.0/12.5 | 0.0/0.0/0.0 | 0.0/0.0/0.0 | 0.0/0.0/0.0 | 0.0/0.0/0.0 |
| WebArena | 0.0/0.0/1.7 | 0.0/1.7/3.4 | 0.0/1.7/3.4 | 0.0/0.0/0.0 | 0.0/0.0/25.0 | 0.0/4.0/14.0 | 3.4/19.0/24.1 | 5.2/25.9/34.5 | 0.0/0.0/0.0 | 0.0/0.0/0.0 | 0.0/0.0/0.0 | 0.0/0.0/0.0 |
| WebLINX | 0.0/0.0/0.0 | 0.0/0.0/0.0 | 0.0/0.9/0.9 | 0.0/0.9/0.9 | 0.0/0.0/5.0 | 0.0/2.0/8.0 | 9.9/44.3/48.7 | 13.0/47.0/54.8 | 0.0/0.0/0.0 | 0.0/0.0/0.0 | 0.0/0.0/0.0 | 0.0/0.0/0.0 |
| GUIAct | 0.0/0.0/0.1 | 0.0/0.0/0.0 | 0.1/0.1/0.1 | 0.1/0.3/0.3 | 1.4/2.8/3.9 | 0.4/2.7/3.7 | 0.1/27.6/34.3 | 23.6/39.8/44.7 | 0.1/0.1/0.2 | 0.0/0.0/0.0 | 0.0/0.0/0.0 | 0.0/0.0/0.0 |
| **TongUI-3B** | | | | | | | | | | | | |
| Mind2Web | 0.0/0.0/0.0 | 0.0/0.2/0.5 | 0.2/0.3/0.7 | 0.0/0.0/0.2 | 0.8/3.2/5.6 | 1.0/3.2/5.7 | 4.0/35.3/39.8 | 4.4/36.8/39.8 | 0.0/0.0/0.0 | 0.0/0.0/0.0 | 0.3/0.9/0.9 | 0.8/0.8/0.8 |
| AutoWebGLM | 0.0/0.0/0.0 | 0.0/2.5/5.0 | 0.0/0.0/5.0 | 0.0/5.0/15.0 | 0.0/5.0/15.0 | 0.0/8.0/19.0 | 0.0/0.0/0.0 | 0.0/0.0/0.0 | 0.0/2.5/2.5 | 0.0/2.5/2.5 | 0.0/3.3/5.0 | 0.0/10.0/10.0 |
| WebArena | 0.0/0.0/0.0 | 0.0/0.0/1.7 | 0.0/0.0/0.0 | 0.0/1.7/1.7 | 5.0/15.0/30.0 | 4.0/17.0/28.0 | 5.2/13.8/17.2 | 5.2/19.0/22.4 | 0.0/1.7/1.7 | 0.0/0.0/0.0 | 0.0/2.7/4.1 | 0.0/0.0/0.0 |
| WebLINX | 0.0/0.0/0.9 | 0.0/0.0/0.0 | 0.0/0.0/0.0 | 0.0/0.0/0.0 | 0.0/15.0/25.0 | 1.0/15.0/31.0 | 7.0/47.0/52.2 | 10.4/46.1/53.9 | 0.0/0.0/0.0 | 0.0/0.0/0.0 | 0.0/0.0/0.0 | 0.0/0.0/0.0 |
| GUIAct | 0.0/0.0/0.1 | 0.0/0.0/0.0 | 0.1/0.1/0.1 | 0.0/0.1/0.1 | 0.4/2.5/5.3 | 0.4/2.5/4.8 | 3.8/29.3/37.1 | 20.2/41.7/47.8 | 0.0/0.0/0.0 | 0.0/0.0/0.0 | 0.0/0.1/0.1 | 0.0/0.0/0.0 |
| **GAE-Retriever** | | | | | | | | | | | | |
| Mind2Web | 9.4/31.1/49.7 | 0.9/14.8/30.4 | 21.5/76.4/88.5 | 3.3/59.3/76.3 | 65.9/93.7/97.6 | 32.4/72.5/84.3 | 18.8/77.7/88.8 | 16.8/71.7/88.8 | 9.2/33.2/47.5 | 1.4/24.1/45.0 | 34.1/75.0/85.3 | 46.0/74.6/80.2 |
| AutoWebGLM | 25.0/77.5/92.5 | 2.5/57.5/82.5 | 12.5/97.5/97.5 | 7.5/97.5/100.0 | 60.0/90.0/95.0 | 45.0/73.0/82.0 | 10.0/87.5/100.0 | 5.0/95.0/95.0 | 40.0/70.0/95.0 | 2.5/57.5/82.5 | 65.0/96.7/100.0 | 45.0/65.0/75.0 |
| WebArena | 10.3/27.6/34.5 | 1.7/13.8/22.4 | 12.1/36.2/50.0 | 5.2/32.8/41.4 | 25.0/50.0/70.0 | 13.8/37.9/48.3 | 12.1/41.4/48.3 | 10.3/27.6/39.7 | 0.0/17.2/22.4 | 26.0/58.9/72.6 | 0.0/25.0/40.0 |  |
| WebLINX | 7.0/21.7/37.4 | 0.0/4.3/13.0 | 13.0/73.9/77.4 | 5.2/51.3/64.3 | 80.0/100.0/100.0 | 58.0/87.0/98.0 | 17.4/71.3/77.4 | 20.9/66.1/74.8 | 7.0/20.9/34.8 | 2.6/14.8/36.5 | 45.8/85.6/94.1 | 35.0/85.0/90.0 |
| GUIAct | 47.4/95.6/98.4 | 23.2/60.5/78.9 | 25.0/54.8/59.6 | 22.7/57.5/64.3 | 91.9/97.9/99.3 | 59.4/85.9/94.0 | 16.2/53.4/59.4 | 20.6/57.5/64.6 | 34.6/93.8/98.4 | 8.6/41.9/67.2 | 41.8/81.8/89.7 | 19.0/44.0/49.3 |

**Multimodal Retrieval Models** This group covers text-image retrieval, document understanding, screenshot retrieval, and mixed-modality pretraining: *Lamra-Ret* (Liu et al., 2024) (language-multimodal retrieval), *VLM2Vec-Qwen2VL-2B* (Jiang et al., 2025) (VLM-based embedding for retrieval), *ColQwen* (Faysse et al., 2024) (visual document / page-level understanding), *UniSE-MLLM* (Liu et al., 2025b) (UI/screenshot retrieval), *GME* (Zhang et al., 2024c) (general multimodal embedding), and *VLM2Vec-v2.2* (Meng et al., 2025) (upgraded mixed-modality embedding). All models are run with consistent top-$k$ retrieval and identical normalization for similarity scoring.

**Multimodal Trajectory Planning Models** We benchmark four GUI-agent models trained on web/mobile/desktop interaction data: *UGround-V1-2B* (Gou et al., 2024) (visual grounding + UI manipulation), *ShowUI-2B* (Lin et al., 2024a) (screen understanding and pointer actions), *UI-TARS-2B-SFT* (Qin et al., 2025) (UI navigation with supervised finetuning), and *TongUI-3B* (Zhang et al., 2025) (multiplatform GUI navigation). We use zero-shot execution without task-specific finetuning.

**Common Evaluation Protocol** All baselines are evaluated zero-shot with a unified prompt template and the same decoding hyperparameters (temperature, top-$p$, max tokens). For multimodal inputs, we standardize image resolution and frame sampling. For retrieval models, we fix the index,

similarity metric, and top-$k$. Any model-specific pre/post-processing strictly follows the official recommendations in the cited papers to ensure fairness.

## D.3 Detailed Analysis of Multimodal Retrieval Baselines

Among retrieval-centric baselines, **VLM2Vec-v2.2** provides both the highest scores and the lowest variance across benchmarks. Its consistent improvements over **VLM2Vec-Qwen2VL-2B** are plausibly attributed to the integration of visual document and video corpora during pretraining, which expands the distributional coverage beyond image-text pairs. This aligns with the broader pattern that the **VLM2Vec** family outperforms other approaches: training on fused-modality data (text, images, documents, and videos) with interleaved batching appears to strengthen representation alignment and negative sampling diversity, both crucial for trajectory-oriented retrieval.

By contrast, **LamRA-Ret**, trained primarily on text-image datasets, lags behind models exposed to structured visual content such as **ColQwen2-v1.0** and **GME-Qwen2VL-2B**. While LamRA-Ret still improves over backbone-only baselines, its gap indicates that text-image supervision alone is insufficient when queries/targets involve layout cues, dense text regions, or multi-panel context common in GUI trajectories. Between the document-pretrained lines, **GME-Qwen2VL-2B** surpasses **ColQwen2-v1.0**, likely benefiting from a broader mixture of visual and textual inputs beyond pure documents, which improves cross-domain generalization. Finally, **UniSE-MLLM**, despite screenshot-centric finetuning, records the weakest outcomes and occasionally underperforms backbone baselines, suggesting that narrow, screenshot-only specialization can impede flexibility for open-domain retrieval where both semantic breadth and structural variance matter.

## D.4 Per-Task Evaluation

Figure 8 presents Recall@5 results broken down by retrieval task across multiple datasets for five representative models: ColQwen2-v1.0, GME-Qwen2VL-2B, VLM2Vec-Qwen2VL-2B, VLM2Vec-V2.2, and GAE-Retriever, under both in-domain and out-of-domain settings. As depicted, all models achieve their best results on the $q \rightarrow \tau$ and $q \rightarrow s$ tasks, indicating that semantic retrieval tasks are relatively simple. On the contrary, performance declines significantly on $(q, \tau) \rightarrow \tau'$ and $(q, s) \rightarrow \tau$, which can be attributed to the absence of similar supervision in existing benchmarks.

GAE-Retriever attains the highest scores on nearly every task over all datasets, especially evident on GUIAct and AutoWebGLM. Notably, it performs even better in out-of-domain scenarios than in-domain ones for tasks like $q \rightarrow s$, $(q, \tau) \rightarrow \tau'$, and $(q, s) \rightarrow \tau$, showcasing the robust generalization capability of our proposed framework.

Among baseline methods, VLM2Vec-V2.2 stands out as the most competitive, showing solid results across most tasks. It even surpasses GAE-Retriever on $q \rightarrow \tau$ in the out-of-domain WebArena, which can be attributed to its enhanced transfer capabilities from video pretraining. However, it consistently falls short of GAE-Retriever on more complex tasks, highlighting the necessity of retrieval models optimized for the multimodal trajectory retrieval task.

VLM2Vec-Qwen2VL-2B and GME-Qwen2VL-2B achieve moderate results but remain behind the top two models. It can be observed that they outperform the document-only pretrained ColQwen2-v1.0, proving the importance of training on diverse domains. The relatively poor performance of ColQwen2-v1.0 suggests its limited adaptability beyond documents. Meanwhile, GME-Qwen2VL-2B's gap with its VLM2Vec counterpart points to deficiencies in training.

Overall, the consistent advantage of GAE-Retriever reinforces the value of trajectory-based modeling and retrieval training for building versatile multimodal retrieval systems.

## D.5 Comparison of Different Retrieval Methods.

To substantiate that our retrieval strategy surpasses the conventional text-only trajectory retrieval $q \rightarrow \tau$, we conduct a controlled comparison on 50 tasks from Online-Mind2Web (13 Easy, 26 Medium, 11 Hard). The state-conditioned multimodal formulation $(q, s) \rightarrow s'$ encodes the current page state (DOM/screenshot) together with the task instruction and directly retrieves an executable next step under visually similar states, achieving visual alignment and localized guidance. In contrast, $q \rightarrow \tau$

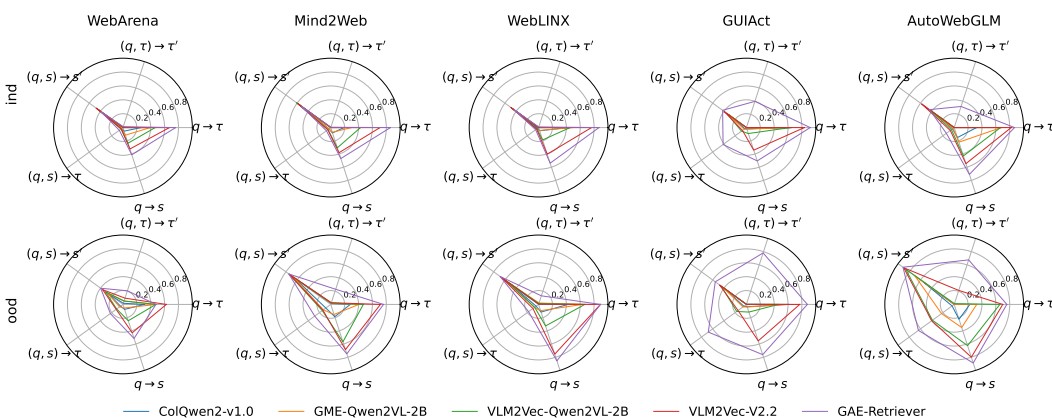

Figure 8: Per-Task Evaluation Results. Recall@5 is reported for five selected models across various retrieval tasks and datasets under in-domain (ind) and out-of-domain (ood) scenarios.

relies solely on textual semantic matching, which often introduces redundant exemplars misaligned with the current layout, leading to a mismatch between guidance and decision needs.

From the results in Figure 11, we observe that the multimodal approach consistently outperforms the text-only baseline across all difficulty levels: on *Easy* tasks, success increases from 69.23% to 76.92%, and on *Medium/Hard* tasks the gains are substantial (approximately +30.77 and +18.18 percentage points, respectively). These results corroborate the superiority of the proposed multimodal, state-conditioned retrieval.

Table 11: Comparison of Different Retrieval Methods.

| Retrieval Strategy | Success Rate (%) | | | | #Average | |
| | Easy | Medium | Hard | All | Step | Reward |
|---|---|---|---|---|---|---|
| $q \to \tau$ | 69.23 | 23.07 | 18.18 | 34 | 33.36 | 7.58 |
| $(q, s) \to s'$ | **76.92** | **53.84** | **36.36** | **56** | 24.21 | **8.06** |

### D.6 EVALUATING TRAJECTORY RETRIEVAL IN ADVANCED AGENTS

To validate the effectiveness of trajectory retrieval across different SOTA agent frameworks, we integrate GAE-Retriever as a plug-and-play module into SeeAct (Zheng et al., 2024) under the same experimental setup. As shown in Table 12, even without extensive prompt tuning, introducing trajectory knowledge consistently improves task performance, yielding an 8–10% gain in success rate. The improvement is particularly pronounced on medium and hard tasks. Comparing text-based and vision-based retrieval, we observe that visual knowledge performs better on Medium/Hard settings, indicating that visual exemplars are more helpful for fine-grained operations in complex interactive scenarios. These results demonstrate that our method is highly extensible across different agent frameworks.

Table 12: Performance of SeeAct with Trajectory Retrieval on Tasks of Different Levels.

| Method | Easy (%) | Medium (%) | Hard (%) | Success Rate (%) |
|---|---|---|---|---|
| SeeAct | 71.42 | 47.36 | 6.25 | 49.00 |
| + $\mathcal{D}_{\text{text}}$ | **78.57** | 55.26 | 18.75 | 57.00 |
| + $\mathcal{D}_{\text{vision}}$ | 76.19 | **60.52** | **25.00** | **59.00** |

### D.7 INFERENCE-TIME EFFICIENCY OF WEBRAGENT WITH GAE-RETRIEVER

In WebRAGent, each planning step introduces an additional round of GAE Retriever encoding, vector retrieval, and LLM reranking. Compared with the baseline, this does introduce some extra computation, but the overhead is very small relative to the entire task execution: it results in roughly

10% additional latency per step, which we consider fully acceptable in practice. We have applied several engineering optimizations to the retrieval and LLM reranking stages, including a two level cache on the LLM side and optimized and compressed image encoding. Our efficiency experiments are mainly conducted on an NVIDIA RTX A6000 GPU. Table 13 shows that the dominant time cost does not come from the GAE Retriever retrieval stage, but from the LLM reranking, while the overall time per step remains modest. When the cache is hit, most queries require almost no extra reranking time and the corresponding inference cost is nearly negligible.

Table 13: Inference-time efficiency of GAE-Retriever and LLM reranking.

| Setting | Retrieval (s/step) | Rerank (s/step) | Total (s/step) | Speedup |
|---|---|---|---|---|
| No Cache | 0.3877 | 3.5374 | 3.9251 | 1.0× |
| With Cache | 0.3828 | 0.0002 | 0.3830 | 9.24× |

### D.8 DISCUSSION ON THE RISK OF MISSING SMALL, LOW-CONTRAST UI ELEMENTS

In our framework, the UI graph token selection module is only applied during training, with the primary goal of saving GPU memory and enabling a larger batch size. During inference, we always feed the full-resolution patch grid produced by the standard Qwen-VL preprocessing pipeline.

To address the potential risk of losing small yet critical UI elements, we only merge patches that are spatially adjacent and whose color difference is below a certain threshold (i.e., $\|\mathbf{c}_i - \mathbf{c}_j\|_2 \le 1$ in RGB space). In typical cases, common buttons or checkboxes exhibit a noticeable contrast against the background (as shown in the left part of Figure 9). If a checkbox is so similar to the background that it is indistinguishable even to the human eye (as in the right part of Figure 9), then, in such an extreme case, the agent would also be unable to distinguish it. We thus treat this checkbox as effectively nonexistent, which is reasonable from the agent's perspective. Moreover, even in extreme cases where a very small number of UI elements might be merged away, this can only happen during training; at inference time, we still feed the full patch grid as input.

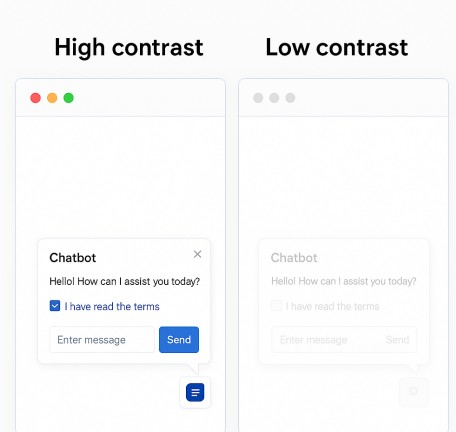

Figure 9: Comparison of High-Contrast vs. Low-Contrast UI Checkboxes

## E PROMPT

### E.1 ANNOTATION PROMPTS AND ACTION TEMPLATES

This section demonstrates the prompts used in the paper. Table 14 includes prompts for data annotation such as state description generation, HTML rendering, and silver data generation. Snippet 1 specifies the action space definitions used for Mind2Web, WebLINX, WebArena, GUIAct, and AutoWebGLM. Snippets 2 and 3 show the instruction templates used to augment retrieval queries, where *description* refers to the natural language summary of a trajectory.

Table 14: Data annotation prompts.

| Function | Content |
|---|---|
| Description Generation | Generate a concise one-sentence description of the content and layout of the provided webpage screenshot. |
| HTML Rendering | Your task is to convert the simplified HTML input provided by the user into a fully renderable, standard HTML format while preserving all original information intact. Enhance the HTML with appropriate styling to make it visually appealing and resemble a typical, functional website. Return only the HTML code without any additional text. HTML: *html*. Ensure that the returned HTML code includes the ID [*id*] (mentioned in *context*) with the same element exactly as provided. |
| Silver Generation | **Step 1**: You are a helpful AI assistant proficient in Named Entity Recognition (NER). Analyze the following sentence and provide the most comprehensive NER results for each noun in JSON format, using greedy matching. Labels should be specific contextual descriptions of the entity. Sentence: *instruction*. |
| | **Step 2**: You are an AI assistant skilled in generating alternatives. Given a sentence and a list of named entities, generate five alternative texts for each entity that align with its semantic label while being entirely different in meaning from the original text. Ensure the alternatives fit naturally and consistently within the sentence, maintaining the original representation (e.g., text remains text, emojis remain emojis). Sentence: *instruction*, Named Entities: *ners*. |
| | **Step 3**: You are an AI assistant specialized in rewriting user queries. Your task is to refine the following five queries to ensure they are consistent, natural, concise, logical, and human-like. Rewrite each query by varying the wording, structure, and style to ensure diversity in expression. Your response should align with real-world common sense and factual accuracy. |

---

## SNIPPET 1: ACTION SPACE

**Mind2Web**

1. click: Simulates a mouse click on the target element (bounding box).
2. type: Types the specified value (str) into the target text input element (bounding box).
3. select: Selects the specified value (str) from a target dropdown element (bounding box).

**WebLINX**

1. click: Simulates a mouse click on the target element (bounding box).
2. hover: Simulates hovering over the target element (bounding box).
3. textInput: Types the value (str) into the target element (bounding box).
4. change: Changes the value of the target element (bounding box) to the specified value (str).
5. load: Loads the webpage at the specified url value (str).
6. submit: Submits the form identified by the target element (bounding box).
7. scroll: Scrolls the page to the specified coordinate values in the list of floats [x, y].
8. copy: Copies the specified text value (str) from the target element (bounding box).
9. paste: Pastes the specified text value (str) into the target element (bounding box).

**WebArena**

1. click: Simulates a mouse click on the target element (bounding box).
2. press: Simulates the pressing of a key combination value (str) on the target element (bounding box).
3. selectOption: Selects the specified option value (str) from the target dropdown element (bounding box).
4. check: Checks the target checkbox element (bounding box).

**GUIAct**

1. click: Clicks on the target element (bounding box).
2. hover: Hovers over the target element (bounding box).
3. input: Inputs the given text value (str) into the target element (bounding box).
4. scroll: Scrolls the screen by the values in the list of coordinate floats [down, right], where down represents vertical scroll and right represents horizontal scroll.
5. select_text: Selects text by dragging across the specified coordinate values in the list of floats [x1, y1, x2, y2], where (x1, y1) is the starting point and (x2, y2) is the ending point.
6. copy: Copies the specified text value (str) to the clipboard.
7. enter: Simulates pressing the Enter key.
8. select: Selects the text value (str) in the target element (bounding box).
9. answer: Provides an answer or response specified by text value (str) to the user.

**AutoWebGLM**

1. click: Clicks on the target element (bounding box).
2. hover: Hovers over the target element (bounding box).
3. select: Selects the option value (str) from a dropdown target element (bounding box).
4. type_string: Types the specified content (str) into the target element (bounding box) and presses Enter if press_enter (bool) is True. The action value is a list [content, press_enter].
5. scroll_page: Scrolls the page in the specified direction value ('up' or 'down').
6. go: Navigates browser history in the specified direction value ('forward' or 'backward').
7. jump_to: Opens the specified url (str) and optionally in a new tab if new_tab (bool) is True. The action value is a list [url, new_tab].
8. switch_tab: Switches to a browser tab specified by the value tab_index (int).
9. user_input: Displays the specified message (str) to obtain user input.
10. finish: Completes the task with an optional answer value (str or None).

SNIPPET 2: INSTRUCTION TEMPLATE

$(q, \tau_{1:i}) \rightarrow \tau_{i+1:n}$

Determine the next web navigation trajectory using the task instruction "*description*" and the prior trajectory below.
Retrieve the upcoming web navigation trajectory as specified by the task "*description*" and the previous trajectory provided.
Search the next phase of the web navigation trajectory based on the user query "*description*" and the earlier trajectory.
From the user input "*description*" and the past navigation steps, locate the subsequent navigation sequence for GUI agents.
Apply the request "*description*" to the previous web navigation steps to derive the next trajectory.
Represent the previous trajectory for web agents below to determine the next trajectory based on the task "*description*".
According to the previous web agent trajectory below, identify the next sequence of steps to complete the user instruction "*description*".
Based on the previous trajectory below, search the next interaction sequence of web agents for the user request "*description*".
With the previous GUI navigation trajectory below as a guide, look for the next trajectory for the user intention "*description*".
Given the goal "*description*" and the earlier navigation sequence from GUI agents, match the subsequent trajectory.

$(q, \tau_{i+1:n}) \rightarrow \tau_{1:i}$

Find the previous web browsing trajectory based on the user input "*description*" and the current trajectory.
Identify the former web navigation history by analyzing the user request "*description*" and the provided trajectory.
With the instruction "*description*" and the following interaction sequence, extract the earlier trajectory for web agents.
Determine the past trajectory using the goal "*description*" and the succeeding navigation sequence for GUI agents.
Retrieve the preceding GUI navigation trajectory with the task description "*description*" and the succeeding trajectory below.
Using the user query "*description*" and the succeeding navigation steps for web agents, locate the preceding interaction steps.
Analyze the query "*description*" along with the provided web agent trajectory to derive the former navigation steps.
Based on the request "*description*" and the later web interaction trajectory, look for the prior navigation sequence.
Consider the task "*description*" together with the given trajectory to retrieve the prior web navigation trajectory.
Represent the current trajectory for web agents below to determine the previous trajectory according to the task "*description*".

$(q, \tau_{1:i}) \rightarrow s_{i+1}$

Identify the upcoming state from the earlier web navigation trajectory below and the instruction "*description*".
Extract the next state from the provided previous navigation sequence for web agents and the directive "*description*".
Locate the following state from the given former web navigation trajectory and the task input "*description*".
Determine the subsequent observation from the task "*description*" and the earlier web interaction trajectory.
Find the next observation by considering the command "*description*" and the former GUI navigation trajectory.
Using the user instruction "*description*" and the former web navigation trajectory, ascertain the subsequent state.
Based on the former web interaction history provided and the user request "*description*", deduce the next state.
Use the user intention "*description*" and the earlier navigation sequence for GUI agents to derive the following state.
Find the next state based on the goal "*description*" and the earlier interaction history for web agents.
Represent the given GUI navigation history to locate the upcoming state according to the user intention "*description*".

$(q, \tau_{i+1:n}) \rightarrow s_i$

Find the antecedent state using the command "*description*" and the current web navigation trajectory.
Identify the prior state by applying the directive "*description*" along with the present navigation trajectory for GUI agents.
Locate the former observation using the instruction "*description*" and the upcoming web interaction trajectory.
Ascertain the preceding observation based on the task input "*description*" and the provided web agent browsing sequence.
Determine the previous state by employing the goal "*description*" along with the upcoming navigation trajectory for web agents.
Find the antecedent state with the user instruction "*description*" and the succeeding GUI navigation trajectory.
Retrieve the former observation given the request "*description*" and the upcoming web navigation history.
Identify the prior state based on the task "*description*" and the succeeding web interaction trajectory.
Represent the provided GUI navigation history to retrieve the previous state using the user query "*description*".
Recognize the prior observation using the user task "*description*" and the given web navigation trajectory.

$q \rightarrow \tau_{\equiv}$

Determine the complete web navigation trajectory based on the following instruction "*description*".
Locate the equivalent web navigation trajectory derived from the following user input "*description*".
Find the GUI navigation history aligning with the following goal "*description*".
Match the corresponding trajectory for web agents using the following user query "*description*".
Pinpoint the equivalent navigation trajectory for GUI agents with the following task "*description*".
Ascertain the corresponding web interaction trajectory using the request "*description*".
Identify the unique navigation trajectory for web agents according to the provided instruction "*description*".
Determine the complete GUI navigation trajectory from the user task "*description*".
Ascertain the unique GUI interaction history by considering the task "*description*".
Locate the exactly equivalent web navigation trajectory based on the given query "*description*".

$q \rightarrow \tau_{\sim}$

Determine the analogous web navigation trajectory based on the following directive "*description*".
Identify the similar web navigation history using the task input "*description*".
Locate the akin navigation sequence for GUI agents as dictated by the user input "*description*".
Retrieve the similar web browsing trajectory as specified by the instruction "*description*".
Identify the similar GUI interaction history based on the task description "*description*".
Locate the analogous navigation trajectory for web agents using the instruction "*description*".
Retrieve the analogous interaction history for GUI agents based on the provided command "*description*".
Find a similar GUI navigation history following the task "*description*".
Extract a similar web browsing trajectory based on the instruction "*description*".
From the user query "*description*", match the similar web agent navigation trajectory.

SNIPPET 3: INSTRUCTION TEMPLATE (CONT.)

$(q, s_i) \rightarrow s_{i+1}$

Retrieve the following web navigation state according to the instruction "*description*" and the prior state.
Determine the subsequent web navigation observation given the task description "*description*" and the preceding state.
Retrieve the upcoming observation for web navigation agents following the user input "*description*" and the previous state.
Identify the next navigation state for GUI agents using the goal "*description*" and the preceding state.
Using the provided instruction "*description*" and the former state, what is the next GUI navigation state?
From the query "*description*" and the prior observation, derive the next state in the web navigation trajectory.
Given the user input "*description*" together with the current state, find the next web navigation state.
Taking the task input "*description*" and the former state, what is the subsequent GUI navigation state?
Considering the directive "*description*" and the preceding observation, determine the next web navigation state.
With the task "*description*" and the previous state from web agents as inputs, determine the subsequent state.

$(q, s_{i+1}) \rightarrow s_i$

Retrieve the prior web navigation state using the task "*description*" along with the current state.
In light of the instruction "*description*" and the current state provided, deduce the prior GUI navigation state.
Based on the provided user input "*description*" and the current observation, find the prior navigation state for web agents.
With the directive "*description*" and the current state, determine the prior web browsing state.
Considering both the current web agent observation provided and the user intention "*description*", locate the prior navigation state.
Given the present state and the goal "*description*", determine the previous GUI navigation state.
Combine the task description "*description*" with the current state to identify the preceding navigation state for GUI agents.
Taking the description "*description*" and the current state into account, search the previous web agent state.
Utilize the user request "*description*" alongside the present state to extract the prior GUI navigation state.
Use the directive "*description*" with the current state to look for the state directly preceding in the web agent navigation trajectory.

$(q, s_i) \rightarrow \tau_{i+1:n}$

Find the subsequent web navigation trajectory based on the instruction "*description*" and the previous state.
Based on the task "*description*" and the previous observation, identify the subsequent GUI navigation trajectory.
Locate the next GUI navigation trajectory by applying the instruction "*description*" to the previous state.
With the user input "*description*" and the previous state in hand, identify the next navigation trajectory for web agents.
What is the following navigation trajectory for GUI agents when applying the user intention "*description*" to the previous state?
Given the user goal "*description*" and the previous state, search the next web navigation trajectory.
When given the user instruction "*description*" and the former state, what is the next trajectory for web navigation?
Identify the next web navigation trajectory by merging the task "*description*" with the previous state.
From the directive "*description*" and the prior state, look for the subsequent GUI navigation trajectory.
Determine the subsequent browsing trajectory for web agents with the task "*description*" and the previous state as references.

$(q, s_{i+1}) \rightarrow \tau_{1:i}$

Find the previous web navigation history based on the instruction "*description*" and the current state.
Retrieve the preceding web navigation trajectory using the intention "*description*" along with the present state.
From the instruction "*description*" and the present state, find the prior GUI navigation history.
What does the previous navigation history for web agents look like when derived from the user input "*description*" and the current state?
Locate the prior GUI navigation history by combining the description "*description*" with the current observation.
Identify the web navigation trajectory preceding the current state according to the task "*description*".
Derive the previous navigation trajectory for GUI agents by combining the instruction "*description*" with the current state.
Search the browsing history for web agents that came before the provided current observation with regard to the user query "*description*".
Based on the current state and the user intention "*description*", extract the trajectory that came before in the web navigation.
Recognize the GUI navigation history that predates the current state by considering the user need "*description*".

$q \rightarrow s_i$

Find the specific web navigation state corresponding to the description "*description*".
Identify the equivalent web navigation state as defined by the description "*description*".
Extract the GUI navigation observation that corresponds with "*description*".
Locate the web navigation state as dictated by the description "*description*".
Identify the navigation state for GUI agents that is equivalent to the details provided in "*description*".
Search the observation for web navigation that best fits the details described in "*description*".
Determine the precise GUI navigation observation that reflects the input "*description*".
Retrieve the navigation observation for web agents that best aligns with the input "*description*".
What is the corresponding web browsing state described by the input "*description*"?
From the description "*description*", identify the specific navigation observation for GUI navigation.

$q \rightarrow s_n$

Retrieve the last web navigation observation based on the task "*description*".
From the task "*description*", locate the final state in the web navigation sequence.
What is the ultimate web navigation state for the task instruction "*description*"?
Find the end state in the GUI navigation as defined by the task "*description*".
Determine the last state in the web browsing process for the task "*description*".
Locate the final observation of the web navigation for the task "*description*".
Identify the concluding status in the GUI agent trajectory for the task "*description*".
Based on the task "*description*", extract the final navigation observation for web agents.
What is the final GUI navigation status according to the task "*description*"?
Search the terminal observation in the web navigation for the task instruction "*description*".

## E.2 PROMPT FOR WEBRAGENT'S PLANNING MODEL

The system prompt used by WebRAGent's planning model is given in Snippet 4, while Snippet 5 presents the reward prompt and Snippet 6 presents the LLM-reranking prompt.

---

**SNIPPET 4: SYSTEM PROMPT USED FOR PLANNING MODEL**

**Prompt**

You are an assistant who not only helps to browse and operate web pages to achieve certain goals, but also needs to explore the information on the page to answer the questions raised by the target task. Please answer the following questions as much as possible.
  ## There are key information you will get
    ∗∗Key Information∗∗:
      − Previous trace: all thoughts, actions and reflections you have made historically.
      − Current webpage screenshot: The webpage screenshot of the current execution step.
      − Similar Task Reference: The solution of a similar task, which can be appropriately used as a reference.
    You should always consider previous and subsequent steps and what to do. Do not ask for permission or confirmation. Act decisively.
    ∗∗Thought Space∗∗:
      − What action do you think is needed now to complete the task?
      − What's the reason of taking that action?
    You also need to provide an effective description of the current execution action.
    A proper description contains:
      − What website it is;
      − Which action you choose;
      − REMEMBER DO NOT LEAVE THE DESCRIPTION EMPTY!
  ## ALLOWED ACTIONS
  − click, double_click, type, scroll, keypress, drag, wait, get_final_answer
  ## COMPLETION RULES
  − INFORMATION tasks: when all requested info is visible  use get_final_answer with the content.
  − ACTION tasks: when final action succeeds and success signal/confirmation is visible  get_final_answer.
  ∗∗Special Circumstances Guidelines∗∗:
    − When performing a search on a website, if you find the search results do not display sufficient content, consider simplifying or modifying your search query. Reducing the complexity of your search query or altering keywords may yield more comprehensive results.
  ## RAG Usage Protocol (Do Not Violate)
    You may receive a list of retrieved example steps from a knowledge base. Each example contains:
      − task_name: the original task title
      − step_number: the step index in that task
      − observation_description: what the page looked like in that step
      − action_description: what was done and why
    ### How to use RAG (5−stage procedure)
    1) Goal Alignment:
      − Summarize the current sub−goal in one short clause (mentally).
      − If the examples sub−goal differs substantially (domain mismatch, authentication state mismatch, paywall/login state mismatch, or different product/site section), DO NOT reuse low−level details.
    2) Evidence Anchors (from the accessibility tree):
      − Extract concrete anchors you can actually verify now: role (link/button/input/select), visible text substrings, aria−label/hint words, URL host/path pattern, presence of known sections (nav/search/filter/cart/profile).
      − Only consider an example highly similar if 2 anchors match.
    3) Reuse Level Decision (choose exactly one):
      − ∗∗NONE∗∗ (similarity too low): Ignore example details. Use only generic high−level strategy.
      − ∗∗HIGH_LEVEL∗∗ (some similarity, <2 anchors matched): Borrow the intent/strategy but ∗∗do not∗∗ reuse element−level moves.
      − ∗∗DETAIL∗∗ (2 anchors match and same site section): Borrow the next action ∗∗type∗∗ (click/fill/select/go_back/ goto) and ∗∗selection logic∗∗, but you MUST re−locate the element by current accessibility tree (NEVER copy example element_id).
    4) Adaptation Plan:
      − Translate example−specific values into current−task values (e.g., search keywords, product IDs, email addresses).
      − If an example uses a search then filter pattern, replicate the pattern only if the corresponding widgets exist now.
    5) Safety & Fallbacks:
      − Never reuse example 'element_id'.
      − If no viable anchor is found within the current accessibility tree, downgrade reuse level and choose a safer generic action (e.g., open site search, go to category page, reveal filters).
      − If the last action failed, avoid repeating it; try a sibling anchor or a simpler query.
    ### Mandatory Reporting (inside your 'description')
    At the END of your 'description', append a single line tag block (keep the exact keys):
    '[RAG_USED=<yes/no>; RAG_REUSE=<NONE|HIGH_LEVEL|DETAIL>; RAG_SOURCES=<task_name:step,... or −>; RAG_REASON=<short>; CONF=<0.00−1.00>]'''
    ## Response Format:
    − "thought": Your reasoning and immediate action plan (use decisive language, no questions or confirmations)
    − "action": The specific action to execute NOW
    − "action_input": Parameters for the action
    − "element_id": Target element identifier
    − "description": Brief description of the action being executed (not what you "could" do)
    − "example_reference": How you applied insights from examples (if applicable)
  """

---

## SNIPPET 5: PROMPT FOR REWARD

**Prompt**

"""You are an assistant to help navigate and operate the web page to achieve certain task.
Your goal is to evaluate the previous series of traces(thoughts and actions) and think about what key steps are needed to complete the task in the future.
There are key information you will get:
**Key Information**:
  − Previous trace: all thoughts, actions and reflections you have made historically.
  − Accessibility tree: characteristic expression of the current web page.
  − Screenshot: visual information of the current web page (may include).

You also need to combine the previous trace to give the completion status of the current task.
**Status Of Task Completion**
  − doing: You have completed the intermediate steps of the target task but not entirely finish the target task.
  − finished: You are entirely certain about completing the target task.
  − loop: You find that the the last two steps of previous actions are the same, it is determined that the process is stuck in a local optimum solution.

You will judge and score the task completion and reasonableness of previous actions. The score ranges from 1−10, but the score you give can only be selected from [1, 3, 7, 9, 10].
**Judging and Scoring Criteria**:
  − score = 1: You find that the status of the task is stuck in a loop by analyzing the previous trace.
  − score = 3: You find that performing the previous trajectories(thoughts and actions) is not likely helpful in completing target task and you need to adjust the direction of your planning and action or start over from beginning.
  − score = 7: You find that performing the previous trajectories(thoughts and actions) are helpful in completing the target task.
  − score = 9: You find that performing the previous trajectories(thoughts and actions) are a very critical intermediate step to complete this task.
  − score = 10: You find that performing the previous trajectories(thoughts and actions) have completed the task perfectly.
You need to provide an effective evidence of scoring for the series of the previous trace.
  − Why do you give this score?
  − What is the reason?

You also need to provide an effective description or summary of the above requirements through key information and characteristics of the current web page.
**A proper description contains**:
  − What is the current completion status of the task? (IMPORTNAT)
  − REMEMBER DO NOT LEAVE THE DESCRIPTION EMPTY!

**Output Requirements**:
− Ensure your output strictly follows this format:
  ```json
  {
      "status": "ACTUAL_STATUS",
      "score": "ACTUAL_SCORE",
      "reason": "ACTUAL_REASON",
      "description": "ACTUAL_DESCRIPTION"
  }
  ```
− A VALID JSON BLOB EXAMPLE AS FELLOWS:
  ```
  {
      "status": "doing",
      "score": "3",
      "reason": "You need to complete a search for camping tents that can accommodate 2 people and sort the results in rei by price from low to high. According to your previous trajectory, you navigated to the rei official website and clicked the 2−person button, which are correct actions. But when you complete the final step of sorting prices, you actually click on a link to a tent product. This is a completely unreasonable action. So I give it 3 points."
      "description": "According to the current web page information, you can know that this is the homepage of a tent product, which is not very consistent with the purpose of the target task."
  }
  ```

---

SNIPPET 6: PROMPT FOR LLM-RERANKING

**Prompt**

"""
You are a web interface image matching expert. I will provide you with one query web interface image, a query task description, and {len(candidates_text)} candidate web interfaces, each with corresponding web interface images and task descriptions.

Query Task Description: {query_task_description}

Please carefully analyze the query image and candidate images, and match them based on task descriptions, including:
– Interface elements (buttons, input fields, text, etc.)
– Interface layout and design
– Interface visual similarity
– Matching degree between query task and candidate task descriptions

Candidate Information:
{"".join(candidates_text)}

Please select the most matching candidate from the above options, considering the following main factors:
1. Image visual similarity
2. Interface element matching degree
3. Relevance between query task and candidate task descriptions

Please answer:
The most matching candidate number (1–{len(candidates_text)})

Answer format:
Best matching candidate: [number]
"""

---

# F    CASE STUDY

## F.1    CASE STUDY: TEXTUAL TRAJECTORIES OF SUCCESSFUL RETRIEVAL

Below is the trajectory data (text) of some successful cases using the RAG method. Purple text denotes the models internal thoughts, while red text highlights the portions that reference retrieved knowledge.

---

RAG Successful Case 1

**Task name:** Browse recipes for gluten-free chocolate chip cookies that can be made without nuts.

| Step 1 |
| --- |

[think]
To search for gluten-free chocolate chip cookie recipes without nuts, I should enter a precise query into the search bar on the Cookpad homepage. The example confirms this is the correct first interaction, recommending using the search bar with a clear query.
[/think]
**[action_history]**
gluten-free chocolate chip cookies without nuts  fill_form
**[/action_history]**

| Step 2 |
| --- |

[think]
Since I have already entered the search query into the input for 'gluten-free chocolate chip cookies without nuts', the correct next step is to submit the search using the associated button to trigger recipe results. The example confirms this is the standard flow.
[/think]
**[action_history]**
button  click
**[/action_history]**

---

**Step 3**

[think]
The search button has been clicked, so the next step is to review the displayed search results for recipes matching 'gluten-free chocolate chip cookies without nuts'.textcolorred I should identify recipe cards or links that describe gluten-free chocolate chip cookies and assess their suitability, and if possible, further refine the filtering to exclude nuts using any visible filters or input boxesjust as done in the example, where a filter input for 'Show me recipes without' was used. First, I need to look for any recipe list or individual results related to the search to begin this review.
[/think]

[action_history]
Extract and cache the titles and descriptions of recipe links/results visible after submitting the search for 'gluten-free chocolate chip cookies without nuts', as well as any available filters or search refinement tools (such as input boxes for excluding ingredients). This data will help determine whether the resulting recipes fit the criteria (gluten-free and nut-free) and guide the next steps if further filtering is needed. cache_data
[/action_history]

**Step 4**

[think]
The search did not return a focused list of recipe results; instead, the page mostly displays links to popular ingredients and general featured recipes. There do not appear to be targeted search results or a visible recipe list matching 'gluten-free chocolate chip cookies without nuts.' Given that the relevant recipes are not surfaced, textcolorredI should simplify the searchre-entering just 'gluten-free chocolate chip cookies' may return more comprehensive results or trigger filters, as shown in the RAG example when initial queries do not immediately yield the desired recipe list. After submission, I will check for an ingredient exclusion filter where 'nuts' can be entered. First, I will try making the search query less restrictive.
[/think]

[action_history]
gluten-free chocolate chip cookies fill_form
[/action_history]

**Step 5**

[think]
The accessibility tree shows a 'Filters' link and also an input labeled 'Type ingredients...' under the recipe/filter section. To align with the workflow in the similar example, the next step is to use the available filter to exclude nuts. textcolorredI should first click the 'Filters' link to open the filtering interface. After this, I can use the input field to specify 'nuts' for exclusion, ensuring the recipe results fit the user's dietary requirements.
[/think]

[action_history]
link click
[/action_history]

**Step 6**

[think]
Now that the 'Filters' interface has been opened, my next step is to exclude 'nuts' using the appropriate filter input. The accessibility tree shows two input boxes labeled 'Type ingredients...' under the filter section. Based on the example and the standard UI flow, I should type 'nuts' into the second filter input to specify ingredients to exclude, as it most likely corresponds to 'Show me recipes without:' from the similar example. This will update the recipe list to only include gluten-free chocolate chip cookies without nuts.
[/think]

[action_history]
nuts fill_form
[/action_history]

**Step 7**

[think]
Now that the nut exclusion filter has been applied for 'gluten-free chocolate chip cookies', I should review the updated recipe list and open a few promising recipes to verify their ingredient lists for both gluten-free and nut-free compliance. Recipes such as 'GF Chocolate Chip Cookies', 'Gluten-Free Chocolate Chip Cookies', 'Gluten-Free! Chocolate Chip Cookies', and similar titles are likely candidates. I will select one, open the recipe, and check the detailed ingredients. This is exactly what the similar example did after setting the dietary filters.
[/think]
[action_history]
link click
[/action_history]

**Step 8**

[think]
To validate if 'GF Chocolate Chip Cookies' are both gluten-free and nut-free, I need to review its ingredient list. I will extract and analyze the main listed ingredients now, specifically checking for any mention of wheat, barley, rye (gluten sources), or nuts including almond flour (as 'almond flour' is often used in gluten-free baking but is a tree nut, which may not meet the 'nut-free' requirement). I need to capture the precise ingredient data before determining suitability.
[/think]
[action_history]
Extract and analyze the full ingredient list used in the 'GF Chocolate Chip Cookies' recipe to check for the presence of nuts or gluten-containing ingredients. cache_data
[/action_history]

**Step 9**

[think]
Having determined that 'GF Chocolate Chip Cookies' are not nut-free due to the inclusion of almond flour, the correct next step is to return to the recipe results list. I need to open another promising recipeone with a title like 'Classic Chocolate Chip Cookies', 'EZ Chocolate Chip Cookies', or 'Best Chocolate Chip Cookies'to review its ingredients and check compliance. I will select the next most promising recipe from the visible results list.
[/think]
[action_history]
link click
[/action_history]

**Step 10**

[think]
To determine if 'Classic Chocolate Chip Cookies' by Joraine Villanueva meets the gluten-free and nut-free requirements, I need to extract and review the recipe's ingredient list. The key checkpoints: (1) if 'all purpose flour' is used, it contains gluten (so not gluten-free); (2) check for nuts or nut flours among the ingredients to ensure it is nut-free.
[/think]
[action_history]
Extract the detailed ingredient list for 'Classic Chocolate Chip Cookies' by Joraine Villanueva: butter, granulated sugar, brown sugar, vanilla, all purpose flour, baking soda, salt, chocolate chips. Analyze for presence of gluten or nuts. cache_data
[/action_history]

**Step 11**

[think]
Having determined that the current recipe ('Classic Chocolate Chip Cookies') is not gluten-free (contains all-purpose flour), I should proceed to open the next promising recipe in

the filtered results list with a gluten-free-focused title, such as 'Best Chocolate Chip Cookie', and repeat the ingredient review process. This aligns closely with both the detailed workflow and fallback loop in the provided RAG examplesystematically opening subsequent recipes and checking ingredients until a compliant gluten-free and nut-free option is found or all candidates are exhausted.
[/think]
[action_history]
link click
[/action_history]

**Step 12**

[think]
I need to examine the ingredient list for the 'Best Chocolate Chip Cookie' recipe by Helen Klassen to determine if it is both gluten-free and nut-free. Specifically, I am looking for signs of gluten (such as wheat flour) or any nuts/nut-derived ingredients. The ingredient section typically appears near prominent keyword links on Cookpad recipes, e.g., 'butter', 'brown sugar', 'flour', 'baking soda', 'chocolate chips'. I will extract these linked ingredient keywords for analysis.
[/think]
[action_history]
Extract the main ingredients for 'Best Chocolate Chip Cookie' by Helen Klassen, focusing on links near ingredient-related keywords such as 'salted butter', 'brown sugar', 'vanilla', 'flour', 'baking soda', 'baking powder', 'salt', 'chocolate chips', and 'chocolate' to verify the presence or absence of gluten and nuts. cache_data
[/action_history]

---

RAG Successful Case 2

**Task name:** Browse the latest negative reviews from players with over 100 hours of playtime for the game that won the 2023 VR Game of the Year Award.

**Step 1**

[think]
To determine the 2023 VR Game of the Year winner, I should navigate to the 'Steam Awards' section. I see a link labeled 'The Steam Awards' under 'New & Noteworthy,' which likely leads to the awards list. Opening this should reveal the winning VR title, which is my required first step before proceeding to reviews.
[/think]
[action_history]
link click
[/action_history]

**Step 2**

[think]
To identify the 2023 VR Game of the Year winner, I should closely examine the content of this 'Steam Awards Archive' page for the relevant award and the corresponding game name before proceeding to its store page to seek reviews.
[/think]
[action_history]
Extract the name of the game that won the 2023 VR Game of the Year Award from this 'Steam Awards Archive' page. cache_data
[/action_history]

**Step 3**

[think]

The critical next step is to determine which game won the 2023 VR Game of the Year Award by examining the text or content on the current 'The Steam Awards Archive' page. There isn't a direct text node or easy-to-access label in the accessibility tree showing the award winners, nor are there section headings or static content nodes revealing the information. Therefore, in the absence of any clearly labeled VR Game of the Year winner in the tree, I should use the page's search functionality to look up '2023 VR Game of the Year' or similar, so I can quickly find the specific winner's name before proceeding to its store page for the review substeps.
**[/think]**
**[action_history]**
2023 VR Game of the Year  fill_form
**[/action_history]**

**Step 4**

[think]
The current step is to identify the game that won the 2023 VR Game of the Year Award so I can proceed to negative reviews filtering. From the RAG example, the user used the Steam search function with the query
[/think]
**[action_history]**
link 'Arizona Sunshine Remake'  click
**[/action_history]**

**Step 5**

[think]
Now that I am on the Arizona Sunshine Remake store page, my immediate sub-goal is to display user reviews and apply filters. Based on the accessibility tree, there is a cluster of review filter buttons ('Review Type', 'Playtime', etc.) around element_ids 258-264. The first logical move is to open the 'Review Type' or 'Playtime' filter so I can start applying the conditions (over 100 hours, negative). Following the example, the correct pattern is to interact with these filter controls. 'Playtime' is crucial for the 100+ hours condition, so I'll open that filter first.
[/think]
**[action_history]**
button 'Playtime'  click
**[/action_history]**

**Step 6**

[think]
Having opened the 'Playtime' filter, the next logical step is to select the minimum hours played option and set it to 100 to isolate users with more than 100 hours. In the accessibility tree, there is a link labeled 'Minimum hours played' (element_id 265), which likely leads to an interface where the minimum threshold can be set. Selecting this will allow me to input or choose 100 as required for the task.
[/think]
**[action_history]**
Minimum hours played  click
**[/action_history]**

**Step 7**

[think]
Having clicked the 'Minimum hours played' option, my next step is to actually set the minimum value to 100, as required by the task. There does not appear to be a slider or dropdown, so the next logical interaction is to check for an input form or field to directly specify the hours. If present, I should type '100' into this field, fulfilling the playtime criteria. After this, I will continue with negative review filtering.
[/think]
**[action_history]**
100  fill_form

**[/action_history]**

### Step 8

[think]
Now that the playtime limit UI is open from previous steps, I need to locate and interact with a form or input field where I can enter '100' as the minimum hours played. There are no non-form filter sliders visible in the accessibility tree, so the likely next step is to find and fill any visible input for minimum hours with '100'. This mirrors the RAG example in setting a precise threshold, which is required before proceeding to filter for negative reviews.
[/think]
**[action_history]**
100  fill_form
**[/action_history]**

### Step 9

[think]
With the playtime filter set to a minimum of 100 hours, the next critical step is to filter reviews to only show negative ones. The accessibility tree shows a button labeled 'Review Type' (element_id 258) near the other filter controls. Clicking this button should open options (such as 'All', 'Positive', 'Negative'), allowing me to select 'Negative' so that only negative reviews from players with over 100 hours are displayed. This directly mirrors the example workflow after applying the playtime filter.
[/think]
**[action_history]**
button  click
**[/action_history]**

### Step 10

[think]
Having opened the 'Review Type' filter, the next required move is to select 'Negative' so that reviews are limited to only negative ones from players with over 100 hours, as per the target task. This mirrors the action taken in the example after playtime criteria were applied.
[/think]
**[action_history]**
Negative  click
**[/action_history]**

### Step 11

[think]
Now that the negative review filter has been activated, the final step is to sort these filtered reviews by 'most recent' so that the latest negative feedback from players with over 100 hours is shown on the Arizona Sunshine Remake Steam store page. In the accessibility tree, there is a 'Display' button at element_id 267, which is likely to serve as a sorting or display options dropdown this follows the workflow from the similar example, where the reviews were explicitly sorted by latest/most recent after filtering by review type and playtime. Clicking this will allow me to ensure the reviews are displayed from newest to oldest, finishing the multi-filter process required by the task.
[/think]
**[action_history]**
button  click
**[/action_history]**

### Step 12

[think]
The display/sort menu has just been opened (by clicking the 'Display' button at element_id 267) and, per the workflow from both my prior steps and the highly similar RAG example, the critical next step is to explicitly choose the sort orderspecifically, to select 'Most Recent'

or 'Latest' from the available display/sort options. This will sort the already filtered reviews (negative, 100+ hours) so that the most recent feedback is shown at the top, precisely meeting the requirements of the task. The accessibility tree does not show child elements under 'Display,' so a click is necessary to reveal/select the sort order. Once sorted, I will be able to confirm and browse these filtered, latest negative reviews.
**[/think]**
**[action_history]**
Sort reviews by 'Most Recent' (likely via the display dropdown that was opened)  click
**[/action_history]**

### F.2 CASE STUDY: VISUAL TRAJECTORIES OF SUCCESSFUL RETRIEVAL

Tables 15–17 present successful cases under the Vision RAG mode in a visualized manner. For each step, we provide the corresponding screenshot and Action information, enabling straightforward and intuitive inspection.

Table 15: Vision RAG Success Example Steps 0-2

**Instruction:** Find Farms land in Wilkes County, NC with the lowest price.

**Observation 0**

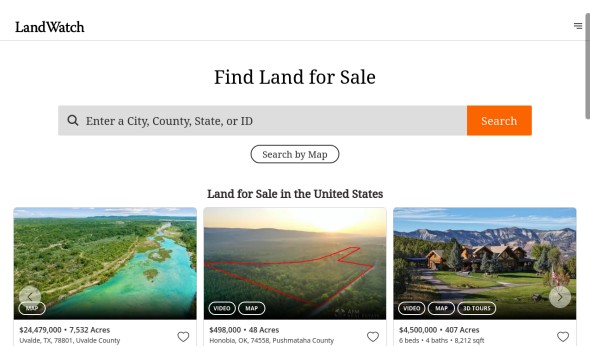

**Action 1**

`operator_click(328,233)`

**Observation 1**

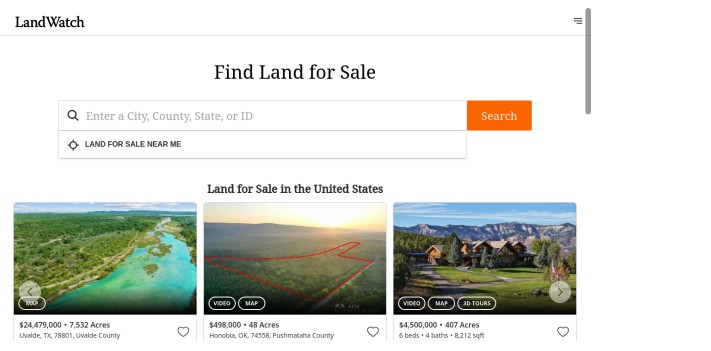

**Action 2**

`operator_type(Wilkes County, NC)`

**Observation 2**

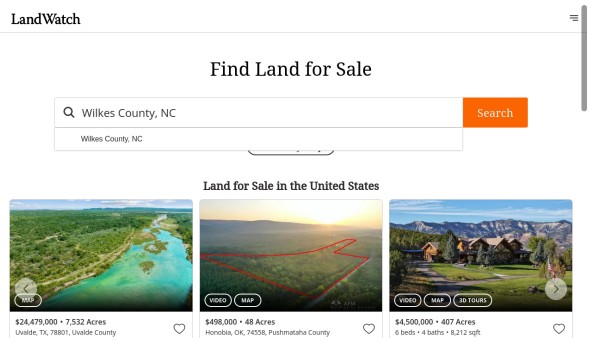

**Action 3**

`operator_click(445,292)`

Table 16: Vision RAG Success Example Steps 3-5

**Observation 3**

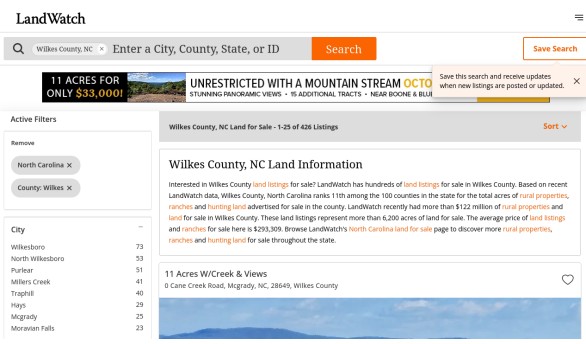

**Action 4**

`operator_click(1219,255)`

**Observation 4**

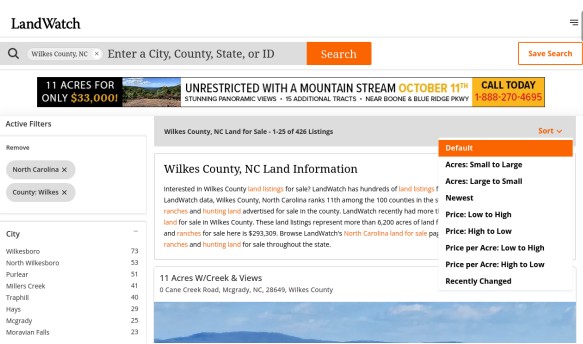

**Action 5**

`operator_click(1005,440)`

**Observation 5**

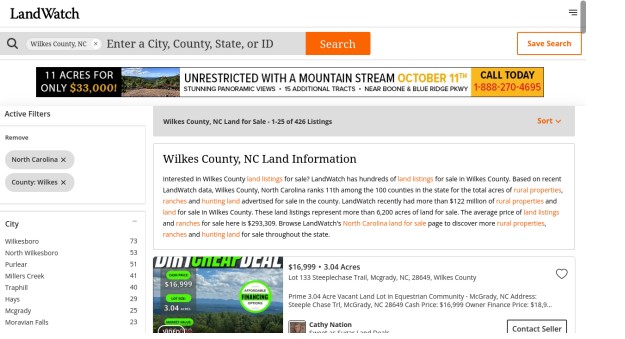

**Action 6**

`operator_scroll(0,420)`

Table 17: Vision RAG Success Example Steps 6-7

**Observation 6**

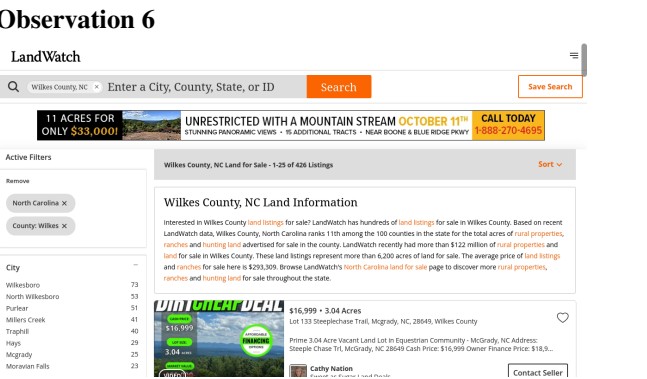

**Action 7**

`operator_scroll(0,420)`

**Observation 7**

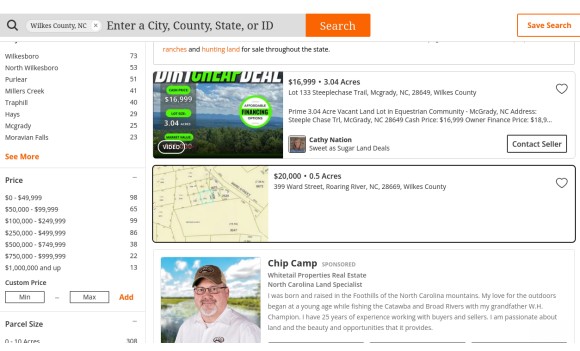

**Action 8**

`operator_wait(1000)`

## F.3 TEXTUAL COMPARISON: RAG VS. NON-RAG

We present below, in textual form, whether RAG is utilized or not, followed by a detailed comparative analysis.

---

### RAG vs. Non-RAG 1

**Task name:** Browse iPhone X for sale that is in good condition, has a max price of 400, and searches in titles only.

#### RAG Step 1

[think]
Based on the typical Craigslist flow and the strong match with the example, my next step should be to enter 'iPhone X' into the search box to start browsing listings relevant to the task requirements. This aligns with the similar example, where the search begins by entering the product name.
[/think]
[action_history]
iPhone X  fill_form
[/action_history]

#### RAG Step 2

[think]
The next step is to filter the results by 'good' condition since we've just completed entering the search term. The example shows that clicking the 'condition' filter button will bring up the options for selecting item states like 'good.' This must be done before applying other filters.
[/think]
[action_history]
button 'condition'  click
[/action_history]

#### RAG Step 3

[think]
The condition filter pop-up is now open, so I need to select the 'good' condition option. The RAG example shows this as the immediate next step. After checking 'good', I'll apply the filter, then advance to pricing and titles-only filters.
[/think]
[action_history]
Select 'good' within the condition filter pop-up to filter listings to only those described as in 'good' condition.  click
[/action_history]

#### RAG Step 4

[think]
With the 'good' condition selected, the next RAG-aligned step is to click 'apply' to confirm the filter within the pop-up, just as in the example. This ensures the search is updated to only show items in good condition before proceeding to max price and titles-only filters.
[/think]
[action_history]
button  click
[/action_history]

#### RAG Step 5

[think]
With the 'good' condition filter applied, the next step is to set the maximum price by entering '400' into the appropriate input field. According to both the RAG example and the accessibility tree, the 'max' input field is available and ready for interaction. This will further refine the search to only include iPhone X listings with prices that meet the specified maximum.
[/think]

**[action_history]**
400 fill_form
**[/action_history]**

## RAG Step 6

**[think]**
With the 'good' condition and max price filters set, the last required filter is 'search in titles only.' This is usually a checkbox near the search box or within sidebar filters on Craigslist. Scanning the provided accessibility tree, the related option isn't immediately visible, suggesting it may be in a collapsed, sidebar, or advanced filter section—often found near the main search input. To proceed like in the RAG example, I should look for a button or section that could expand, reveal, or activate additional search options, especially one near the search form. The 'submit' button (element_id 11) is next to the 'search for sale' input and is a likely action to refresh/apply current filters and surface any missing advanced options, especially if UI renders more filtering tools after submission. Clicking it is a safe way to expose or set the titles-only filter, aligning with common Craigslist interface patterns and the analogous example.
**[/think]**
**[action_history]**
button 'submit' click
**[/action_history]**

## RAG Step 7

**[think]**
The 'search in titles only' option is a required filter for this task, but it's not visibly present in the current accessibility tree—even after refreshing search filters. The rest of the filters (max price and condition) are already set. Since there's no titles-only checkbox, the best next step in reference to Craigslist's interface (and the RAG example, where this option is often in the sidebar near search or under advanced search options) is to look for any additional buttons or expandable areas near the search input or filter section that could reveal more search options. The 'for sale' filter button (element_id 4) is at the top of the category/filter controls, and sometimes Craigslist nests advanced filters under category/section buttons. By clicking this, I may reveal or trigger additional options, including the sought 'titles only' filter.
**[/think]**
**[action_history]**
button click
**[/action_history]**

## RAG Step 8

**[think]**
To comprehensively address the remaining requirement of 'search in titles only', I need to perform a final thorough check for any advanced, sidebar, or otherwise hidden filter settings that could reveal a corresponding option—since nothing titled 'titles only' appears in the main or expanded filter controls thus far. Scanning all buttons near the filter/search area (such as the gear/settings icon, secondary search controls, or miscellaneous options buttons) is the most methodical approach. The 'related searches' button (element_id 29) stands out, as sometimes additional or advanced filters are exposed via such miscellaneous controls. Activating this may reveal an area not currently visible in the accessibility tree and aligns with the example's step of checking all sidebar/advanced controls. If it doesn't reveal a titles-only option, this will serve as final confirmation that the filter is unavailable in this Craigslist region, allowing the task to be concluded accordingly.
**[/think]**
**[action_history]**
button click
**[/action_history]**

## Non-RAG Step 1

**[think]**

To browse iPhone X for sale, I need first to navigate to the 'for sale' category, and more specifically to 'cell phones', to better target the listings. This will allow me to subsequently apply relevant filters for condition, price, and title search.
**[/think]**
**[action_history]**
link click
**[/action_history]**

**Non-RAG Step 2**

[think]
Now that I'm in the 'cell phones' for sale section of Craigslist, I need to locate and interact with the search form. This will allow me to enter 'iPhone X' as the keyword, filter for 'good' condition, set the max price to 400, and select 'search titles only'.
[/think]
**[action_history]**
iPhone X fill_form
**[/action_history]**

**Non-RAG Step 3**

[think]
I've previously indicated the intent to enter 'iPhone X' into the search input. Now, to further meet the requirements, I should fill in the 'max' input with '400' to set the maximum price. This incrementally builds the required search criteria.
[/think]
**[action_history]**
400 fill_form
**[/action_history]**

**Non-RAG Step 4**

[think]
My next step is to enter 'iPhone X' in the search input to begin filtering for listings matching the target device. The search input has element_id 10, and this will set up the query before applying other filters like condition and title search.",
[/think]
**[action_history]**
iPhone X fill_form
**[/action_history]**

**Non-RAG Step 5**

[think]
To continue with the next most logical step and further refine the search as specified, I should now click the 'condition' button (element_id 18) so that I can select 'good' as the target condition for listings. This directly addresses a key filter in the task description.
[/think]
**[action_history]**
iPhone X fill_form
**[/action_history]**

**Non-RAG Step 6**

[think]
Now that the 'condition' filter is open, the next logical action is to select 'good' as the target condition for the listings. This will ensure that only listings in good condition are shown, as required by the task.
[/think]
**[action_history]**
button click
**[/action_history]**

**Non-RAG Step 7**

[think]
The 'condition' filter panel is open, so now I need to choose 'good' as the desired condition. This action is essential to limit results to iPhone X listings that are in good condition, as required by the task. Based on the accessibility tree, the interface may use button selections for conditions. Likely options such as 'all', 'owner', or 'dealer' do not fit 'good' condition; the 'condition' filter may require another button press or direct text selection, but 'apply' is typically used to confirm a selection. Therefore, the correct next step is to click 'apply' (element_id 3) to confirm the current 'good' condition setting and return to the main filter area.
[/think]
[action_history]
apply click
[/action_history]

**Non-RAG Step 8**

[think]
Now that 'good' condition should be applied, the next step is to look for the 'search titles only' option. Since there does not appear to be an explicit 'titles only' checkbox in the accessibility tree, the most likely way to proceed is to try the main 'submit' button (element_id 14) to run the search with all entered criteria. Sometimes Craigslist restricts 'titles only' to advanced search, or it is embedded as an option within the main search submission. This step is necessary to see if the results page or subsequent filters will expose the 'search titles only' option, or if search results already reflect it.
[/think]
[action_history]
submit click
[/action_history]

**Non-RAG Step 9**

[think]
Since there still does not appear to be a 'search titles only' checkbox or filter explicitly shown in the accessibility tree, the next step is to check for advanced search options, nearby filters, or contextual help that may reveal or explain this feature. There is a 'search cell phones' input and standard Craigslist UI sometimes places a 'titles only' checkbox near the search inputbut it's not present here. I will investigate the nearby filter buttons (e.g., 'condition', 'mobile OS'), or look for additional options or advanced search links. Clicking 'related searches' (element_id 26) may also reveal advanced filtering options or extra search configurations including titles-only selection.
[/think]
[action_history]
button click
[/action_history]

**Non-RAG Step 10**

[think]
I've checked for 'related searches' and did not see a 'search titles only' option in the accessibility tree. There are no dedicated buttons or checkboxes labeled for searching in titles only near the search input or in filters. The logical next move is to closely examine the displayed search results and see if the search results are already filtered by title (i.e., do they display posts with 'iPhone X' in their titles?). This will confirm if further filtering is necessary or if the system already applies a titles-only filter by default. To proceed, I will scan the search result section for listings, titles, or headers matching 'iPhone X.' If there are no search results, I will document this and reflect that either no matches exist or the query returns no results under the given filters. Browsing the actual results is essential to complete the assessment.
[/think]
[action_history]
Extract the displayed search results and their titles for the 'iPhone X' query under the

specified filters. Determine if any listings are returned and check if 'iPhone X' appears in the titles, confirming if Craigslist's search function matches title-only by default or if no listings match. cache_data
**[/action_history]**

**Analysis**

**1. DOM-RAG-Assisted Agent Performance: The RAG-assisted agent demonstrated a clear, methodical approach, consistently validating each step against a known-good workflow. Its cognitive trace is replete with references to the "RAG example" and "typical Craigslist flow." This external knowledge provided a precise sequence for applying multiple filters: `search term condition apply max price titles only`. Crucially, when the "titles only" option was not immediately visible, the RAG example guided the agent to explore all sidebar and advanced controls, a sophisticated heuristic that a non-guided agent might miss.**

**2. DOM-Non-RAG Agent Performance: The Non-RAG agent attempted a different, less efficient strategy by first navigating to a sub-category (`cell phones`). Its filtering process was disorganized and redundant, for example, filling the "iPhone X" search term multiple times. Most importantly, when it could not find the "search in titles only" checkbox, it lacked a clear strategy. Its final action was to give up on filtering and attempt to infer the answer by manually inspecting the results.**

**3. Role of the RAG System: The RAG system's primary role was to provide a correct and robust procedural script for multi-filter application. It transformed the task from a chaotic exploration of UI elements into a deterministic, sequential process. It also supplied the vital heuristic for locating hidden or advanced options, preventing the agent from prematurely abandoning a crucial task constraint.**

---

RAG vs. Non-RAG 2

**Task name:** Show me the comparison of the first two personal credit cards that do not charge foreign transaction fees.

**RAG Step 1**

**[think]**
To progress towards comparing the first two personal credit cards with no foreign transaction fees, I need to access the list of personal credit cards. The best approach is to click 'View All Credit Cards' under the 'Personal Cards' sectionthis should display a comprehensive card selection, including features relevant to the task.
**[/think]**
**[action_history]**
View All Credit Cards click
**[/action_history]**

**RAG Step 2**

**[think]**
The next required action is to click the 'No Foreign Transaction Fee' button to filter the list of personal credit cards to only those without foreign transaction fees. This aligns with my plan and the example workflow, allowing us to efficiently identify eligible cards for comparison.
**[/think]**
**[action_history]**
No Foreign Transaction Fee filter button click
**[/action_history]**

### RAG Step 3

[think]
Now that I am on the American Express personal credit card selection page, the immediate action is to apply the 'No Foreign Transaction Fee' filter. This will show only those personal credit cards that meet the user's criteria, making it easy to select the first two for comparison in subsequent steps.
[/think]
[action_history]
No Foreign Transaction Fee filter button click
[/action_history]

### RAG Step 4

[think]
Having already identified the 'No Foreign Transaction Fee' filter button, the logical next step is to apply this filter to show only the relevant credit cards. By clicking this filter now, the listing will update to display only personal credit cards that do not charge foreign transaction fees, enabling me to easily select the first two for side-by-side comparison as required by the task.
[/think]
[action_history]
No Foreign Transaction Fee button click
[/action_history]

### RAG Step 5

[think]
The next immediate step is to select the first credit card for comparison by clicking its 'Compare' button. The accessibility tree shows each card listing has a 'Compare' button (button role, label 'Compare'), and the sequence in the list matches the task requirement to select the first two. By clicking the first available 'Compare' button after applying the 'No Foreign Transaction Fee' filter, I will set up for a side-by-side comparison.
[/think]
[action_history]
Compare click
[/action_history]

### RAG Step 6

[think]
To proceed, I need to select the second credit card for comparison by clicking the 'Compare' button for the next card in the filtered list. Having already clicked 'Compare' on the first card (element_id 153), the second card's 'Compare' button is the next in sequence, which corresponds to element_id 169. This step will mark the two cards so I can use the comparison feature to display them side-by-side.
[/think]
[action_history]
button click
[/action_history]

### RAG Step 7

[think]
After clicking 'Compare' for the first card in the list, the next logical step is to select the 'Compare' button for the second card in the filtered results. This will mark both cards for side-by-side comparison. Once both are marked, I will look for a 'Compare Cards' button to generate the comparison. As the click for the first 'Compare' succeeded, I now proceed to the second card in sequence.
[/think]
[action_history]

Compare button for the second card in the filtered list  click
**[/action_history]**

### RAG Step 8

**[think]**

Since clicking the 'Compare' button for the first card resulted in an error, it's possible that the standard multi-step compare workflow may not function as expected. However, the accessibility tree reveals that each card panel has 'View Details' links adjacent to their 'Compare' button. Instead of relying on the compare overlay, I can open both cards' detail pages (in new tabs or windows) by clicking the first two 'View Details' links after applying the 'No Foreign Transaction Fee' filter. This enables me to manually collect and present their fees, rewards, and other features side-by-side, fulfilling the main intent of the comparison. Therefore, the next step is to click 'View Details' for the first card.

**[/think]**
**[action_history]**
View Details  click
**[/action_history]**

### RAG Step 9

**[think]**

The current step is to gather details for the second card in the list of personal credit cards with no foreign transaction fees. The previous step opened the detail page for the first card, so now I need to click the 'View Details' link for the second eligible card (after the first one just viewed). This will allow me to manually compare the two cards' fees, rewards, and benefits.

**[/think]**
**[action_history]**
View Details  click
**[/action_history]**

### Non-RAG Step 1

**[think]**

To proceed with the task, I need to access the section which lists all personal credit cards. The most logical action is to click the 'View all Personal Cards' link, as this should provide a comprehensive view of card options, enabling me to identify those with no foreign transaction fees.

**[/think]**
**[action_history]**
View all Personal Cards  click
**[/action_history]**

### Non-RAG Step 2

**[think]**

I've reached the section showing all personal cards, and there are multiple "Compare Card" buttons. To find cards with no foreign transaction fees, I need to examine the details. The best next step is to cache the feature blurbs or descriptive text next to the first few 'Compare Card' buttons, so I can check for a 'no foreign transaction fees' statement and then select the first two eligible cards for comparison.

**[/think]**
**[action_history]**
Extract all visible card names and their visible features/benefits, especially relating to foreign transaction fees, in the region near the first few 'Compare Card' buttons (element_ids 231, 235, 239, 256). Focus on feature lists or descriptions mentioning 'no foreign transaction fees' or similar wording.  cache_data
**[/action_history]**

### Non-RAG Step 3

[think]
The accessibility tree shows several 'Compare Card' buttons (element_ids 231, 235, 239, 256, etc.) next to 'Learn More' links, likely each for a different credit card. However, the tree does not provide detailed feature text about each card directly. To identify cards with no foreign transaction fees, I need to examine the visible details/benefits associated with each card. The wisest next step is to cache (extract and capture) the feature blurbs or descriptive text next to the first few 'Compare Card' buttons, so I can check for a 'no foreign transaction fees' statement and then select the first two eligible cards for comparison.
[/think]
[action_history]
Extract the descriptive and feature text or blurbs associated with cards having the first few 'Compare Card' buttons (element_ids 231, 235, 239, 256) to identify which explicitly offer no foreign transaction fees. cache_data
[/action_history]

### Non-RAG Step 4

[think]
Having already cached relevant card feature descriptions, the next logical step is to identify the first two personal credit cards with no foreign transaction fees. Then, use the existing 'Compare Card' buttons to initiate a direct side-by-side comparison of those two cards, if the website supports comparing two cards simultaneously. This involves clicking 'Compare Card' for the first qualifying card, then for the second, to trigger the comparison functionality.
[/think]
[action_history]
Compare Card button click
[/action_history]

### Non-RAG Step 5

[think]
Since the previous step involved clicking 'Compare Card' for the first qualifying card, the next logical action is to click 'Compare Card' for the second eligible card (adjacent to element_id 235). This should trigger the built-in comparison view for both cards side-by-side, allowing for direct comparison, which aligns with the user's request.
[/think]
[action_history]
Compare Card click
[/action_history]

### Non-RAG Step 6

[think]
After clicking 'Compare Card' for the first two qualifying cards, the site should present a side-by-side comparison, either as an overlay or as a new section. The most effective next step is to extract and summarize this comparison, highlighting whether both cards have no foreign transaction fees and displaying their key features, thus directly answering the user's request.
[/think]
[action_history]
Extract the contents of the side-by-side comparison table or feature display for the two personal credit cards just selected, with a focus on their foreign transaction fees and main benefits. cache_data
[/action_history]

### Analysis

1. DOM-RAG-Assisted Agent Performance: The RAG-assisted agent demonstrated a clear, methodical approach, consistently validating each step against a known-good workflow. Its

cognitive trace is replete with references to the "RAG example" and "typical Craigslist flow." This external knowledge provided a precise sequence for applying multiple filters: search term condition apply max price titles only. Crucially, when the "titles only" option was not immediately visible, the RAG example guided the agent to explore all sidebar and advanced controls, a sophisticated heuristic that a non-guided agent might miss.

2. DOM-Non-RAG Agent Performance: The Non-RAG agent attempted a different, less efficient strategy by first navigating to a sub-category (cell phones). Its filtering process was disorganized and redundant, for example, filling the "iPhone X" search term multiple times. Most importantly, when it could not find the "search in titles only" checkbox, it lacked a clear strategy. Its final action was to give up on filtering and attempt to infer the answer by manually inspecting the results.

3. Role of the RAG System: The RAG system's primary role was to provide a **correct and robust procedural script for multi-filter application**. It transformed the task from a chaotic exploration of UI elements into a deterministic, sequential process. It also supplied the vital heuristic for locating hidden or advanced options, preventing the agent from prematurely abandoning a crucial task constraint.

---

### RAG vs. Non-RAG 3

**Task name:** Check the specifications of the best-selling HP FHD laptop with 16 GB RAM and core i7 running on Windows 11.

#### RAG Step 1

[think]
To proceed, I need to click the search bar on the Best Buy homepage so I can enter a query for HP FHD laptops. This matches the next logical step in the provided example and aligns with my overall plan.
[/think]
[action_history]
Search Best Buy  click
[/action_history]

#### RAG Step 2

[think]
To advance the task, I need to enter the search query 'HP FHD laptop 16GB RAM Core i7 Windows 11' into the search bar to find relevant products matching all criteria. This matches the next logical action from the similar example and my own planned approach.
[/think]
[action_history]

HP FHD laptop 16GB RAM Core i7 Windows 11  fill_form
[/action_history]

#### RAG Step 3

[think]
To ensure the search results match all target criteria (HP brand, 16GB RAM, Core i7, Windows 11), I need to use the sidebar filters for HP (brand), 16GB RAM, and Intel Core i7 processor. This directly matches the high-level approach from the similar example and ensures the results are not cluttered with non-matching models, maximizing my chance to identify the best-selling product quickly.
[/think]

**[action_history]**
checkbox click
**[/action_history]**

### RAG Step 4

**[think]**
To further refine the search results and meet the task specifications, I need to apply the 16GB RAM and Intel Core i7 processor filters from the sidebar. This ensures only relevant HP laptops with required memory and CPU are shown, minimizing irrelevant results before sorting by best selling. Applying RAM first will remove non-matching options.
**[/think]**
**[action_history]**
checkbox click
**[/action_history]**

### RAG Step 5

**[think]**
The next step is to apply the Intel Core i7 processor filter in the sidebar to further narrow down to only those HP laptops with 16GB RAM and a Core i7, as this matches one of the primary task requirements. This will minimize irrelevant results before I further filter by Windows 11 and look for the best seller.
**[/think]**
**[action_history]**
checkbox click
**[/action_history]**

### RAG Step 6

**[think]**
The next step is to apply the 'Windows' OS filter in the sidebar to ensure the search results only show HP laptops with 16GB RAM, Intel Core i7, and running Windowsmeeting all main requirements. This will further refine the product list before sorting for best selling and reviewing individual specs.
**[/think]**
**[action_history]**
checkbox 'Windows' click
**[/action_history]**

### RAG Step 7

**[think]**
Now that all relevant filters are applied (HP brand, 16GB RAM, Intel Core i7, Windows), I need to sort the search results by 'Best Selling' to easily identify the best-selling HP FHD laptop that matches all the required criteria. The sort button is usually labeled something like 'Best Match' or similar, and this **matches the next step in the example which targets the sort dropdown to select 'Best Selling'.**
**[/think]**
**[action_history]**
button click
**[/action_history]**

### RAG Step 8

**[think]**
After sorting by 'Best Selling', I need to select the first laptop listing in the filtered results, as it should be the top-selling HP FHD laptop with 16GB RAM, Core i7, and Windows 11. This **mirrors the approach from the example** where the product link at the top of the list is chosen after best-selling sorting, enabling review of full specifications on the detail page.
**[/think]**

**[action_history]**
link click
**[/action_history]**

---

### RAG Step 9

[think]
To complete the task, I need to review the 'Specifications' tab on the HP laptop's detail page. This action will enable me to confirm and summarize all required technical details (FHD display, 16GB RAM, Core i7, Windows 11) as in the example, thus fulfilling the original requirement.
[/think]
**[action_history]**
Specifications tab click
**[/action_history]**

---

### Non-RAG Step 1

[think]
To proceed to the U.S. version of Best Buy in English, I should click the 'United States' link under the 'English' section. This will grant access to the main site and allow me to perform targeted searches for HP FHD laptops.
[/think]
**[action_history]**
link 'United States' click
**[/action_history]**

---

### Non-RAG Step 2

[think]
To begin the process of searching for the best-selling HP FHD laptop with 16 GB RAM and core i7 running on Windows 11, I need to use the site's search bar. This will allow me to input a refined query (e.g., 'HP laptop 16GB RAM Core i7 Windows 11 FHD'), after which I can further filter and sort the results.
[/think]
**[action_history]**
HP laptop 16GB RAM Core i7 Windows 11 FHD fill_form
**[/action_history]**

---

### Non-RAG Step 3

[think]
To accurately fulfill the task, I need to ensure that the search results are sorted by Best Selling. The first visible sort option button is labeled 'Best Match', which typically reveals sorting options; I should click it and then choose 'Best Selling' if available. Afterward, I will proceed to visit the link for the top-listed HP laptop for specification details.
[/think]
**[action_history]**
button 'Best Match' click
**[/action_history]**

---

### Non-RAG Step 4

[think]
To fulfill the task, I need to check the detailed specifications for the best-selling HP FHD laptop with 16 GB RAM and Core i7 running Windows 11. The first result - '15.6...'
[/think]
**[action_history]**
link click
**[/action_history]**

---

### Non-RAG Step 5

[think]
To complete the task, I need to check the detailed specifications of the laptop to confirm the operating system is Windows 11 and to verify the other specs. The 'Specifications' button is likely to provide this information. Therefore, I should click the 'Specifications' button to get the relevant details.
[/think]
[action_history]
Specifications  click
[/action_history]

---

### Non-RAG Step 6

[think]
To complete the task, I need to review the full specifications of the best-selling HP FHD laptop with 16GB RAM and Core i7 running Windows 11. The accessibility tree shows a 'Specifications' button (element_id 47) which likely leads to a dedicated section or tab with these details. Since clicking it failed previously due to a selector error, I should manually cache or review the key accessible elements already visible since much of the relevant specifications (screen size, resolution, RAM, CPU, OS) are shown as buttons. I can extract these for the final answer.
[/think]
[action_history]
Processor: Intel 13th Gen Core i7, RAM: 16GB, Display: 15.6-inch Full HD (1920x1080), Storage: 512GB SSD, Graphics: Intel Iris Xe, Operating System: (to confirm, check visible description/specs but expect Windows 11), Product: HP 15.6\\  cache_data
[/action_history]

---

### Analysis

1. DOM-RAG-Assisted Agent Performance: The RAG-assisted agent executed a highly structured and comprehensive plan. It understood that a complex query requires a **multi-step filtering and sorting process**, not just a single search. Its thoughts consistently reference an "example", showing it followed a validated workflow: `search  filter by brand  filter by RAM  filter by processor  filter by OS  sort by best-selling  select top result  view specifications`. This systematic application of constraints ensured the final result was highly accurate and met all task requirements.

2. DOM-Non-RAG Agent Performance: The Non-RAG agent's approach was far more simplistic and ultimately incomplete. It relied entirely on a single, long search query to do all the filtering work. While it did attempt to sort the results, it failed to apply the necessary sidebar filters for RAM, processor, and OS. This means its "best-selling" result was likely incorrect because it was chosen from a pool of items that did not meet all the user's criteria.

3. Role of the RAG System: RAG's crucial function was to provide the **knowledge of a complete, multi-step filtering workflow**. It taught the agent that for complex e-commerce queries, relying solely on the search bar is insufficient. Instead, a robust strategy involves sequentially applying dedicated filters to systematically narrow down the product space before making a final selection. The Non-RAG agent lacked this strategic depth, leading it to a superficially correct but fundamentally flawed result.

Table 18: RAG Sucess Example Step 0-2

**Instruction:** Check drug interaction for melatonin and Folate Forte.

**Observation 0**

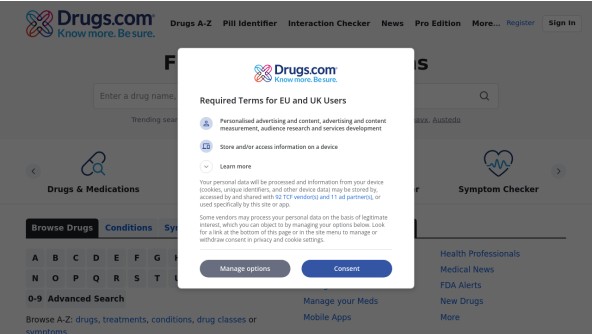

**Action 1**

operator_click(752,576)

**Observation 1**

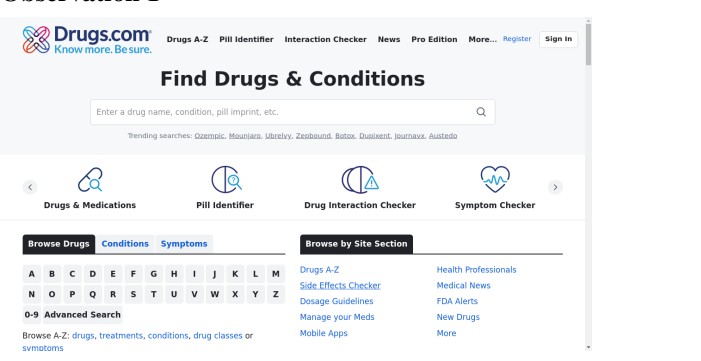

**Action 2**

operator_wait(1000)

**Observation 2**

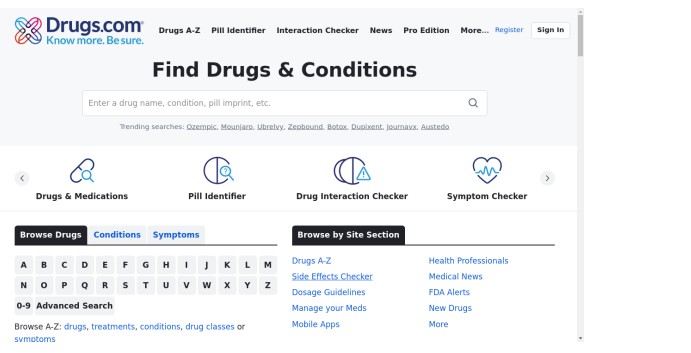

**Action 3**

operator_click(766,369)

## F.4 VISUAL COMPARISON: RAG VS. NON-RAG

Tables 18–24 present screenshot comparisons illustrating the differences with and without RAG, followed by a detailed comparative analysis at the end.

Table 19: RAG Success Example Step 3-5

**Observation 3**

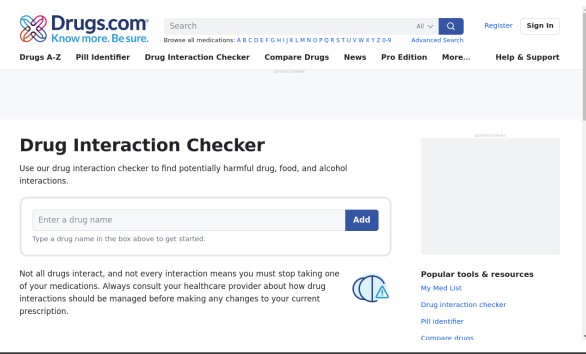

**Action 4**

`operator_wait(1000)`

**Observation 4**

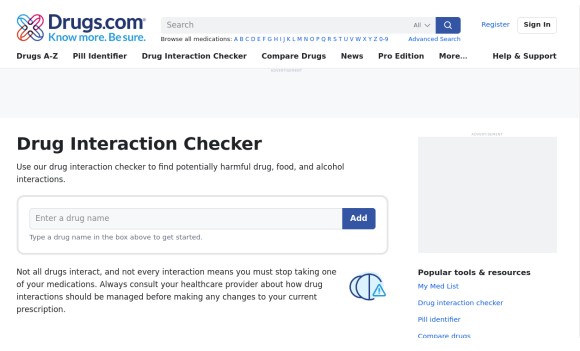

**Action 5**

`operator_click(233,461)`

**Observation 5**

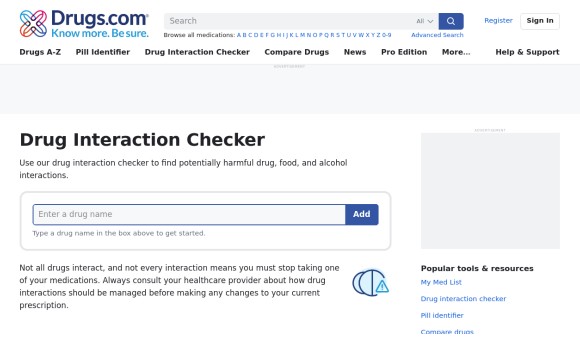

**Action 6**

`operator_type(melatonin)`

Table 20: RAG Success Example Step 6-8

**Observation 6**

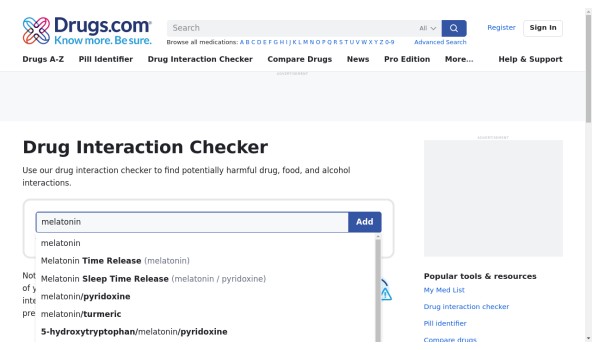

**Action 7**

operator_click(144,504)

**Observation 7**

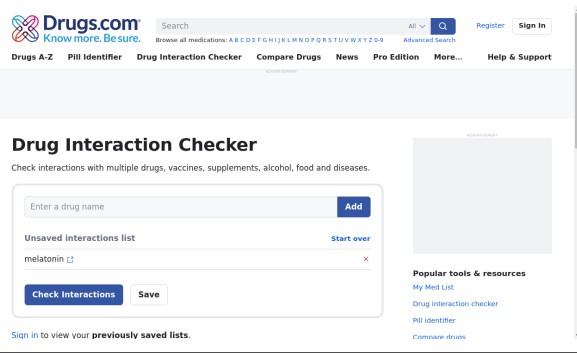

**Action 8**

operator_click(186,430)

**Observation 8**

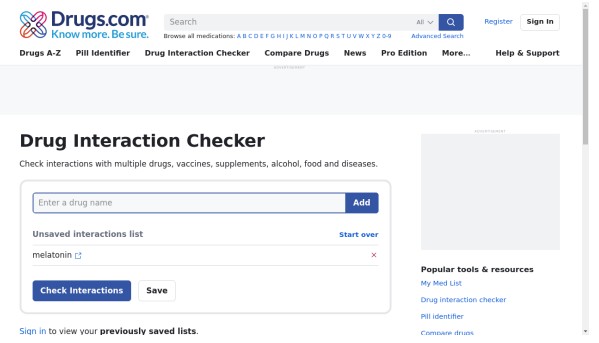

**Action 9**

operator_type(Folate Forte)

Table 21: RAG Success Example Step 9-11

### Observation 9

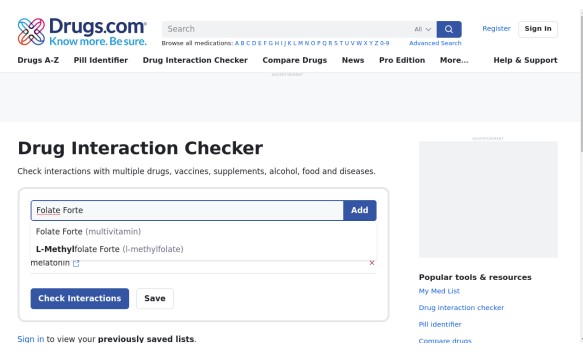

### Action 10

```
operator_click(180,475)
```

### Observation 10

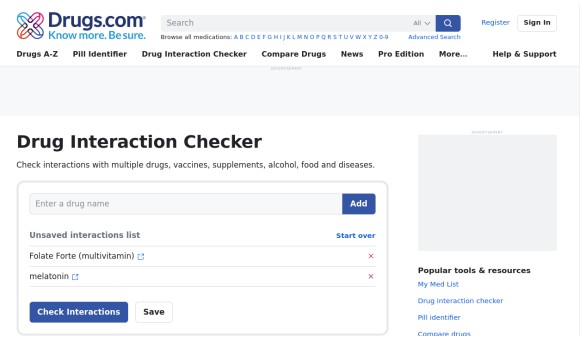

### Action 11

```
operator_click(185,665)
```

### Observation 11

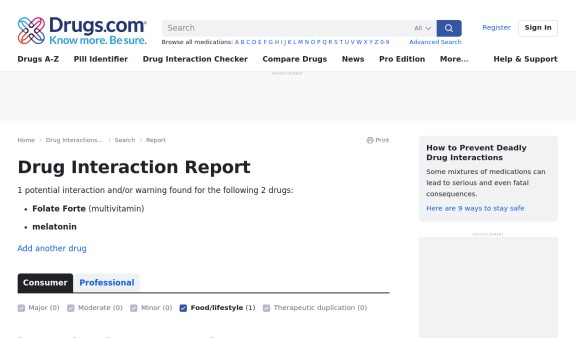

### Action 12

```
operator_wait(1000)
```

Table 22: Non-RAG Failure Example Step 0-2

**Instruction:** Check drug interaction for melatonin and Folate Forte.

**Observation 0**

**Action 1**

`operator_click(819,406)`

**Observation 1**

**Action 2**

`operator_click(164,457)`

**Observation 2**

**Action 3**

`operator_type(melatonin)`

Table 23: Non-RAG Failure Example Step 3-5

**Observation 3**

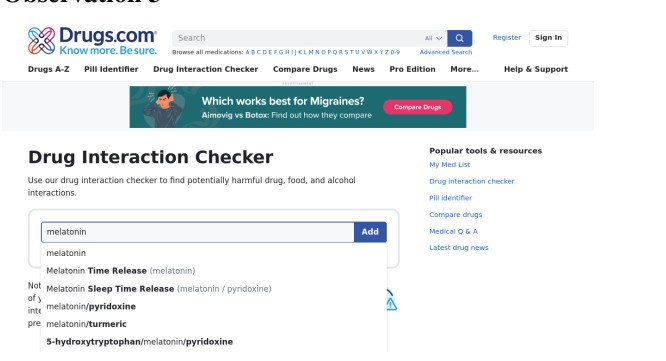

**Action 4**

operator_click(140,509)

**Observation 4**

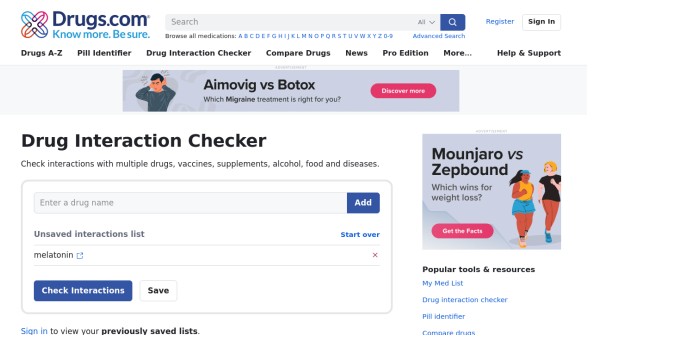

**Action 5**

operator_click(162,436)

**Observation 5**

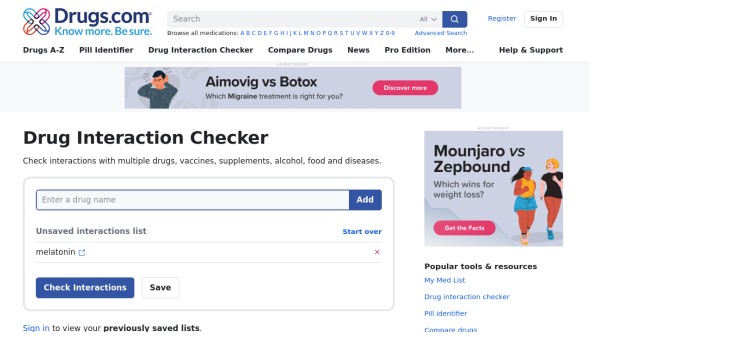

**Action 6**

operator_type(Folate Forte)

Table 24: Non-RAG Failure Example Step 6

**Observation 6**

**Action 7**

`operator_click(228,482)`

---

Analysis of the Task `07ec4a12cba8090e2dc524d558ac7675`

**Task name:** Check drug interaction for melatonin and Folate Forte.

**RAG's Specific Role:**
RAG played a crucial role by visually identifying and confirming the correct input fields and buttons for entering the drug names and initiating the interaction check. It ensured the agent followed the correct sequence of actions by retrieving relevant visual cues from the user interface at each step. This was achieved by retrieving screenshots that validated the correct UI elements were being targeted.

**Advantage Identification:**
The primary advantages were the accurate identification of UI elements, a reduced risk of operational errors, and efficient navigation through the task's interface. RAG prevented errors by ensuring that each click and text input was made on the correct UI element, avoiding potential misclicks or incorrect data entry. This demonstrated a comprehensive understanding of the task interface layout, which the NoRAG agent lacked.

**Critical Steps Analysis:**
- **Step 1: operator_click(752,576):** RAG influenced this step by retrieving a visual confirmation of the correct button to start the drug interaction check process, ensuring the task was initiated correctly from the outset.
- **Step 5: operator_click(233,461):** RAG ensured that the correct input field for the first drug, 'melatonin', was selected by retrieving the visual layout of the input form. This prevented data entry errors by confirming the correct target field.
- **Step 11: operator_click(185,665):** RAG confirmed the correct button to submit the interaction check by retrieving the visual layout of the final confirmation button. This ensured the task was completed successfully, reducing the risk of an incomplete execution.

---

## F.5 QUALITATIVE ANALYSIS ON HARD TASK EXAMPLES

In this subsection, Tables 25–27 present the execution processes of two challenging tasks. We qualitatively explain why retrieved knowledge brings clear gains on such difficult cases. Due to space limitations and the large number of steps, we only analyze screenshots from a subset of key steps.

Table 25: Hard Task Success Example 1 (step 1&2)

**Task description:** Find a condo for rent in Houston, TX, with a monthly rent of no more than 30% of an income of $8000. The condo should have a minimum area of 600 square feet, and the move-in date is the 1st of next month.

**Observation 1**

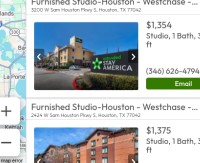

**Thought**

```
To proceed with the task, I need to specifically select 'Houston
Condos for Rent' from the main navigation to narrow the search to
condos in Houston, TX. This matches the example's move to directly
filter by city and property type before applying further filters.
```

**Analysis of retrieved knowledge utilization**

```
It determines the high-level execution logic via retrieved
knowledge and follows a broad-to-narrow search strategy, first
applying filters on geographic location and property type.
```

**Observation 2**

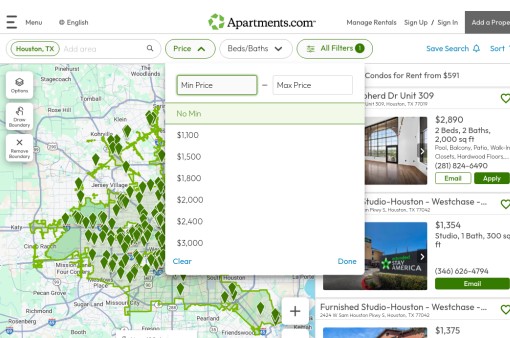

**Thought**

```
The next step is to set the maximum rent to $2,400.  The
accessibility tree shows that under the 'Price' filter, there
are input boxes labeled 'Max Price'.  Following the example, the
typical flow is to click the price filter, enter the cap, and
confirm.  This aligns with the task's next constraint and the
example's approach to applying budget criteria after location and
property type.
```

**Analysis of retrieved knowledge utilization**

```
The next high-priority condition, guided by the retrieved
knowledge, is the budget.  This ensures that the filtering process
remains hierarchical and well-structured.
```

Table 26: Hard Task Success Example 1 (step 19&20)

**Observation 1**

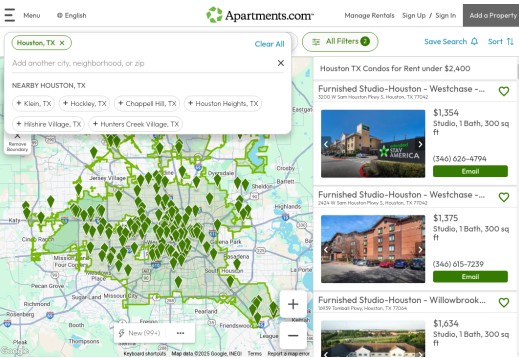

**Thought**

```
Since all direct attempts to set the minimum area filter in the
main and secondary dropdowns did not update the listings (they
still show units <600 sq ft), my next step is to reset all filters
by clicking the 'Clear All' button, then re-apply every constraint
from scratch via the 'All Filters' link. This systematic reset
(as outlined in my plan and corroborated by the RAG example)
maximizes the chance of enforcing all constraints, particularly
the 600 sq ft minimum, then reviewing listings for a true match.
```

**Analysis of retrieved knowledge utilization**

```
When the task encounters a failure where the filter stops working
partway through, the retrieved knowledge makes it reset all
filters instead of continuing trial and error. This enables the
model to take decisive initiative in error correction and the
recovery stage.
```

**Observation 2**

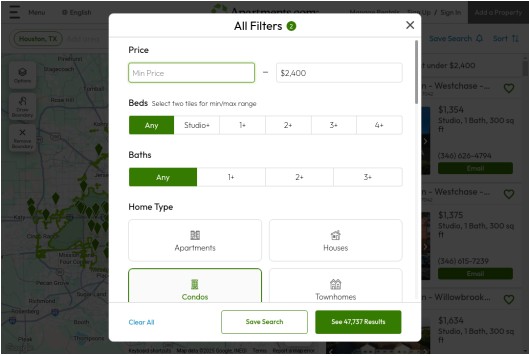

**Thought**

```
After clearing all filters, my next move is to open the 'All
Filters' panel. From there, I can comprehensively reapply
every necessary constraint: setting the maximum rent to $2,400,
selecting 'Condo', establishing a minimum area of 600 sq ft, and
designating the move-in date as the 1st of next month. This
flows directly both from the troubleshooting plan and the similar
example, which confirms that a full reset followed by methodical
use of the consolidated filters panel is the best recovery when
filters are not operating properly.
```

**Analysis of retrieved knowledge utilization**

```
Retrieved knowledge directs the model to use the integrated filter
panel for rescreening instead of applying filters piecemeal.
```

Table 27: Hard Task Success Example 2 (step 3&7)

**Task description:** Check the specifications of the best-selling HP FHD laptop with 16 GB RAM and core i7 running on Windows 11.

**Observation 1**

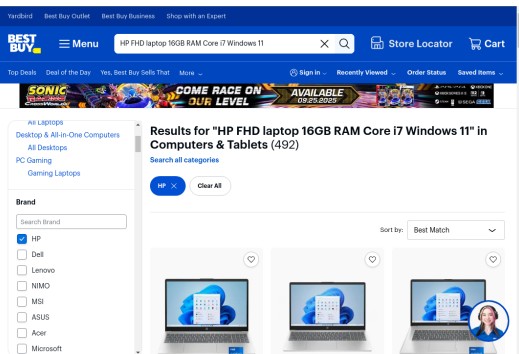

**Thought**

```
Now that all relevant filters are applied (HP brand, 16GB RAM,
Intel Core i7, Windows), I need to sort the search results by
'Best Selling' to easily identify the best-selling HP FHD laptop
that matches all the required criteria.  The sort button is
usually labeled something like 'Best Match' or similar, and
this matches the next step in the example which targets the sort
dropdown to select 'Best Selling'.
```

**Analysis of retrieved knowledge utilization**

```
WebRAGent uses the best selling filter based on retrieved
knowledge to ensure the answer is correct.  In contrast, the non
RAG agent skips this filtering step.
```

**Observation 2**

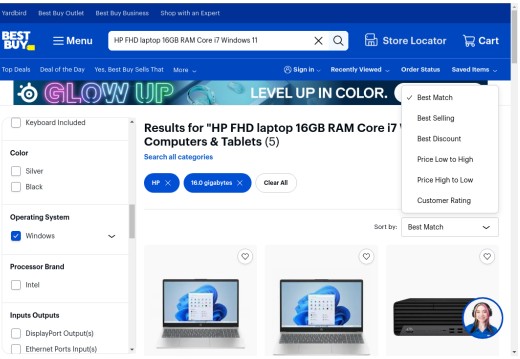

**Thought**

```
The next step is to set the maximum rent to $2,400.  The
accessibility tree shows that under the 'Price' filter, there
are input boxes labeled 'Max Price'.  Following the example, the
typical flow is to click the price filter, enter the cap, and
confirm.  This aligns with the task's next constraint and the
example's approach to applying budget criteria after location and
property type.
```

**Analysis of retrieved knowledge utilization**

```
The next high-priority condition, guided by the retrieved
knowledge, is the budget.  This ensures that the filtering process
remains hierarchical and well-structured.
```

**SNIPPET 7: TEXT GUIDANCE**

**task name**

Find discussions of the community and open one with the most replies on flightaware.

**website**

https://www.flightaware.com/

**steps 1**

"observation_description": "The image displays the homepage of the FlightAware website as viewed in a web browser. The browser's tab shows the title \"FlightAware – Flight Tracker,\" and the URL in the address bar is \"flightaware.com.\" The page features some promotional content about global flight tracking and other FlightAware services, including fleet tracking and a section about unlocking real–time flight tracking data. There are images of aircraft (an airplane wing in–flight and a jet on the ground). At the bottom, there is a cookie consent banner with options to \"Allow All\" or adjust \"Cookie Settings.\" There are also navigational links such as \"See all products\" and \"Compare tiers.\"",

"action_description": "The action being taken is launching or opening the FlightAware website by entering its URL (\"https://www.flightaware.com/\") in a web browser. This initiates access to the homepage, preparing the user to start navigating the site to continue with the next steps of the task."

**steps 2**

"observation_description": "The current screen displays the main page of the FlightAware website. The visible content includes a promotional banner at the top right advertising real–time global flight tracking and ADS–B flight data, along with a call–to–action link to learn more about FlightAware Firehose. There's also a prominent image of a jet on a runway with a headline about secure, private fleet tracking and another call–to–action to compare tiers. At the bottom, there is a visible cookie consent banner with options for \"Allow All\" and \"Cookie Settings.\"",

"action_description": "The action being performed is a scroll operation, moving the webpage downward by 2,337 pixels. This scrolling action is likely intended to navigate past the current promotional content and cookie banner to reveal additional sections of the page\u2014potentially aiming to locate links or sections related to community discussions or forums on FlightAware.",

**steps 3**

"observation_description": "The image displays the footer section of the FlightAware website. In this section, there are various categories such as \"ABOUT,\" \"COMMUNITY,\" \"PRODUCTS,\" and \"CUSTOMER SERVICE.\" Under the \"COMMUNITY\" category, several links are listed, including \"Squawks,\" \"Discussion,\" \"Photos,\" and \"ADS–B Statistics.\" The \"Discussion\" link is visually highlighted, indicating that it is being hovered over or is the intended target for the next action.",

"action_description": "The action being taken is a user clicking on the \"Discussion\" link under the \"COMMUNITY\" section of the website's footer. This action is intended to navigate to the community discussions area of the FlightAware website, where users can view and participate in forum–style discussions.",

**steps 4**

"observation_description": "The screenshot displays the \"FlightAware Discussions\" page, which appears to be a forum or discussion board for the FlightAware community. The interface shows several navigation tabs at the top, including \"categories,\" \"Categories\" (selected), and \"Latest.\" Each category (such as \"FlightAware\" and \"Aviation\") is outlined with subcategories listed below, and to the right, a list of recent discussion threads is visible, showing thread titles, categories, and the number of replies for each (e.g., \"Welcome to the new and improved Discourse!\" with 52 replies, \"Running BananaPI M2 Zero\" with 102 replies, and \"HOWTO: Airspy mini and Airspy R2: Piaware / dump1090–fa configuration\" with 4.5k replies). One specific area of the interface\u2014the \"Latest\" tab\u2014is visually highlighted with a red box, indicating focus.",

"action_description": "The action being taken is a click operation on the \"Latest\" tab (the tab in the top navigation menu just right of \"Categories\" and specifically marked with a red box). This click will update the view to show the most recent discussion threads in the community, likely ordered by most recent activity instead of by category or another grouping. This will help the user find the most active or current discussions on the FlightAware forum."

**steps 5**

"observation_description": "The screenshot displays the \"FlightAware Discussions\" page, specifically showing the \"Top\" discussions for the year range May 11, 2024 \u2013 May 11, 2025. The list is organized in a table with columns for Topic, Replies, Views, and Activity. Each row represents a discussion topic, accompanied by its category, participant avatars, number of replies, views, and the date of last activity. A red rectangle highlights the \"Replies\" column header, indicating that it is being interacted with or selected.",

"action_description": "The action being taken is a click on the \"Replies\" column header. This interaction is likely intended to sort the discussion topics by the number of replies, either in ascending or descending order. Sorting by replies will make it easier to identify the discussion with the most replies, which is relevant to the task of selecting the most active or engaged community discussion."

SNIPPET 8: VISUAL GUIDANCE

**task name**

Find the list of neighborhood maps for Brooklyn on new.mta.info.

**website**

https://new.mta.info/

**Action Space**

1. click: Clicks on the target element (bounding box).\n2. drag: Drags the target element (bounding box) to a new position.\n3. launchApp: Launches a new URL/app with the specified value (URL).\n4. scroll: Scrolls the page by the specified value (pixels) in the given direction.\nPositions are represented in relative coordinates within the range [0,1] on the observation screenshot.

**steps 1**

Observation 1: <|image_1|>\n
Action 1: {\"operation\": \"launchApp\", \"value\": \"https://www.mta.info/\", \"target\": null}\n
"cand_image_path": "['Online−Mind2Web/4091bdd3fa64a5b0d912bc08eaf9c824_step_0.png']

**steps 2**

Observation 2: <|image_1|>\n
Action 2: {\"operation\": \"click\", \"value\": null, \"target\": {\"x\": 0.44659, \"y\": 0.0, \"width\": 0.04845, \"height\": 0.084656}}",
"cand_image_path": "['Online−Mind2Web/4091bdd3fa64a5b0d912bc08eaf9c824_step_1.png']

**steps 3**

Observation 3: <|image_1|>\n
Action 3: {\"operation\": \"scroll\", \"value\": 1200, \"target\": null}",
"cand_image_path": "['Online−Mind2Web/4091bdd3fa64a5b0d912bc08eaf9c824_step_2.png']

**steps 4**

Observation 4: <|image_1|>\n
Action 4: {\"operation\": \"click\", \"value\": null, \"target\": {\"x\": 0.174728, \"y\": 0.809524, \"width\": 0.105969, \"height\": 0.025397}}",
"cand_image_path": "['Online−Mind2Web/4091bdd3fa64a5b0d912bc08eaf9c824_step_3.png'

**steps 5**

Observation 5: <|image_1|>\n
Action 5: {\"operation\": \"click\", \"value\": null, \"target\": {\"x\": 0.4036, \"y\": 0.326587, \"width\": 0.059051, \"height\": 0.02328}}",
"cand_image_path": "['Online−Mind2Web/4091bdd3fa64a5b0d912bc08eaf9c824_step_4.png'

**steps 6**

Observation 6: <|image_1|>\n
Action 6: {\"operation\": \"click\", \"value\": null, \"target\": {\"x\": 0.174728, \"y\": 0.308598, \"width\": 0.640327, \"height\": 3.518519}}",
"cand_image_path": "['Online−Mind2Web/4091bdd3fa64a5b0d912bc08eaf9c824_step_5.png'

**steps 7**

Observation 7: <|image_1|>\n
Action 7: {\"operation\": \"drag\", \"value\": null, \"target\": {\"x\": 0.0, \"y\": 0.084656, \"width\": 0.989782, \"height\": 3.835582}}",
"cand_image_path": "['Online−Mind2Web/4091bdd3fa64a5b0d912bc08eaf9c824_step_6.png']",

