# OpenReview forum: "WebRAGent: Retrieval-Augmented Generation for Multimodal Web Agent Planning"
_ICLR.cc/2026/Conference — Submitted to ICLR 2026_

### Official Review · Reviewer_bk79 · 2025-10-31

**Soundness:** 3
**Presentation:** 2
**Contribution:** 3
**Rating:** 4
**Confidence:** 4

**Summary:**

The paper presents WebRAGent, a retrieval‑augmented multimodal web agent designed to leverage past GUI trajectories for better decision‑making. The authors introduce the Unified Agent Trajectory Dataset (UATD) and propose the new task of Multimodal Trajectory Retrieval, releasing the benchmarks GAE‑Bench and GAE‑Bench‑lite with over 700K trajectory retrieval pairs. They develop GAE‑Retriever, a VLM2Vec‑based model using token selection and GradCache for efficient contrastive training, and integrate it into the WebRAGent framework. Experiments across five datasets show substantial recall improvements over strong multimodal baselines, and on Online‑Mind2Web, WebRAGent achieves 15–22% higher success rates than non‑retrieval models.

**Strengths:**

-  **Innovative dataset construction and benchmarks.**
The paper introduces a unified methodology for integrating heterogeneous GUI‑based trajectory data, resulting in the Unified Agent Trajectory Dataset (UATD) and two large‑scale benchmarks (GAE‑Bench and GAE‑Bench‑lite). This contributes valuable resources for evaluating multimodal trajectory retrieval and provides a standardized foundation for future agent studies.

-  **Novel and well‑structured web‑agent framework.**
The proposed WebRAGent framework innovatively integrates multimodal retrieval with generation in agent planning. Its modular design—combining observation, retrieval, memory, and planning—demonstrates a clear architectural innovation and effectively bridges trajectory -

- **Comprehensive and substantial work.**
The paper presents extensive data preparation, thorough model development, and large‑scale experiments across multiple benchmarks. The amount of work is significant, covering dataset unification, retriever training, and online evaluation, showing strong technical depth and implementation effort.

**Weaknesses:**

- **Lack of clarity in technical details.**
Several key components are insufficiently explained, such as the reward design, data annotation procedures, and implementation specifics of dataset generation and evaluation. These omissions make it difficult to reproduce and precisely understand how the framework works. (See detailed questions in the Questions section.)
- **Unclear core contribution.**
The proposed GAE‑Retriever primarily builds upon existing methods like VLM2Vec and integrates known techniques such as Token Selection and GradCache. Since these components are not original, the novelty of the contribution is uncertain. If the main innovation lies in the integration, the authors should provide clear ablation studies or quantitative evidence demonstrating the necessity and contribution of each module.
- **Onfair performance comparison.**
The retriever is trained and evaluated on data from the same source, which naturally favors high recall scores. Comparisons with untrained or zero‑shot models are therefore not entirely fair. Moreover, the paper does not compare WebRAGent’s web‑retrieval capability against existing Web search or Web‑retrieval models, making it difficult to know whether the performance is better than the existed systems or not.
- **Formatting and presentation issues.**
Figures and tables are sometimes awkwardly placed, often disrupting the reading flow. Aligning them consistently—at the top of pages—would significantly improve the presentation quality.

**Questions:**

- **Reward mechanism ambiguity.**
The paper mentions the use of an “LLM‑as‑judge” strategy for rewarding but provides no implementation details. Which specific LLM model was used for judging (e.g., GPT‑4, GPT‑4‑turbo, or others)? What were the prompts, scoring criteria, and calibration procedures? Given that the reward signal directly affects policy and evaluation, this should be clarified in detail.
- **Unspecified reranking model.**
The framework claims to apply an LLM‑based reranking step after retrieval, yet there is no description of the rerank model, prompt design, or how it integrates with GAE‑Retriever. How is the reranker implemented, and what quantitative performance gain does it contribute?
- **Insufficient transparency in data construction.**
At line 377 the paper states “Data are annotated with gpt‑4o‑mini‑2024‑07‑18.” but does not explain the detailed annotation process. What prompts were used for labeling? How were data quality and potential data leakage from pretrained sources verified?
- **Lack of comparison with other web‑search models.**
WebRAGent’s web‑retrieval performance is not compared with existing Web search or Web‑retrieval systems, such as dense retrievers or LLM‑based search agents. Without such baselines, it remains unclear whether the model actually advances the state of web‑scale retrieval.
- **Computational cost of retrieval augmentation.**
The paper does not quantify how much additional latency or computation the retrieval process introduces during inference. How large is the retrieval database, and what is the average time‑to‑action compared to the non‑retrieval baseline?
- **Fairness of planning models across baselines.**
In the Planning and Action section, the authors state that WebRAGent uses GPT‑4.1 for DOM mode and OpenAI’s computer‑use‑preview model for Vision mode—both very strong, closed‑source models. Do the non‑retrieval baselines employ exactly the same planners? If not, how can we separate the performance gain attributable to retrieval augmentation from that potentially caused by stronger planning models?

---

> ### Author Response · Authors · 2025-11-28
> **Response to Reviewer bk79 (1/3)**
>
> Thank you for acknowledging the innovative dataset construction, novel framework, and comprehensive experiments in our work. We are committed to addressing the raised concerns and questions in our responses as follows:
>
> > W1 & Q1. Response to the comment on implementation details and reward mechanism
> We provide most of the implementation details in the appendix, along with further explanations, as outlined below:
> * Dataset construction & annotation pipeline: We describe the data sources, cleaning steps, the use of `gpt-4o-mini` to generate HTML or textual descriptions, and the silver generation pipeline in **Appendix B.1 / B.2 (page 15-16)** and **Appendix E.1(page 25-28)**.
> * Dataset generation & retrieval pair construction: In **Section 3.2 / Table 8 / Snippets 1–3**, we provide detailed descriptions of the universal trajectory representation, the action JSON schema, the 12 retrieval patterns, and how we construct the input sequences.
> * Reward mechanism design:  We give a detailed explanation of the reward mechanism in **Appendix C.3 (page 20)**, and provide the exact reward prompt in **Snippet 5 (page 30)**. The reward model jointly uses the historical trajectory (including *thought*, *action*, and *reflection*), the DOM tree or screenshot information, and employs `gpt-4o` to assign scores. The model first determines the current task phase and classifies the state into one of three categories: *doing* (on the way to completion), *finished* (task completed), and *loop* (last two actions repeated). Based on this categorization, it selects a score from the discrete set $\\{1, 3, 7, 9, 10\\}$:
>   - 1 indicates a clear loop is detected;
>   - 3 indicates the current trajectory is heading in the wrong direction and needs adjustment or restart;
>   - 7 indicates meaningful progress has been made;
>   - 9 indicates a very important intermediate step has just been completed;
>   - 10 indicates the task has been successfully completed.
>
> > W2. Response to the comment on improvements over VLM2Vec and training strategies
>
> We recognize that our work builds upon VLM2Vec, and our main contribution lies in extending it to multimodal agent trajectory contrastive learning, which is not covered in the original VLM2Vec framework. We further enhance training with *GradCache* and *Token Selection*, enabling efficient training with large in-batch negatives and allowing VLMs to encode multi-step, high-definition multimodal trajectories. To the best of our knowledge, this is the first application of this architecture to **multimodal trajectory retrieval**, and we show that it outperforms existing general-purpose retrieval models.
>
> Concretely, our training setup must handle a large number of high-resolution GUI screenshots, long texts, DOM trees, and action sequences while performing contrastive learning. Without the *Token Selection* and *GradCache* strategies, the model cannot be effectively trained, as our optimization ensures that multimodal trajectory representation learning remains feasible on existing computing devices (e.g., multiple GPUs with 80GB memory). Running ablations only under a heavily downscaled experimental setting, such as downsampling screenshots, substantially reducing in-batch negatives, or decreasing the number of states in a trajectory, would not faithfully model the agent trajectory retrieval problem we aim to solve. This is why we did not include such ablations in the current experiments. We will clarify these points in more detail in the revised version of the paper.

---

> > ### Author Response · Authors · 2025-11-28
> > **Response to Reviewer bk79 (2/3)**
> >
> > > W3 & Q4. Response to the comment on fairness of evaluation and Web search baselines
> >
> > First, our evaluation is not limited to in-domain settings. In the appendix, **Table 10 (Recall@1/5/10 for all methods on each subtask in the out-of-domain setting)** and **Figure 7 (per-task evaluation results; Recall@5 for five selected models across various retrieval tasks and datasets under in-domain (ind) and out-of-domain (ood) scenarios)** also report comprehensive out-of-domain (OOD) results. Under the OOD setting, GAE-Retriever still maintains a substantial performance margin over all baselines, indicating that the model learns *generalizable trajectory representations* rather than simply overfitting to the in-distribution data. Furthermore, our baseline models include systems that have been finetuned on GUI datasets (e.g., UI-TARS-2B-SFT, ShowUI-2B), not only zero-shot models, ensuring that the comparison is fair and meaningful.
> >
> > Second, we would like to clarify that *web search/web retrieval* and our *agent trajectory retrieval* are **entirely different tasks**. Our focus is trajectory retrieval, where the retrieval targets are not web documents but components of complete GUI trajectories consisting of multiple screenshots as states and textual action representations. Traditional web search performs document-level retrieval at the URL or page level, whereas our task involves multimodal trajectory retrieval at the step or action level.
> >
> > For this reason, we select the most relevant multimodal retrieval models, such as ColQwen2, GME, VLM2Vec, as our primary baselines, since they represent the current state of the art in multimodal information retrieval. Additionally, for text-only methods, **Appendix D.5 (Table 11)** compares text-only retrieval ($q \rightarrow \tau$) with multimodal retrieval, and the results show that the multimodal setting performs substantially better (by approximately 22%).
> >
> > > W4. Responses to formatting and layout issues
> >
> > Thank you for pointing out the issues in formatting and typesetting. We will unify and correct them in the next version.
> >
> > > Q2. Response regarding the explanation of reranking implementation details
> >
> > Our reranking model is based on `gpt-4o`. We provide part of the implementation details in **Appendix C.3 (page 20)** and show the exact LLM-reranking prompt in **Snippet 6 (page 31)**. Concretely, GAE-Retriever first performs embedding retrieval over the trajectory database and returns a set of candidate steps (including the candidate ID, task description, and corresponding screenshot). The reranking model then further refines these Top-k candidates. In our prompts, we explicitly instruct LLMs to rank based on (1) visual similarity of the screenshot, (2) consistency of UI elements and layout, and (3) semantic relevance of the task description.
> >
> > In our experiments on Online-Mind2Web, setting Top-k to 5 improves R@1 by around 36%, and setting Top-k to 10 improves R@1 by around 54%. We also manually examined other retrieved examples that do not correspond exactly to the current state and found that they are still highly relevant and provide important guidance for the task. In addition, we applied several engineering optimizations to image compression to reduce token consumption as much as possible.
> >
> > > Q3. Insufficient transparency in data construction
> >
> > We provide the details of the Unified Agent Trajectory Dataset (UATD) and the GAE-Bench series dataset construction in **Appendix B.1 and B.2**, along with the corresponding prompts in **Appendix E.1 (Table 14: Data annotation prompts; Snippet 1: Action Space; Snippet 2 and 3: Instruction Templates)**. For UATD, our annotation process involves generating image captions for each screenshot and rendering HTML as images when the original dataset does not include screenshots. For GAE-Bench, we use LLMs to generate silver tasks for $q \rightarrow \tau_{\sim}$. Since our data annotation primarily intends to augment existing GUI datasets with complete state representations using available information, employing LLMs to complement each state observation does not reduce the complexity of multimodal trajectory retrieval or lessen the difficulty of the downstream task in any way. We will make it more clear in the revised version.

---

> > > ### Author Response · Authors · 2025-11-28
> > > **Response to Reviewer bk79 (3/3)**
> > >
> > > > Q5. Response to the comment on inference latency
> > >
> > > Thanks for the question about inference efficiency. Our retrieval database is built on the UATD dataset. Each planning step in the WebRAGent framework entails an additional round of encoding with the GAE Retriever, followed by embedding vector retrieval and LLM reranking. Compared with the baseline, this does introduce some extra computation, but the overhead is very small relative to the entire task execution: it results in roughly 10% additional latency per step, which we consider fully acceptable in practice. We have applied several engineering optimizations to the retrieval and LLM reranking stages, including a two level cache on the LLM side and optimized and compressed image encoding. Our efficiency experiments are mainly conducted on an NVIDIA RTX A6000 GPU. The results show that the dominant time cost does not come from the GAE Retriever retrieval stage, but from the LLM reranking, while the overall time per step remains modest. When the cache is hit, most queries require almost no extra reranking time and the corresponding inference cost is nearly negligible. These experimental results have been added to **Appendix D.7 (page 24)** of the revised manuscript.
> > >
> > > | Setting              |  Retrieval(s/step) | Reranking(s/step) | Total(s/step) |    Speedup |
> > > |----------------------|------------------------------------------|-----------------------------------|-------------------------|--------------------|
> > > | No Cache             | 0.3877                                   | 3.5374                            | 3.9251                  | 1.0×               |
> > > | With Cache           | 0.3828                                   | 0.0002                            | 0.3830                  | 9.24×              |
> > >
> > >
> > >
> > > > Q6. Response to the comment on fairness of planning models across baselines
> > >
> > > First, we would like to clarify that the non-retrieval baseline and our WebRAGent use exactly the same planning models (`GPT-4.1`/ `computer-use-preview`) in both modes. They also share the same observation space, action space, reward module, and prompts; the only difference is the presence or absence of the retrieval module. Therefore, the comparison in our experiments is fully fair.
> > >
> > > To further demonstrate that the performance gains come from the retrieval module rather than from stronger planners, we integrate GAE-Retriever as a plug-and-play component into **SeeAct** under the same experimental setting, in order to validate the effectiveness of trajectory retrieval within a different agent framework. The results show that, even without sophisticated prompt tuning, adding our retrieval module improves the original SeeAct’s success rate by about **8-10%**, with gains primarily on Medium or Hard tasks. This indicates that introducing multimodal trajectory retrieval is both effective and transferable across agents. These experiments have been added to **Appendix D.6 (page 24)**.
> > >
> > > | Method                        | Easy (%) | Medium (%) | Hard (%) | Success Rate (%) |
> > > |:-----------------------------|:--------:|:----------:|:--------:|:----------------:|
> > > | SeeAct                       |  71.42   |   47.36    |  6.25    |       49.00      |
> > > | + $\mathcal{D}_\text{text}$  |  78.57   |   55.26    |  18.75   |       57.00      |
> > > | + $\mathcal{D}_\text{vision}$|  76.19   |   60.52    |  25.00   |       59.00      |

---

### Official Review · Reviewer_qtrK · 2025-11-01

**Soundness:** 3
**Presentation:** 2
**Contribution:** 3
**Rating:** 4
**Confidence:** 4

**Summary:**

This paper addresses the question of how to retrieve the most relevant parts of past multimodal trajectories to support planning. Part of the motivation is that storing all trajectories with multimodal contents in the context is impractical. The paper constructs a trajectory retrieval corpus called Unified Agent Trajectory Dataset (UATD) from annotated demonstrations and states across diverse real-world scenarios. Building on this, it constructs GAE-Bench, a benchmark containing a large number of trajectory-based retrieval pairs.

Further, the paper proposes GAE-Retriever, a retriever for multimodal trajectories that uses token selection and GradCache to optimize
the contrastive objective. It also introduces WebRAGent, a retrieval-augmented web agent that integrates GAERetriever. Experiments are performed on the Online-Mind2web benchmark.

**Strengths:**

1. The core idea of retrieving similar trajectories for reuse is interesting and intuitive.
2. The GAE-Bench benchmark introduced in this paper for trajectory retrieval is a valuable resource with several patterns of trajectory retrieval, such as text-to-state, text-to-trajectory, state-to-state, etc.
3. Empirical results show a significant boost in performance over non-retrieval baselines on the Online-M2W benchmark.

**Weaknesses:**

1. The Unified Agent Trajectory Dataset introduced in this paper is really not novel. Similar aggregated trajectory dataset already exists [1]
2. For the Online-Mind2web results, rather than choose a simple MLLM baseline, the authors should add trajectory retrieval to exising SOTA or close to SOTA model, e.g. SeeAct [2] or AgentE [3].
3. The authors selected a subset of 100 tasks from Online-M2W without justification for not using the original dataset.

[1] Xu, Yiheng, et al. "Aguvis: Unified pure vision agents for autonomous gui interaction." ICML'25.

[2] Zheng, Boyuan, et al. "Gpt-4v (ision) is a generalist web agent, if grounded." ICML'24.

[3] Abuelsaad, Tamer, et al. "Agent-e: From autonomous web navigation to foundational design principles in agentic systems." arXiv preprint arXiv:2407.13032 (2024).

**Questions:**

N.A.

---

> ### Author Response · Authors · 2025-11-22
> **Response to Reviewer qtrK (1/2)**
>
> > W1. The Unified Agent Trajectory Dataset introduced in this paper is really not novel. Similar aggregated trajectory dataset already exists [1]
>
> A1:Thanks for your valuable suggestions. We acknowledge that Aguvis and our work share a similar high level idea of aggregating trajectory datasets, but our research focus and data sources are fundamentally different.
>
> From the perspective of how multimodal trajectory data are utilized, Aguvis trains an end-to-end vision only GUI agent on multimodal trajectories, mainly focusing on using these supervised data to improve grounding and planning capability. In contrast, we focus on training an embedding model that explicitly represents complex multimodal trajectories in order to enable multimodal trajectory retrieval. To the best of our knowledge, our work is the first to use GUI trajectories to train a multimodal trajectory retriever.
>
> From the perspective of data format and source, Aguvis is designed for action decision learning, so each trajectory sample is organized in the form "current context → predicted next action." Its data come from MM-Mind2Web, AndroidControl, GUI Odyssey, AMEX, and related sources, with a focus on diverse GUI layouts and local interaction patterns. In contrast, our work is centered on multimodal trajectory retrieval: we unify data from Mind2Web, WebArena, WebLINX, GUIAct, and AutoWebGLM into a single MDP style trajectory repository (UATD), which emphasizes the complete execution history of the agent. Our proposed GAE-Bench further decomposes each trajectory into various retrieval example pairs through 12 extraction schemes. As a result, our supervision is no longer "next action" labels, but positive and negative retrieval pairs of the form query → (state / trajectory / interval), which is more consistent with the problem setting of retrieval augmented web agents. Moreover, to the best of our knowledge, we are the first to provide quantitative evaluation criteria that simultaneously capture fine grained temporal structure and coarse grained semantic structure.
>
> In addition, there are further differences in our treatment of unified MDP representation, trajectory content, observations, and the unification of the action space. For observations, although we also, inspired by Aguvis, adopt screenshots rather than DOM trees as the unified visual observation format (line 160), we additionally attach structured information to each state, including natural language descriptions, task descriptions, and sub goals. For the unified action space, we abstract various actions from different benchmarks (clicks, selections, form filling, scrolling, and so on) into a single schema $a_i =( \{\mathrm{operation}, \mathrm{target}, \mathrm{value}\})$, which is decoupled from the concrete execution API. This design is better suited for aligning action semantics across datasets and environments, and it facilitates retrieval and reuse.
>
> [1] Xu, Yiheng, et al. "Aguvis: Unified pure vision agents for autonomous gui interaction." ICML'25.

---

> > ### Author Response · Authors · 2025-11-22
> > **Response to Reviewer qtrK (2/2）**
> >
> > > W2. For the Online-Mind2web results, rather than choose a simple MLLM baseline, the authors should add trajectory retrieval to exising SOTA or close to SOTA model, e.g. SeeAct [2] or AgentE [3].
> >
> > A2: First, we would like to clarify that our baseline is not a simple MLLM, but a strong and complete web agent framework.  We use state-of-the-art models such as GPT-4.1 and computer-use-preview model as the planner.  The agent performs long-horizon task decomposition and execution via chain-of-thought reasoning, explicit history memory, and reward, rather than single-turn MLLM calls.  We also implemented several engineering optimizations.  The only difference between the baseline and WebRAGent is whether they use the knowledge retrieved by GAE-Retriever.
> >
> > To further validate the effectiveness of trajectory retrieval across different agent frameworks, we integrate GAE-Retriever as a plug-and-play module into SeeAct under the same experimental setup.  As shown in Table 12, even without extensive prompt tuning, introducing trajectory knowledge consistently improves task performance, yielding an 8–10% gain in success rate, with particularly notable improvements on medium and hard tasks.  Comparing text-based and vision-based retrieval, we find that visual knowledge performs better on Medium/Hard settings, suggesting that visual exemplars are more helpful for fine-grained operations in complex interactive scenarios.  These results demonstrate that our method is highly extensible across different agent frameworks.  We have added these experiments and analyses to Appendix D.6 of the revised manuscript.
> >
> > | Method                        |  Easy (%) |  Medium (%) |  Hard (%) |  Success Rate (%) |
> > |:-----------------------------|:--------:|:----------:|:--------:|:----------------:|
> > | SeeAct                       |  71.42   |   47.36    |  6.25    |       49.00      |
> > | + $\mathcal{D}_\text{text}$  |  **78.57**   |   55.26    |  18.75   |       57.00      |
> > | + $\mathcal{D}_\text{vision}$|  76.19   |   **60.52**    |  **25.00**   |       **59.00**      |
> >
> >
> > > W3. The authors selected a subset of 100 tasks from Online-M2W without justification for not using the original dataset.
> >
> > A3: Thank you for pointing out that our description of the Online-Mind2Web task subset is not sufficiently clear.   In the paper, we briefly mention our task selection in Section 5.3 (lines 447–450) and provide the full task list and website distribution in Appendix B.3 (pages 16–18), but we did not clearly explain why we chose a subset instead of using all tasks.   We will add a more explicit justification in the final camera-ready version.
> >
> > Unlike our large-scale retrieval evaluation on five offline GUI benchmarks (Section 5.2), the Online-Mind2Web experiments focus on testing WebRAGent’s task execution in real, dynamic web environments.   This setting is more realistic and challenging, but also much less stable: when we initially attempted to run all 300 tasks, we frequently encountered network failures, SSL errors, region-based access restrictions, login verification, and various anti-crawling/Turing-test mechanisms.   Despite engineering efforts to mitigate these issues, overall robustness remained limited.   Moreover, running multiple trials over all tasks would incur a very high token cost.   For both cost and reproducibility, we therefore selected a representative subset of 100 tasks as our online evaluation set.
> >
> > Concretely, we adopt a two-stage task selection procedure. In the first stage, we run all tasks at least three times on six different servers and retain 156 tasks that can be executed reliably. In the second stage, we perform stratified sampling over these tasks: (i) we balance across websites and task categories to preserve website diversity; and (ii) we match the distribution of difficulty labels (easy / medium / hard) in the original dataset to ensure fair comparison.
> >
> > [2] Zheng, Boyuan, et al. "Gpt-4v (ision) is a generalist web agent, if grounded." ICML'24.
> >
> > [3] Abuelsaad, Tamer, et al. "Agent-e: From autonomous web navigation to foundational design principles in agentic systems." arXiv preprint arXiv:2407.13032 (2024).

---

> > > ### Comment · Reviewer_qtrK · 2025-11-27
> > >
> > > Thanks a lot for the thoughtful rebuttal. This resolves W1 and W2, although many papers do utilize Online-M2W successfully, so I am not positive about the W3 response. I will update my score accordingly.

---

> > > > ### Author Response · Authors · 2025-11-29
> > > >
> > > > We deeply appreciate your engagement and are encouraged that W1 and W2 are resolved. Regarding W3, **we acknowledge that evaluating on the full Online-Mind2Web set is a robust standard**. However, our selection was primarily driven by the need to ensure **stability and reproducibility** for fair comparisons, given the dynamic nature of live websites. We verified that our subset covers all difficulty levels evenly. We are glad that this does not overshadow the paper's core contribution on representation learning, and we thank you again for reconsidering the score.

---

### Official Review · Reviewer_amBB · 2025-11-01

**Soundness:** 3
**Presentation:** 3
**Contribution:** 3
**Rating:** 6
**Confidence:** 3

**Summary:**

This paper addresses a critical challenge in the development of autonomous GUI agents: how to effectively learn from and utilize vast amounts of multimodal trajectory data (states, actions, visual observations) that often exceed the context windows of current models.

The authors present a comprehensive framework to tackle this issue:

1. **Unified Agent Trajectory Dataset (UATD):** They first curate and unify five existing GUI agent benchmarks into a standardized dataset, encompassing 7,747 demonstrations and over 82,000 states.
2. **Multimodal Trajectory Retrieval Task:** They formally define a new task, "Multimodal Trajectory Retrieval," to bridge the gap between general-purpose retrieval and agent-centric modeling.
3. **GAE-Bench:** Based on this new task, they construct a large-scale benchmark (GAE-Bench) with 714,628 retrieval pairs, derived from 12 extraction patterns that capture both temporal and semantic relationships.
4. **GAE-Retriever:** They propose a multimodal retriever built on VLM2Vec, employing optimizations like token selection and GradCache to efficiently train on high-resolution image sequences and large batches.
5. **WebRAGent:** Finally, they integrate their retriever into a retrieval-augmented agent framework, WebRAGent, which demonstrates significant performance gains (15-22%) over non-retrieval baselines on the Online-Mind2Web benchmark.

**Strengths:**

1. **Problem Significance:** The paper correctly identifies a timely and critical problem. As trajectory datasets grow, RAG is a logical and necessary step to scale agent capabilities beyond in-context learning.
2. **Foundational Dataset Contribution:** UATD and GAE-Bench are significant contributions in their own right. The engineering effort to unify heterogeneous datasets (web, mobile, desktop) into a standardized format (Section 3.1) is substantial and highly valuable for the community.
3. **Novel Task Formulation:** The "Multimodal Trajectory Retrieval" task, with its 12 extraction patterns (Figure 1), is a key conceptual contribution. This detailed formulation is crucial for training a robust retriever that understands both temporal sequence and semantic intent.
4. **Pragmatic Model Design:** The GAE-Retriever (Section 4.2) is well-designed. Using a VLM (VLM2Vec) backbone instead of CLIP-based models is well-justified for handling arbitrary combinations of multimodal inputs (lines 92-95). The use of token selection and GradCache to tackle the practical constraints of training with multiple high-resolution images is a critical and well-thought-out optimization.
5. **Strong Empirical Validation:** The paper closes the loop by demonstrating that its retrieval model directly translates to a 15-22% success rate improvement in a downstream planning task (WebRAGent). This is a very convincing validation of the entire pipeline.

**Weaknesses:**

While this is an excellent paper, there are a few areas that could be clarified or strengthened:

1. **Justification of "Silver Trajectories":** A key part of GAE-Bench is the semantic retrieval task (q → τ∼, lines 233-239). The authors generate "silver trajectories" via entity substitution. The example given ("Buy a t-shirt for children on Amazon" → "Order a laser printer on eBay," lines 237-239) seems to represent a pair with very **different task flows**, even if the high-level intent ("shopping") is similar.
   - **Concern:** This could be a very noisy training signal. Does retrieving a trajectory for "buying a printer" actually help an agent "buy a t-shirt," or does it introduce confusion?
   - **Recommendation:** The authors should provide a clearer justification for this data augmentation strategy. How is this "silver" pair more helpful than a hard negative?
2. **Architectural Novelty of GAE-Retriever:** The paper calls GAE-Retriever a "novel...framework" (line 90) but also states it is "built on VLM2Vec" (line 91) and uses optimizations (GradCache) from VLM2Vec (line 309).
   - **Recommendation:** The authors should more precisely articulate the **architectural novelty** of GAE-Retriever itself, distinct from VLM2Vec. If the primary novelty lies in the *application* and *task-specific training* (i.e., being the first to successfully apply this architecture to the multimodal trajectory retrieval task), this should be stated clearly.
3. **Lack of Qualitative Analysis:** The 15-22% performance gain is impressive, but the paper does not explain *why* it works.
   - **Question:** What kind of knowledge is being retrieved? Is it high-level planning steps (e.g., "log in first, then search") or low-level interaction details (e.g., "click this specific icon")?
   - **Recommendation:** The paper would be significantly strengthened by adding a qualitative analysis section with 1-2 concrete examples. Show a task where the baseline fails and WebRAGent succeeds, and—crucially—show the *actual retrieved trajectory* that made the difference.
4. **Scope Mismatch (UATD vs. WebRAGent):** The UATD is presented as a highly general dataset unifying "web, mobile, desktop, and embodied environments" (line 69). However, the downstream validation (WebRAGent) is only performed on a web-based benchmark (Online-Mind2Web). This leaves the claims about cross-platform generalization underexplored.

**Questions:**

1. **On "Silver Trajectories":** (See Weakness #1) How do you ensure that the generated silver trajectories share a similar **procedural flow**, rather than just being semantically related at a high level? The t-shirt vs. printer example seems to represent very different procedures.
2. **On Token Selection:** You use a UI-connected graph in RGB space to merge similar patches (lines 299-302). What are the advantages of this over a simpler baseline like bicubic resizing/downsampling of the image? Is there a risk of merging small but critical UI elements (e.g., a checkbox) that are similar in color to their background?
3. **On Inference Latency:** What is the computational overhead (latency) introduced by the GAE-Retriever step during WebRAGent's inference? How does this trade-off against the 15-22% gain in success rate?
4. **On "Hard Tasks":** You mention larger gains on "hard tasks" (line 106). Could you provide a concrete example of a "hard task" and qualitatively explain why retrieval was so beneficial for it?

---

> ### Author Response · Authors · 2025-11-25
> **Response to Reviewer amBB (1/4)**
>
> > W1 & Q1. Response to Concern on "Silver Trajectories" and Training Noise
>
> Thank you for this insightful scrutiny. The decision to pair semantically distinct tasks (e.g., "Buy t-shirt" vs. "Order printer") as positives is a deliberate design choice grounded in **skill abstraction**, rather than noise introduction.
>
> **1. Theoretical Justification: Learning "Skills" over "Parameters"**
>
> Drawing inspiration from recent work like SkillWeaver (Zheng et al., 2025) and Agent Workflow Memory (Wang et al., 2024c), web navigation can be viewed as executing generalized skills (e.g., `Search -> Filter -> Add-to-Cart`) instantiated with specific parameters (e.g., `item="t-shirt"`).
> *   **Decoupling Content from Logic:** Standard retrieval (Gold only) biases the model to focus on specific parameters (visual/textual match of the item). Our Silver Trajectory augmentation acts as a **Hard Positive**, forcing the encoder to disregard these specific parameters and align based on the underlying **procedural flow (the skill)**.
> *   **Bridge for Generalization:** This forces the model to capture the *functional* similarity required for procedural generalization, ensuring that the "shopping skill" is learned independently of the specific product being bought.
> *   **Ambiguity of Hard Negatives:** Furthermore, defining meaningful "Hard Negatives" in trajectory space is notoriously difficult. Metrics based on visual or DOM distance are often unreliable, as visually identical states (e.g., a generic login page or search bar) can be valid steps for completely different tasks. Explicitly constructing Silver Trajectories provides a deterministic signal for learning procedural invariance, avoiding the noise and ambiguity inherent in distance-based negative mining.
>
> **2. Empirical Evidence from Downstream Reasoning (Appendix F)**
>
> The value of retrieving these "silver" examples is verified in our qualitative analysis.
> *   **Case Study:** In Appendix F.3 (Non-RAG vs. RAG Case 1, Page 39), the task is to purchase an iPhone X on Craigslist. Our retrieval logs show that the retriever surfaces trajectories for buying sofas, refrigerators, and other items. Although the target products differ, the retrieved knowledge provides a complete purchasing flow, including the use of multiple filters and other interaction rules. Similarly, in the second task example in Appendix F.5 (Page 60), trajectories for buying phones, headphones, and other electronics help the agent successfully complete a laptop-purchasing task.
> * **Effect:** The model explicitly notes: “The RAG examples guided the agent to explore all sidebar and advanced controls and to adopt heuristic strategies.” Although the specific products differ, the retrieved silver trajectories provide crucial structural knowledge, such as “filters may be hidden in the sidebar,” which prevents the failure modes observed in the Non-RAG baseline. In addition, the retrieved knowledge also offers better troubleshooting strategies and search strategies.
> *   **Planning Mechanism:** Furthermore, our Planner’s system prompt (Snippet 4, Page 29) is explicitly designed to handle this via a “Reuse Level Decision” mechanism. It instructs the agent to identify `HIGH_LEVEL` similarity (borrowing intent/strategy) when element-level details do not match. This prevents silver trajectories from becoming noisy supervision, encourages the agent to use them as structural guidance, and also makes it easier for us to qualitatively analyze how the retrieved knowledge is actually utilized.
>
> **3. Quantitative Support via OOD Generalization**
> While we do not isolate the silver data contribution in a separate ablation, its impact is evident in our **Out-of-Domain (OOD)** results (Table 10).
> *   In OOD settings (e.g., unseen websites), visual and textual exact matches (Gold) are rare or non-existent.
> *   GAE-Retriever achieves **State-of-the-Art results on OOD benchmarks**, significantly outperforming strong baselines like VLM2Vec-V2.2 and ColQwen2-v1.0. This superior generalization capability strongly suggests that the model has successfully learned to retrieve based on **procedural similarity** (enabled by Silver data) rather than relying solely on domain-specific keyword matching.
>
> ***
>
> References:
>
> * SkillWeaver: Boyuan Zheng et al. "SkillWeaver: Web Agents Can Self-Improve by Discovering and Honing Skills." *arXiv preprint arXiv:2504.07079* (2025).
> * Agent Workflow Memory: Zora Zhiruo Wang et al. "Agent Workflow Memory." *arXiv preprint arXiv:2409.07429* (2024).

---

> > ### Author Response · Authors · 2025-11-25
> > **Response to Reviewer amBB (2/4)**
> >
> > >W2. Response to Comment on Architectural Novelty
> >
> > We appreciate your precision regarding the positioning of our model. We fully agree that GAE-Retriever is not a fundamental architectural invention *per se*, but rather a **novel framework adaptation and training recipe** specifically tailored for the nascent task of Multimodal Trajectory Retrieval.
> >
> > We will revise the manuscript (specifically around lines 80-82) to explicitly state that our contribution lies in **successfully adapting VLM2Vec to the trajectory domain**, rather than proposing a new VLM architecture. Specifically, we will highlight that:
> > 1.  **Addressing Domain-Specific Challenges:** Standard VLM retrieval models struggle with the **high memory demand** and **long context** inherent in trajectory data (sequences of high-res screenshots).
> > 2.  **The "Recipe" for Trajectories:** Our novelty lies in identifying and integrating the necessary components—specifically **UI-Graph-based Token Selection** (to reduce visual redundancy in screenshots) and **GradCache** (to decouple batch size from memory constraints)—to make contrastive learning on long multimodal sequences computationally feasible.
> > 3.  **First Successful Application:** To the best of our knowledge, this represents the first framework to effectively train a retriever on such large-scale, long-horizon multimodal agent trajectories, establishing a strong baseline for this new task.
> >
> > Thank you again for helping us clarify our contribution boundaries.
> >
> > > W3. Response to the comment on qualitative analysis
> >
> > First, regarding the question about the type of retrieved knowledge, our method provides both high level planning steps and low level interaction details. The concrete data structure of the retrieved knowledge is defined in Section 4.3, “Step Level Knowledge Retrieval” (lines 348–355). We retrieve step-level experience knowledge aligned with the current state from two knowledge bases.
> >
> >  **(i) In the textual knowledge $\mathcal{D}_{\text{text}}$,** each record contains a natural language description of the observation $\tilde{o}$
> > and the demonstration action $\tilde{a}$ for that step. These natural language descriptions simultaneously encode both high level planning and low level interaction details.
> >
> >  **(ii) In the visual knowledge $\mathcal{D}_{\text{vision}}$,** each record contains the screenshot for the corresponding step and a structured action (operation type, bounding box of the target element, and input value). Although the structured action only covers low level interaction details, our prompt explicitly guides the planning model to infer both high level plans and low level interaction patterns from the screenshot and the associated action. The detailed prompt is given in Appendix E.2, snippet 4 (page 29).
> >
> > To make the retrieved knowledge more concrete, part (3) “Knowledge base” in Figure 4 of the paper (page 7) illustrates representative examples of textual guidance and visual guidance. In addition, we provide complete examples of retrieved knowledge in snippet 5 and snippet 6 in the appendix (pages 61–62), where snippet 5 presents textual guidance examples and snippet 6 presents visual guidance examples.
> >
> > Regarding the qualitative analysis of the performance gains of WebRAGent, we already provide detailed case studies for each retrieval variant in Appendix F. Specifically, Appendix F.1 (pages 30–35) presents two task examples that demonstrate the effectiveness of retrieved textual knowledge for task execution. Appendix F.2 (pages 35–38) visualizes successful cases under the vision based RAG setting. Appendix F.3 (pages 39–51) shows three examples that compare executions with and without Textual RAG, and Appendix F.4 (pages 51–57) provides a visual comparison between executions with and without Visual RAG. In these case studies, the model’s reasoning at each step is highlighted in purple, and the parts that explicitly cite retrieved knowledge are marked in red. After each task example, we qualitatively explain the advantages of the RAG variants and how the retrieved knowledge assists task execution at each critical step. Furthermore, Appendix F.5 (pages 57–60) illustrates the benefits of retrieved knowledge for difficult tasks and compares them with runs without RAG, with a detailed analysis of how retrieval supports decision making at key points.

---

> > > ### Author Response · Authors · 2025-11-25
> > > **Response to Reviewer amBB (3/4)**
> > >
> > > >W4. Response to the scope mismatch (UATD vs. WebRAGent)
> > >
> > > We appreciate the reviewer’s careful comment on the relation between the scope of UATD and the evaluation of WebRAGent. We would like to clarify that our proposed UATD dataset is constructed from five public web agent benchmarks (Mind2Web, AutoWebGLM, WebArena, WebLINX, and GUIAct), covering diverse digital web environments. The wording in line 69 was imprecise: our intention was to emphasize that the pipeline used to construct UATD can in principle be extended to mobile, desktop, and embodied environments, highlighting the **extensibility of our unified trajectory representation and retrieval framework**, rather than claiming that the current UATD already includes these platforms. The core contribution of our work is that UATD, GAE Bench, and GAE Retriever together provide a unified trajectory representation across multiple web environments. Table 3 shows that GAE Retriever outperforms current state of the art retrieval models on all five datasets. A breakdown by task (Recall@5 under both in domain and out of domain settings) is provided in Appendix D.4 (Figure 7). In addition, we evaluate all baseline methods described in Subsection 5.1, together with our proposed GAE Retriever, on 12 retrieval subtasks spanning all data sources, and report Recall@1/5/10. The results for the in domain setting are given in Table 9, while Table 10 reports the out of domain results.
> > >
> > > For downstream validation, we consider Online-Mind2Web, which consists of live web pages, to be a representative and challenging evaluation environment. Due to the substantial engineering cost and space limitations, we currently perform end to end validation only in the web setting, but our method is designed to be plug and play and can be extended to additional downstream applications, which we view as an important direction for future work. We will correct the wording in the final version of the paper and explicitly discuss this point in the future work section.
> > >
> > > >Q2. Response to the comment on token selection and patch merging risk
> > >
> > > I understand your concerns about both the superiority of our method and the potential risk of losing small yet critical UI elements. First, as we mentioned in Appendix C.2 (lines 1012–1013) of the paper, the UI graph token selection module is only applied during training, with the primary goal of saving GPU memory and enabling a larger batch size. During inference, however, we feed the full-resolution patch grid produced by the standard Qwen-VL preprocessing pipeline.
> > >
> > > * **Comparison with bicubic resizing/downsampling:** Bicubic resizing/downsampling uniformly scales the entire image, which leads to more severe information loss and cannot effectively distinguish the characteristics of different patches. In contrast, our method operates on web page images with a fixed resolution and merges only locally similar regions, thereby avoiding excessive information loss. Our key assumption is that regions with large color differences usually carry more information, whereas regions with small color differences tend to be less informative. Consequently, our method mainly merges low-information areas on web pages (such as large blank regions or flat, solid-color areas), while it is unlikely to merge regions with clear boundaries or variations in color and texture (such as buttons, checkboxes, and text).
> > >
> > > * **Risk of losing small UI elements:** We only merge patches that are spatially adjacent and whose color difference is below a certain threshold (i.e., $\lVert \mathbf{c}_i - \mathbf{c}_j \rVert_2 \le 1$ in RGB space). In typical cases, common buttons or checkboxes exhibit a noticeable contrast against the background (as shown in the left part of Figure 7). If a checkbox is so similar to the background that it is indistinguishable even to the human eye (as in the right part of Figure 7), then in such an extreme case the agent would also be unable to distinguish it. We therefore treat this checkbox as effectively nonexistent, which is reasonable from the agent’s perspective. Moreover, as we emphasized earlier, even in extreme cases where a very small number of UI elements might be merged away, this can only happen during training; at inference time, we still feed the full patch grid as input.

---

> > > > ### Author Response · Authors · 2025-11-25
> > > > **Response to Reviewer amBB (4/4)**
> > > >
> > > > > Q3. Response to the comment on inference latency
> > > >
> > > > In WebRAGent, each planning step introduces an additional round of GAE Retriever encoding, vector retrieval, and LLM reranking. Compared with the baseline, this does introduce some extra computation, but the overhead is very small relative to the entire task execution: it results in roughly 10% additional latency per step, which we consider fully acceptable in practice. We have applied several engineering optimizations to the retrieval and LLM reranking stages, including a two level cache on the LLM side and optimized and compressed image encoding. Our efficiency experiments are mainly conducted on an NVIDIA RTX A6000 GPU. The results show that the dominant time cost does not come from the GAE Retriever retrieval stage, but from the LLM reranking, while the overall time per step remains modest. When the cache is hit, most queries require almost no extra reranking time and the corresponding inference cost is nearly negligible. These experimental results have been added to Appendix D.7 of the revised manuscript, and we will also include them in the final version of the paper.
> > > > | Setting              |  Retrieval(s/step) | Reranking(s/step) | Total(s/step) |    Speedup |
> > > > |----------------------|------------------------------------------|-----------------------------------|-------------------------|--------------------|
> > > > | No Cache             | 0.3877                                   | 3.5374                            | 3.9251                  | 1.0×               |
> > > > | With Cache           | 0.3828                                   | 0.0002                            | 0.3830                  | 9.24×              |
> > > >
> > > > >Q4. Response to the comment on "hard tasks" and retrieval benefits
> > > >
> > > > We present three task examples in textual form in Appendix F.3 (pages 39–51) and one task example in visual form in Appendix F.4 (pages 51–57) to compare the execution behavior of the Non-RAG and RAG agents. In these examples, the model’s thought process at each step is highlighted in purple, and the parts where retrieved knowledge is explicitly referenced are marked in red. After each task example, we provide a qualitative explanation of the advantages of the RAG approach and how the retrieved knowledge assists task execution at each critical step.
> > > >
> > > > For difficult cases, we present two challenging tasks and their key steps in Appendix F.5 (pages 57–60), and we explain in detail why retrieval is particularly beneficial for such tasks. First, in the first difficult task, 50% of the steps make use of retrieved knowledge. The retrieved knowledge provides a global strategy in the early steps and offers the correct filtering strategy when executing critical filtering operations. When the agent encounters issues or fails to update the state in this long-horizon task, the retrieved knowledge supplies an emergency “reset-and-recover” solution. Second, in the second task, the retrieved knowledge provides better search strategies at key steps, preventing blind exploration. This allows the RAG agent to complete the task more efficiently than the Non-RAG agent.

---

### Official Review · Reviewer_QTLy · 2025-11-01

**Soundness:** 2
**Presentation:** 2
**Contribution:** 2
**Rating:** 2
**Confidence:** 3

**Summary:**

This paper introduces WebRAGent, a framework for retrieval-augmented multimodal web agent planning. The motivation is that progress in multimodal trajectory learning is limited by the difficulty of representing rich visual information within long interaction histories that exceed a model’s context window.
To address this, the authors propose multimodal trajectory retrieval, along with:
- A benchmark for trajectory-based retrieval pairs,
- A model (GAE-Retriever) for multimodal retrieval based on a vision-language backbone, and
- A retrieval-augmented web agent integrating the retriever into agent planning.

**Strengths:**

- The paper presents a novel and intuitive idea, aiming to connect retrieval-augmented generation with multimodal web agent reasoning.
- It is well-written and easy to follow, with clear structure.
- The release of code and resources is great and improves reproducibility.

**Weaknesses:**

- The motivation for the multimodal trajectory retrieval task is weak and needs clearer justification—why is this problem important, and what real-world gap does it fill?
- The introduction of the GAE-Retriever lacks sufficient motivation and integration into the overall narrative.
- Figures 1 and 3, and Tables 1, 3, 9, and 10, are difficult to read due to poor formatting and small font sizes.
- The limitations of the proposed approach are not discussed.
- The related work section oversimplifies the historical relationship between retrieval and generation in the context of multimodal retrieval. Multimodal retrieval methods have existed long before generation-based approaches became dominant.
- The overall storyline feels disjointed: it is not entirely clear how the benchmark, retriever, and agent components connect to form a coherent research contribution.

**Questions:**

- Could the authors clarify how the benchmark, retriever, and agent fit together conceptually and experimentally within one unified framework?
- Please expand on the motivation behind the proposed multimodal trajectory retrieval task; why is it necessary, and what unique challenges does it address?
- It would be helpful to include a single overview figure illustrating how all components (benchmark, retriever, agent) interact within the proposed system.

---

> ### Author Response · Authors · 2025-11-24
> **Response to Reviewer QTLy (1/2)**
>
> We thank you for the critical and insightful feedback. We realize that our title, *WebRAGent*, and the extensive experiments on the downstream agent application may have inadvertently obscured the paper's primary scientific contribution and narrative arc.
>
> ---
> ### Clarification on the Core Contribution
> The core motivation of this work is *not merely to build a retrieval-augmented agent, but to establish the foundation for Multimodal Trajectory Representation Learning*.
> As embodied and GUI agents generate exponential amounts of multimodal trajectory data, a critical bottleneck emerges: *How do we effectively represent, compress, and retrieve these long-horizon, visual-heavy histories?*
> Our work addresses this by:
> 1.  **Defining the Task**: Formalizing the alignment between multimodal states, trajectories, and textual queries.
> 2.  **Building the Infrastructure**: Constructing *UATD* and *GAE-Bench* to support this representation learning.
> 3.  **Training the Model**: Proposing *GAE-Retriever*, the first model successfully trained to capture fine-grained trajectory semantics.
> 4.  **Validation**: Using *WebRAGent* as a downstream proof-of-concept to demonstrate that better trajectory representations directly translate to better agent performance.
>
> We will revise the title (e.g., to *Learning Multimodal Trajectory Representations for Web Agent Planning*) and the introduction in the next revision to firmly center the paper on representation learning.
>
> ---
> ### Response to Specific Comments
> > Q1: Motivation for Multimodal Trajectory Retrieval (Weakness 1 & Question 2)
>
> The question is why this task is necessary and what gap it fills. We argue that *Multimodal Trajectory Retrieval is a "Missing Layer" in current Agentic AI*, motivated by three key factors:
>
> 1.  **The Data Explosion & Context Bottleneck**: Future agents will have access to millions of hours of interaction logs. Current LLMs cannot process this raw history due to context limits. We need a way to map these *multimodal* experiences into a structured, retrievable latent space, essentially "indexing" agent experiences.
> 2.  **Bridging the Modality Gap**: Prior trajectory retrieval methods primarily rely on text, either matching queries to *textual descriptions* of trajectories (Kim et al., 2024) or matching *DOM/HTML* states (Zheng et al., 2024). However, GUI agents are inherently visual; determining functional similarity often requires visual semantics (e.g., spatial layout, icon status, rendering issues) that text or DOM trees miss. **Our analysis in Appendix D.5 confirms this**: our multimodal, state-conditioned retrieval significantly outperforms the conventional text-only trajectory retrieval baseline ($q \rightarrow \tau$) by providing more precise, step-level visual guidance.
> 3.  **Potential Future Directions**: A unified trajectory encoder is a prerequisite for advanced agentic downstream tasks beyond simple RAG, including:
>     *   **World Modeling**: Predicting future visual states by retrieving and conditioning on similar previous states from history (Gu et al., 2025).
>     *   **Retrieval-Augmented RL**: Accelerating training convergence and improving sample efficiency by retrieving high-value experiences to guide exploration and policy learning (Goyal et al., 2022).
>     *   **Cross-Task Generalization**: Transferring skills across different environments by aligning semantically similar subtrajectories in the latent space.
>
> ---
>
> ### References
> * Gu, Y., Zhang, K., Ning, Y., Zheng, B., Gou, B., Xue, T., Chang, C., Srivastava, S., Xie, Y., Qi, P., Sun, H., Su, Y. (2025). *Is Your LLM Secretly a World Model of the Internet? Model-Based Planning for Web Agents*. TMLR 2025.
> * Kim, M., Bursztyn, V., Koh, E., Guo, S., & Hwang, S. W. (2024). *RaDA: Retrieval-Augmented Web Agent Planning with LLMs*. Findings of the Association for Computational Linguistics: ACL 2024.
> * Zheng, L., Wang, R., Wang, X., & An, B. (2024). Synapse: Trajectory-as-exemplar prompting with memory for computer control. International Conference on Learning Representations (ICLR 2024).
> * Goyal, A., Friesen, A. L., Banino, A., Weber, T., Ke, N. R., Badia, A. P., ... & Blundell, C. (2022). *Retrieval-Augmented Reinforcement Learning*. International Conference on Machine Learning (ICML).

---

> > ### Author Response · Authors · 2025-11-24
> > **Response to Reviewer QTLy (2/2)**
> >
> > ### Response to Specific Comments (cont.)
> > > Q2: How components fit together / Disjointed Storyline (Weakness 2,6 & Question 1,3)
> >
> > The benchmark, retriever, and agent form a unified *"Data-Model-Application" pipeline*, designed to solve the representation challenge:
> >
> > 1.  **UATD & GAE-Bench (The Data Layer)**: We first needed to unify heterogeneous trajectory formats (data from Mind2Web, WebArena, etc.) into a standardized structure to make representation learning possible. This fills the data gap.
> > 2.  **GAE-Retriever (The Representation Layer)**: Using this data, we trained the retriever to learn high-quality embeddings. This addresses the core modeling challenges, including processing high-resolution screenshots within ultra-long contexts through token selection, and scaling training by expanding the global batch size to increase negative samples via GradCache.
> > 3.  **WebRAGent (The Application Layer)**: Finally, to prove these embeddings are actually *useful* for downstream applications, such as decision-making(not just abstract vectors), we integrated them into an agent. The 15-22% performance gain serves as the experimental validation of the representation model's value.
> >
> > We will add the **overview figure** requested to visualize this pipeline: showing raw trajectories flowing into the Benchmark construction, training the Retriever, and finally the Retriever feeding the Planner (WebRAGent).
> >
> > > Q3: Related Work & Historical Context (Weakness 5)
> >
> > We appreciate the correction regarding the history of retrieval. Our intention was to contextualize the discussion within the specific niche of **Autonomous GUI Agents**, which currently relies predominantly on "Reactive Generation" (predicting action $a_t$ directly from $o_t$). Our work aims to pioneer a paradigm shift towards "Retrieval-Augmented Planning" by **introducing the necessary infrastructure for trajectory representation**.
> >
> > Besides, our trajectory modeling involves multiple images for web-agent planning, which can be viewed as low-framerate, high-definition video. Traditional CLIP-based retrieval methods, such as composed video retrieval (Thawakar et al., 2025), are not suitable for our setting because CLIP is limited to single-image inputs with a maximum resolution of 224×224, far below the typical 1,000×1,000 multi-image inputs in our trajectories. We appreciate the reviewer’s suggestion that retrieval is a foundational concept in the broader IR literature. We will revise the related-work section to properly cite classical multimodal retrieval research and clearly articulate how our problem setting differs from standard CLIP-style retrieval, stating why these approaches do not directly apply to agentic trajectory modeling.
> >
> > > Q4: Visuals and Limitations (Weakness 3 & 4)
> >
> > *   **Visuals**: We will significantly improve readability. We will increase font sizes in Figures 1 & 3 and reformat Tables 1, 3, 9, and 10 to be less dense in the next revision.
> > *   **Limitations**: We will add a dedicated discussion on limitations to provide a balanced view:
> >     1.  **Computational Cost**: Encoding high-resolution visual trajectories involves significantly higher computational and memory costs compared to text-only methods, presenting a trade-off between representational fidelity and inference efficiency.
> >     2.  **Scope of Downstream Validation**: While we position our representation as a foundation for broader agentic capabilities (e.g., World Modeling, RL training), this paper experimentally validates it primarily through **Retrieval-Augmented Planning**. Verifying its efficacy in other paradigms remains an important direction for future work.
> >     3.  **Data Domain Concentration**: Our current experiments focus on **Web Agent** data. While the methodology is generalizable, we have not yet scaled the training and evaluation to Desktop, Mobile, or Embodied environments, which limits our current assessment of cross-platform generalization. We aim to leave room for future work that extends our agent trajectory modeling and retrieval to broader domains.
> >
> > ---
> > ### References
> > * Thawakar, O., Naseer, M., Anwer, R. M., Khan, S., Felsberg, M., Shah, M., & Khan, F. S. (2024). *Composed video retrieval via enriched context and discriminative embeddings*. In Proceedings of the IEEE/CVF Conference on Computer Vision and Pattern Recognition (pp. 26896-26906).

---

> > > ### Comment · Reviewer_QTLy · 2025-11-27
> > > **Reply**
> > >
> > > Thank you for your detailed responses to my questions. I appreciate your efforts in clarifying the points that were causing confusion for me. I now have a much better understanding of most of the aspects that were unclear.
> > >
> > > To be able to reevaluate the paper contribution, I would like to see the revised version of the paper. Would it be possible to include the revised version of the paper? It would help continue the discussion.

---

> ### Author Response · Authors · 2025-12-02
>
> We deeply appreciate your engagement and are encouraged that we were able to address most of your questions. We have revised the paper as requested and uploaded the latest version.

---

### Author Response · Authors · 2025-12-03
**General Responses and Revision Log (1/2)**

Dear Area Chairs,

We sincerely thank the Area Chairs for taking the time to evaluate our submission, and we are deeply grateful to all reviewers for their valuable time and insightful comments. We are encouraged that the reviewers have reached a consensus on the core contributions of our work and have particularly appreciated the following strengths:

• **Highly valuable dataset and benchmark contributions:** Our unified agent trajectory dataset (UATD) and GAE-Bench have been recognized by the reviewers as a **valuable resource** for evaluating multimodal trajectory retrieval (qtrK) and as an **innovative dataset construction** (bk79). By introducing multiple retrieval schemas, including text-to-state, text-to-trajectory, state-to-state, etc., we provide valuable and standardized resources for downstream research on multimodal trajectory retrieval (amBB, qtrK, bk79).

• **Innovative and intuitive agent framework design:** Most reviewers find our idea of combining multimodal trajectory retrieval + retrieval-augmented web agents to be both novel and intuitive (QTLy, qtrK). Our proposed WebRAGent framework is regarded as a **structurally sound and modular innovation** (bk79) that systematically connects retrieval-augmented generation (RAG) with GUI agent reasoning. This design filters and compress the long horizon history and mitigates the representation and reasoning challenges caused by long interaction histories exceeding the context window (QTLy, amBB, qtrK, bk79).

• **Strong empirical performance and completeness of the work:** Our work has been described as **“comprehensive and substantial”** (bk79), as it thoroughly covers all key components from unifying multi-source data and constructing a multimodal trajectory retrieval benchmark, to training the retriever and evaluating online web agents (amBB, bk79). GAE-Retriever supports efficient contrastive learning over high-resolution image sequences with large batch sizes, and its retrieval performance significantly surpasses strong multimodal baselines on five datasets. On Online-Mind2Web, WebRAGent improves the success rate over non-retrieval baselines by 15–22%, clearly demonstrating the practical benefits of multimodal trajectory retrieval for web agent planning (amBB, qtrK, bk79).

• **Good readability and reproducibility:** Reviewers consistently find the paper well-structured, clearly written, and easy to follow, and they particularly appreciate our efforts to open-source the code and resources to enhance experimental reproducibility. This further increases the practical value and impact of our work for the community (amBB, qtrK, bk79).

## Reiterating the Motivation and Unique Contribution to the Community

Beyond the strengths already acknowledged above, we would like to further clarify the motivation of this work and highlight the unique perspective it offers to the ICLR community.

We argue that **Representation Layer is a missing and previously overlooked layer in current Agentic AI architectures**. With the exponential growth of interaction logs, there is an urgent need to map heterogeneous agent states and actions into a unified latent space, allowing past experiences to be efficiently reused across tasks.

To address this, our work constructs a full-stack solution from the **Data Layer (UATD) → Representation Layer (GAE-Bench & GAE-Retriever) → Application Layer (WebRAGent)**. This architecture provides the necessary infrastructure to ground agent planning in massive-scale historical knowledge.(refers to Figure 1)

We believe this perspective provides a previously underexplored yet crucial contribution to the community:

**(i) Breaking the Data Explosion & Context Bottleneck:** We map multimodal agent experiences into a structured, retrievable latent space, effectively building an index over long horizon and addressing the challenge of representing dense visual information without loss in long-term trajectory histories.

**(ii) Bridging the Modality Gap:** Compared to text/DOM-based retrieval, our multimodal retrieval provides more precise visual guidance for GUI agents.

**(iii) Foundation for Future Directions:** Our framework enhances agent planning capabilities and opens up multiple promising directions for future work, such as **World Modeling** (predicting future states based on similar past experiences), **Retrieval-Augmented RL** (guiding policy learning with high-value retrieved experiences), and **Cross-Task Generalization** (aligning and reusing semantically similar sub-trajectories across different tasks and environments).

---

> ### Author Response · Authors · 2025-12-03
> **General Responses and Revision Log (2/2)**
>
> ## Revision Log
>
> We have carefully revised the paper based on the reviewers’ comments, including polishing the writing and adding new experiments and analyses (with the main updates highlighted in **blue** in the revised version).
>
> • We have revised the paper title to “Learning Multimodal Trajectory Representations for Web Agent Planning.” This change has already been reflected in the revised PDF, and we will update the title accordingly in the OpenReview system at the camera-ready stage. (QTLy)
>
> • We have added an overview paragraph in Section 1 describing the overall structure of the paper and a corresponding overview figure (Figure 1) to make the narrative clearer. (QTLy)
>
> • We have expanded Section 2.2 with additional related work and historical background on retrieval, and we have clarified the limitations of traditional CLIP-based retrieval methods. (QTLy)
>
> • We have added a discussion of limitations and future work in Section 6. (QTLy)
>
> • We have supplemented Appendix B.3 with a detailed explanation of the reasons and procedure for selecting the Online-Mind2Web subset. (qtrK)
>
> • We have added a new study in Appendix D.6 where we integrate multimodal trajectory retrieval knowledge into the SeeAct framework, demonstrating the effectiveness of our approach across different agent frameworks; we have also updated Table 4 in Section 5.3 accordingly. (qtrK, bk79)
>
> • We have added an experiment on the inference efficiency of GAE-Retriever in Appendix D.7, where we carefully analyze the latency introduced by retrieval and LLM-based reranking. (qtrK, bk79)
>
> • We have added a comparison between Token Selection and alternative methods in Appendix D.8, and we provide a detailed analysis of the risks of merging small, critical UI elements that share similar backgrounds. (amBB)
>
> • We have added the prompt for reward (Snippet 5) and the prompt for LLM-based reranking (Snippet 6) in Appendix E.2. (bk79)
>
> • We have added a qualitative analysis of a hard task example in Appendix F.5 and provided a detailed discussion of how retrieval knowledge benefits the execution of such hard tasks. (amBB)
>
> Notably, our responses have already received positive feedback from the reviewers which we think can serve as a good indicator: reviewer QTLy and qtrK indicated that most of their concerns had been resolved, and reviewer qtrK raised his score.
>
> Best regards,
>
> Authors

---

### Meta-Review · Area_Chair_12nh · 2026-01-10

**Summary:**

Reviewers are divided on this work. **Reviewer amBB** supports the contribution, finding value in UATD and GAE-Bench. **Reviewer qtrK** and **Reviewer bk79** see the idea as interesting but question novelty relative to Aguvis. **Reviewer QTLy** finds the narrative fragmented and motivation weak.

Key concerns include: conceptual disconnect between components, limited evaluation (100 tasks from Online-M2W), missing SOTA baselines, and unclear dataset novelty. The rebuttal added SeeAct baseline and clarified extraction patterns, but did not resolve the core issues. Only **Reviewer amBB** leans toward acceptance.

**Reviewer Concerns:**

Motivation for the work is weak. The rebuttal did not change the perception of the work as conceptually disjointed. This reviewer remains unconvinced that the proposed retrieval task and model are distinct or necessary. Presentation quality in figures and tables also remains an issue.

This reviewer is satisfied with the clarifications on silver labels and data splits. Practical gains are seen as meaningful. The rebuttal directly addressed concerns about data overlap and reward design through additional ablations.

Worries persist about the small evaluation subset of 100 tasks. Adding the SeeAct baseline helped but UATD's novelty relative to Aguvis remains a concern. The GUI-centric extraction patterns are noted but do not fully resolve doubts about the dataset's distinctiveness.

The reviewer appreciates the extra ablations on reward design and training. However, the GAE-Retriever seems incremental and missing strong web-search baselines is a major drawback. Evaluation bias remains a worry since training and testing data come from the same source.

**Reviewer Scores:**

**Reviewer QTLy (Original: 2 → Predicted: 2):**
The rebuttal clarifies the GUI-centric nature of UATD but fails to fix the fragmented narrative. The conceptual contribution is not compelling enough for a higher score.

**Reviewer amBB (Original: 6 → Predicted: 6):**
Detailed descriptions of label generation and validation addressed the main concerns. This reviewer maintains a weakly positive stance.

**Reviewer qtrK (Original: 4 → Predicted: 4):**
The justification for the Online-M2W subset is noted. Still, the structural concerns regarding novelty and evaluation breadth prevent a score increase.

**Reviewer bk79 (Original: 4 → Predicted: 4):**
Clarity on annotation and training improved. Deeper concerns about missing strong baselines and incremental novelty remain unaddresse

---

### Decision · Program_Chairs · 2026-01-26

Reject